# 📖🔍 SCHOLAREVAL: RESEARCH IDEA EVALUATION GROUNDED IN LITERATURE

## ABSTRACT

As AI tools become increasingly common for research ideation, robust evaluation is critical to ensure the validity and usefulness of generated ideas. We introduce SCHOLAREVAL, a retrieval-augmented evaluation framework that assesses research ideas based on two fundamental criteria: *soundness*—the empirical validity of proposed methods based on existing literature, and *contribution*—the degree of advancement made by the idea across different dimensions relative to prior research. To evaluate SCHOLAREVAL, we introduce SCHOLARIDEAS, the first expert-annotated dataset of multi-domain research ideas and reviews, comprised of 117 ideas across four disciplines: artificial intelligence, neuroscience, biochemistry, and ecology. Our evaluation shows that SCHOLAREVAL achieves significantly higher coverage of points mentioned in the human expert annotated rubrics in SCHOLARIDEAS compared to all baselines. Furthermore, SCHOLAREVAL is consistently preferred over our strongest baseline o4-mini-deep-research, a reasoning and search-enabled agentic system by OpenAI, in terms of evaluation actionability, depth, and evidence support. Our large-scale user study also shows that SCHOLAREVAL significantly outperforms deep research in literature engagement, idea refinement, and usefulness. We will openly release our code, dataset, and SCHOLAREVAL tool for the community to use and build on.

## 1 INTRODUCTION

Research ideation stands out as one of the most critical and challenging steps in scientific research, where the success of a research project fundamentally hinges on the technical soundness of the underlying idea and its potential to advance the field. To accelerate this stage, multiple works have developed AI-based systems for research ideation (Wang et al., 2024; Si et al., 2025b; Garikaparthi et al., 2025; Gottweis et al., 2025; Baek et al., 2025). Although AI-generated ideas score higher than human ideas on criteria such as human-evaluated novelty at the *ideation* stage (Si et al., 2025b) , many of these seemingly promising ideas turn out to be ineffective when *executed* (Si et al., 2025a). The execution of faulty ideas can lead to substantial costs, particularly in fields requiring significant computational resources or wet-lab experiments. There is thus a critical need to rigorously evaluate research ideas pre-execution to prioritize the strongest ones for resource investment.

Despite this need, automatic research idea evaluation remains an underexplored area. Some works narrowly frame it as a prediction task: deciding which idea among a pair would lead to better results on predefined benchmarks (Wen et al., 2025). Others focus on one-dimensional approaches to idea evaluation (e.g., novelty) (Afzal et al., 2025; Shahid et al., 2025), or target only specific sub-disciplines (e.g., AI) (Si et al., 2025b). Most importantly, these systems either only generate scores or are limited to sparse feedback such as short rationale statements (Feng et al., 2025; Wen et al., 2025; Shahid et al., 2025). However, to create AI co-scientists that can generate and refine research ideas, giving dense, actionable, and multifaceted feedback is crucial (Wu et al., 2023; Cao et al., 2024). To the best of our knowledge, no existing work addresses the challenge of comprehensive research idea evaluation across disciplines within a framework that provides detailed and actionable feedback for idea refinement. To address this gap, we introduce SCHOLAREVAL (Figure 1), a research idea evaluation system grounded in the most recent literature. SCHOLAREVAL evaluates research ideas based on two fundamental criteria: *soundness* and *contribution*. **(1) Soundness** refers to the empirical validity of each proposed method in the research plan, assessed by systematically

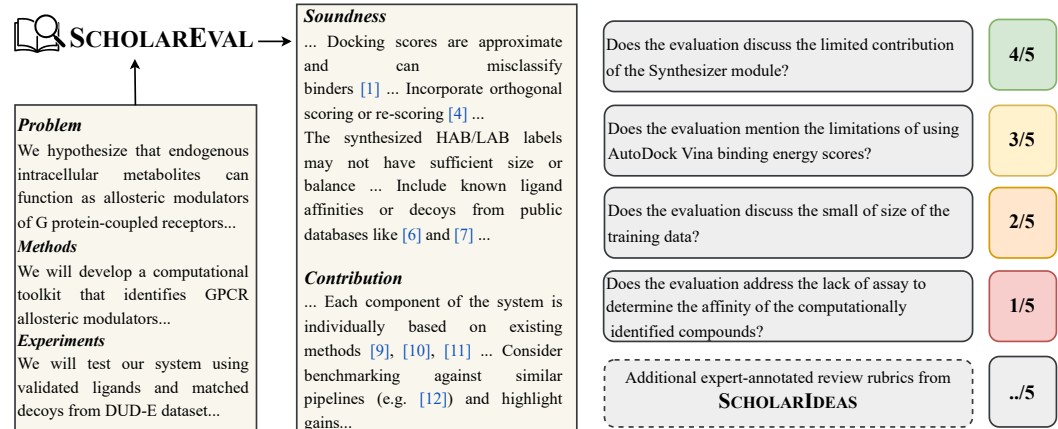

Figure 1: **Left:** Given a research idea SCHOLAREVAL generates a literature-grounded evaluation based on soundness and contribution. **Right:** To evaluate SCHOLAREVAL, we measure the degree of coverage of expert-annotated review rubrics in SCHOLARIDEAS. The final coverage score is the average over all rubrics.

examining whether similar applications of this method in existing literature have demonstrated success or failure. **(2) Contribution** refers to the degree of advancement a research idea offers across different dimensions–e.g., its proposed methodology, data, evaluation approaches, and conceptual framework–relative to existing literature. The rationale for dimension-based evaluation is that novelty is multi-faceted by nature, and an idea can be considered novel relative to certain aspects of prior work, rather than being categorically novel or not (Rubaiat et al., 2025; Radensky et al., 2025). Given a research idea detailing the problem, proposed methodology, and planned experiments, SCHOLAREVAL employs a multi-stage pipeline that first generates targeted search queries to retrieve a large volume of related literature from Semantic Scholar (Kinney et al., 2025) (Figure 2). It then extracts key information from the retrieved literature to assess the research plan's soundness and contribution. Finally, it synthesizes detailed feedback supported by relevant citations.

To evaluate SCHOLAREVAL, we construct SCHOLARIDEAS, a multi-disciplinary dataset of 117 research ideas and their corresponding reviews validated by subject-matter experts across four disciplines: artificial intelligence, biochemistry, neuroscience, and ecology. As showcased in Figure 1, reviews in SCHOLARIDEAS are composed of multiple rubrics, each focusing on a specific point that the evaluation should address, for 1076 rubrics in total. We develop a multi-faceted automatic evaluation framework to assess SCHOLAREVAL against strong baselines, namely state-of-the-art LLMs and deep research systems. Our results show that SCHOLAREVAL achieves greater coverage of the expert-annotated review rubrics in SCHOLARIDEAS, significantly outperforming all baselines and surpassing o4-mini-deep-research by over 20% relative improvement. Our results also demonstrate that SCHOLAREVAL is consistently preferred over deep research in terms of evidence support, depth, and actionability.

Our human study involving 18 experts and 46 evaluations further supports the real-world usefulness of SCHOLAREVAL across our four target disciplines. SCHOLAREVAL outperforms the strongest baseline, deep research, by a significant margin in metrics tied to our core contributions: literature engagement, citation use, feedback validity, idea refinement, and focus on relevant evaluation aspects. The expert evaluators also found our system more useful and were eager for its official release.

Our work makes the following major contributions:

- SCHOLAREVAL, a literature-grounded framework that comprehensively evaluates research ideas based on their soundness and contribution with actionable feedback. We will openly release the SCHOLAREVAL tool and user interface.

- SCHOLARIDEAS, an expert-annotated dataset for research idea evaluation spanning four disciplines and composed of 117 research ideas with 1076 detailed review rubrics.

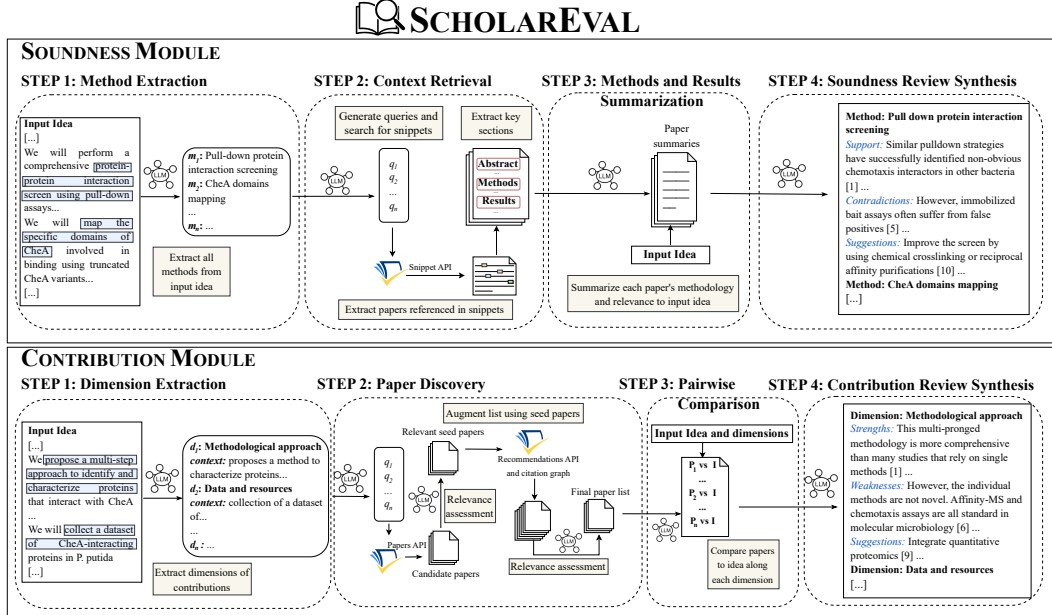

Figure 2: Overview of SCHOLAREVAL and its two modules. **Top:** The soundness module extracts the methods proposed in the research idea and conducts a thorough literature search for similar applications of each method to determine its potential effectiveness. **Bottom:** The contribution module identifies the dimensions along which the research idea is making contributions and conducts detailed comparisons with related papers along each dimension to identify areas of novelty or lack thereof.

- A multifaceted evaluation for long-form research idea review responses, including automatic metrics and a carefully designed human expert evaluation.

## 2 SCHOLAREVAL

SCHOLAREVAL is a retrieval-augmented, multi-stage pipeline designed to give an in-depth evaluation of research ideas based on their soundness and contribution.

**Task Formulation.** Given a research idea $I$, the task is to find papers $P = \{p_1, p_2, p_3, \ldots\}$ that are highly relevant to $I$ and synthesize their findings in the context of $I$ to generate a comprehensive evaluation $\mathcal{E} = (S, C)$ covering the soundness and contribution of the research idea. The evaluation is accompanied by citations, ensuring that all claims are supported by evidence from the literature.

**Overview of SCHOLAREVAL.** As shown in Figure 2, SCHOLAREVAL is composed of two main modules: **Soundness** and **Contribution**. We present both their workflows in §2.1 and §2.2 and provide further details in Appendix C.

### 2.1 SOUNDNESS EVALUATION

The soundness module evaluates the methodological rigor of an idea by extracting methods, searching for relevant literature, and synthesizing evidence to assess whether the proposed approaches are well-supported or contradicted by existing work.

**Method Extraction.** The first objective of the soundness pipeline is to extract distinct methodological components including algorithmic approaches, experimental designs, evaluation protocols, ablation studies, or analytical frameworks from the idea. Formally, we leverage an LLM, referred to hereafter as $\mathcal{M}$, to extract all methods $M = \{m_1, m_2, \ldots, m_k\}$ from the research idea $I$.

**Context Retrieval.** This step gathers information from the literature about the effectiveness of the proposed methods. For each extracted method $m_i \in M$, $\mathcal{M}$ generates a relevant query for

Semantic Scholar snippet search (Kinney et al., 2025), which indexes 285.6M passages extracted from the title, abstract, or body of research papers (Singh et al., 2025). Since the queries are constructed from the description of the method $m_i$, it is likely to retrieve snippets within relevant methodology sections. We extract all papers referenced in these snippets, which provides a dense collection of relevant sources to broaden the understanding of $m_i$. We download the full text of these papers and parse them using GROBID, a state-of-the-art document parsing tool, to extract three key sections from each paper: the methods section, to compare its similarity with the current method $m_i$; the results section, to judge the method's effectiveness in a given context; and the abstract for a holistic view of the paper. At the end of this process, we obtain a list of related papers $P_i = \{p_{i,1}, p_{i,2}, \ldots, p_{i,n_i}\}$ for each method $m_i$, where each paper is represented as a triplet (abstract, methods, results). This list constitutes essential literature context to evaluate each method's effectiveness.

**Methods and Results Summarization.** This stage serves two key functions: first, it filters out extracted papers that are not relevant to assessing the method $m_i$, and second, it condenses the most vital information from relevant papers, since retaining all paper data results in prohibitive context lengths. Specifically, for each paper $p_{i,j} \in P_i$, we instruct $\mathcal{M}$ to identify whether the methodology described in the paper is relevant to the method $m_i$, and if so, generate a compact summary of its methods and results grounded in the context of the method $m_i$ and the overall research idea $I$.

**Soundness Review Synthesis.** Grounded in the condensed context, the soundness pipeline concludes by synthesizing method-level soundness evaluations. Specifically, $\mathcal{M}$ analyzes the paper summaries in the context of the method $m_i$ and the research idea $I$ to synthesize three main sections: (1) *Support*: the support for the method $m_i$ based on the literature. This section details whether there are similar methods in the literature that have shown successful results, and uncovers how they relate to the current $m_i$. (2) *Contradictions*: the contradictions to the method $m_i$ based on the evidence extracted from the literature. In relation to the proposed method $m_i$, this section highlights the limitations that methods in a similar context have faced, signaling its potential ineffectiveness. (3) *Suggestions*: based on the strengths and limitations of the method identified in the previous sections, $\mathcal{M}$ generates actionable suggestions for improvement to refine the proposed methodology.

## 2.2 CONTRIBUTION EVALUATION

The contribution module assesses the novelty and significance of an idea by identifying contribution dimensions, discovering related papers, conducting pairwise comparisons to determine how the idea advances the literature, and synthesizing a dimension-level contribution review.

**Dimension Extraction.** The initial step of contribution evaluation consists of extracting the dimensions along which the research idea is making contributions to the field. Dimensions represent the facets of the idea's potential contributions that are specific and comparable across related literature (e.g., system design, data collection, evaluation methodology, etc.). Instead of imposing pre-defined dimensions as in Radensky et al. (2025), we use LLM extraction to ensure flexibility based on the nature of the research idea (Rubaiat et al., 2025). Specifically, given the research idea $I$, we use $\mathcal{M}$ to extract dimensions $D = \{d_1, d_2, \ldots, d_l\}$. Examples of dimensions include tool or system design, conceptual framework, evaluation methodology, etc. Each $d_i$ also includes the reasoning for how the idea makes contributions along that dimension. These statements will provide important context to ground the query generation in the subsequent step.

**Paper Discovery.** In this step, we conduct a broad search over the literature to identify relevant papers to compare $I$ against. However, unlike the soundness module, which requires searching paper content for methodological details, contribution evaluation can be performed using only abstracts, since a paper's main contributions are typically highlighted there. This also allows us to cast a wider net and gather a broad set of related papers. Such breadth is essential for contribution evaluation, as determining truly novel contributions requires an exhaustive view of the literature. The paper discovery thus consists of the following steps: (1) For each extracted dimension, $\mathcal{M}$ generates queries to search for relevant papers using Semantic Scholar paper search and retrieve their abstracts. (2) $\mathcal{M}$ assesses similarity in contributions of each candidate paper abstract relative to the research idea $I$ and assigns a score on a scale from 1 to 5. Papers that are deemed highly relevant (i.e., score

$\geq 3$) are then used as seeds for the paper augmentation stage. (3) Paper augmentation leverages the Semantic Scholar Recommendations API to find similar papers, and we additionally extract the publications cited by each seed paper. (4) This augmented list of candidate papers then undergoes another stage of relevance assessment by $\mathcal{M}$. However, due to the typically large volume of papers at this stage, we first filter this list to the top $n$ papers based on semantic embedding[1] similarity between abstracts and $I$ before forwarding to $\mathcal{M}$.

**Pairwise Comparison.** Once the final set of papers $P_D = \{p_1, p_2, \ldots, p_m\}$ is identified, we conduct a series of pairwise comparisons to uncover how these papers' contributions compare to those proposed in $I$. Specifically, we prompt $\mathcal{M}$ to compare each paper's abstract to $I$ along each dimension $d_i \in D$. This comparison produces a granular view of the areas in which $I$ is making novel advancements and those in which it is lacking in novelty, compared to existing work.

**Contribution Review Synthesis.** The final stage of contribution evaluation consists of synthesizing dimension-level contribution assessments. For each $d_i \in D$, $\mathcal{M}$ uses the results of the pairwise comparisons to synthesize an evaluation composed of three sections: (1) *Strengths*: the novel contributions that $I$ makes along this dimension compared to prior work. (2) *Weaknesses*: the areas in which the contributions of $I$ are lacking or limited compared to prior work. (3) *Suggestions*: actionable recommendations to improve the novelty of $I$ along this dimension.

## 3 SCHOLARIDEAS: A DATASET FOR RESEARCH IDEA EVALUATION

We design SCHOLAREVAL to provide a holistic evaluation of research ideas within a domain-agnostic framework. A meaningful assessment of SCHOLAREVAL's quality thus hinges on a multi-domain collection of research ideas paired with ground-truth reviews. However, existing datasets and benchmarks fall short: they focus on full paper reviews rather than research ideas (Kang et al., 2018; Weng et al., 2025), capture singular dimensions such as novelty (Shahid et al., 2025), or are restricted to one discipline (e.g., AI; Si et al., 2025b).

To this end, we curate SCHOLARIDEAS, a dataset containing research ideas and reviews covering four disciplines: artificial intelligence, neuroscience, biochemistry, and ecology. We employ a semi-automatic pipeline to construct this dataset, where each example is validated by subject-matter experts (§3.1). Our primary motivation for developing SCHOLAREVAL is to evaluate AI-generated ideas. However, collecting detailed expert reviews for AI-generated ideas at scale is prohibitively expensive and practically infeasible. Our dataset of existing human-written ideas with expert reviews serves as a reliable proxy.

The evaluation of long-form responses, such as the ones generated by SCHOLAREVAL, is also inherently challenging. While other scientific tasks admit easily verifiable success criteria (e.g., code generation (Jansen et al., 2025; Chen et al., 2025; Li et al., 2025b)), such evaluation metrics are not readily available for our use case. We develop a multi-faceted automatic evaluation pipeline detailed in §3.2. We further corroborate our automatic evaluation with a large-scale human evaluation described in §5.

### 3.1 DATA CURATION AND ANNOTATION

**Data selection.** We manually select papers and reviews from two sources: OpenReview (ICLR 2025) for AI-related research ideas and eLife for life sciences. Both sources include high-quality reviews by multiple reviewers for all submitted papers, in contrast to many other sources that do not openly release reviews or restrict them to those of accepted manuscripts. To ensure the suitability of the reviews to be used as ground-truth, we only select papers satisfying the following criteria: (1) The paper must be reviewed by at least two reviewers. (2) All reviews of the paper are in agreement and offer a general consensus about the quality of the work. (3) The reviews mention criticism about the underlying research idea and not exclusively about obtained results or other details known post-execution. For each of the sources we use, we only retrieve the first version of the submission prior to any revisions, along with the first round of reviews. We also annotate each research idea with its

---

[1] We use Titan Text Embedding v2 (Amazon Web Services, 2025)

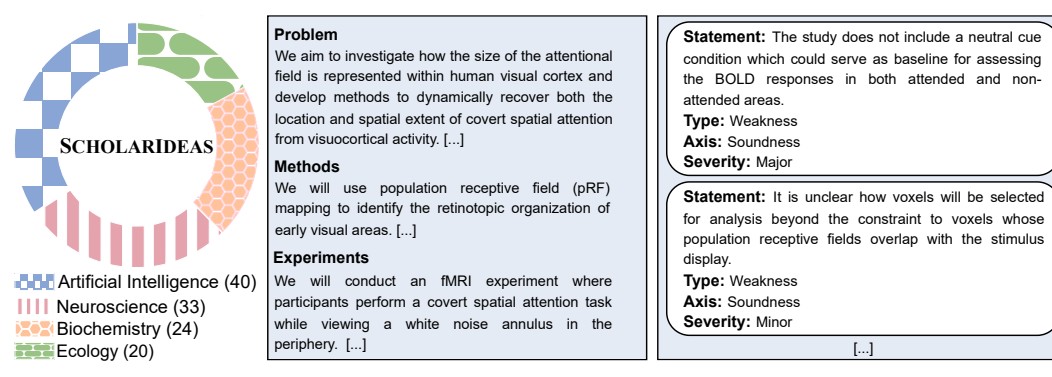

Figure 3: Overview of SCHOLARIDEAS. **Left:** SCHOLARIDEAS includes 117 research ideas across four disciplines. **Right:** Example of a neuroscience research idea and review in SCHOLARIDEAS. Each review is composed of multiple rubrics, for a total of 1076 rubrics across the dataset. This idea-review pair is adapted from (Bloem et al., 2025).

publication date, which we use as a cutoff date for literature search during the execution of baselines and SCHOLAREVAL.

**LLM-based extraction.** After identifying 130 papers across the four disciplines of interest satisfying our criteria, we follow a multi-step approach to extract the research idea and its corresponding reviews. **(1) Document parsing.** We first use GROBID to parse the paper into different sections and employ a heuristic on section names to only keep those related to the background and methodology, while dropping all sections mentioning experiment execution and results. **(2) Research idea extraction.** We instruct an LLM[2] to extract the research idea based on the parsed documents such that each research idea is composed of three components (Problem, Methods, Experiments). **(3) Review rubrics extraction.** Finally, we provide an LLM with the full reviews corresponding to the paper along with the extracted research idea and instruct it to extract statements from the reviews pertaining to assessments of the underlying research idea These statements are further classified by the LLM based on their type (strength or weakness), severity (major or minor), and axis (soundness or contribution) to construct our review rubrics.

**Expert Validation.** We invite 6 subject-matter experts (PhD students, postdocs, and professors) in artificial intelligence, neuroscience, biochemistry, and ecology to validate the quality of each (research idea, review rubrics) pair extracted by the LLM. Namely, the experts are instructed to ensure that the research idea is a faithful representation of the paper at the ideation stage, that the review rubrics do not have mentions to execution, results, or paper presentation, and that the research idea and review rubrics are consistent (i.e. the review only addresses aspects contained in the research idea). Annotators are also asked to verify the correctness of the dimension assignments made by the LLMs. Experts are instructed to make necessary corrections and to discard instances that fall out of their specific area of expertise.

At the end of this process, we obtain 117 validated (research idea, review rubrics) pairs balanced across four disciplines. Figure 3 shows the statistics of SCHOLARIDEAS and an example pair. Further details about SCHOLARIDEAS curation are given in Appendix E.

## 3.2 EVALUATION PROTOCOL

A high-quality research idea evaluation should highlight all strengths and weaknesses that an expert reviewer would mention, ground its claims in relevant literature, engage deeply with the points discussed and the works cited, and go beyond simple good or bad judgment to offer actionable suggestions for improvement. Guided by this desiderata, we develop automatic metrics to assess the performance of SCHOLAREVAL and baselines. Full details of prompts and implementation are provided in the Appendix Appendix F.

---

[2]We use Claude 4 Sonnet

Table 1: Coverage ↑ of baselines and variants of SCHOLAREVAL overall and per-discipline. * and † indicate significant improvement over 1 or all baselines, respectively. Best results are bolded. Statistical significance details are provided in Appendix H.

| | Overall | AI | Neuroscience | Biochemistry | Ecology |
|---|---|---|---|---|---|
| Llama-3.3-70B | $1.83 \pm 0.89$ | $1.88 \pm 0.92$ | $1.82 \pm 0.86$ | $1.81 \pm 0.90$ | $1.74 \pm 0.87$ |
| GPT-4.1 | $2.18 \pm 1.08$ | $2.21 \pm 1.12$ | $2.18 \pm 1.10$ | $2.14 \pm 1.04$ | $2.13 \pm 1.00$ |
| Claude-4-Sonnet | $2.18 \pm 1.04$ | $2.28 \pm 1.02$ | $2.15 \pm 1.08$ | $2.01 \pm 1.01$ | $2.11 \pm 1.03$ |
| GPT-4o-search-preview | $1.90 \pm 0.98$ | $1.95 \pm 0.97$ | $1.84 \pm 0.96$ | $1.86 \pm 1.00$ | $1.95 \pm 1.02$ |
| o4-mini-deep-research | $2.28 \pm 1.07$ | $2.35 \pm 1.07$ | $2.25 \pm 1.04$ | $2.08 \pm 1.06$ | $2.35 \pm 1.11$ |
| SCHOLAREVAL $_{\text{Llama}}$ | $2.04 \pm 1.16^{*}$ | $2.06 \pm 1.14^{*}$ | $2.05 \pm 1.18^{*}$ | $2.04 \pm 1.19$ | $1.94 \pm 1.13$ |
| SCHOLAREVAL $_{\text{GPT}}$ | $2.72 \pm 1.47^{\dagger}$ | $2.84 \pm 1.39^{\dagger}$ | $\mathbf{2.74 \pm 1.55}^{\dagger}$ | $2.61 \pm 1.42^{\dagger}$ | $2.52 \pm 1.51^{\dagger}$ |
| SCHOLAREVAL $_{\text{Claude}}$ | $\mathbf{2.77 \pm 1.40}^{\dagger}$ | $\mathbf{2.91 \pm 1.34}^{\dagger}$ | $2.55 \pm 1.45^{\dagger}$ | $\mathbf{2.64 \pm 1.40}^{\dagger}$ | $\mathbf{2.90 \pm 1.39}^{\dagger}$ |

**Coverage ↑**. This is a recall-based metric which measures the extent to which an evaluation covers the rubrics from the corresponding review in SCHOLARIDEAS. Specifically, we use Prometheus-Eval with a GPT-4 backbone (OpenAI et al., 2024), a framework shown to follow detailed evaluation rubrics effectively (Kim et al., 2024). We instruct it to assign a 1–5 score based on how well each evaluation addresses the points in the reference rubric. The final score is the average over all 1,076 rubrics in SCHOLARIDEAS.

**Reference Inv. ↓**. To compute the reference invalidity rate, we automatically check the references (i.e., paper links) cited in the evaluation to verify whether they correspond to existing papers. This is done by examining the status code returned for each link. Since edge cases arise where status codes do not clearly indicate validity, we conservatively report a lower bound: the proportion of references that are clearly invalid.

**Evidence**, **Act**, **Depth**. We use an LLM-as-judge, Claude 4 Sonnet, to give pairwise preferences between SCHOLAREVAL and our strongest baseline along three criteria: **Evidence Support**, which measures how well claims are grounded in literature and supported by relevant citations; **Action-ability**, which captures the clarity, usefulness, and feasibility of the suggestions; and **Depth**, which evaluates the level of engagement with each point and whether the evaluation mentions specifics about the work it cites rather than relying on generic statements. We also compute agreement with human annotators (see Appendix F).

## 4 EXPERIMENTS AND RESULTS

### 4.1 EXPERIMENTAL DETAILS

**Baselines.** Because there are no existing systems in the literature that evaluate research ideas with feedback at the same scope as SCHOLAREVAL, we compare against strong baselines that are most likely to be used for idea evaluation and feedback. Specifically, we select both frontier open-source and closed-source LLMs—Llama-3.3-70B (Grattafiori et al., 2024), GPT-4.1 (OpenAI, 2025a), Claude-4-Sonnet (Anthropic, 2025), and GPT-4o-search-preview (OpenAI, 2025c) for a web-connected LLM baseline—as well as a deep research system, o4-mini-deep-research (OpenAI, 2025b). We restrict our choice of deep research systems o4-mini-deep-research since it is available via API. All baselines are prompted with a detailed template aligned with SCHOLAREVAL, including method and dimension decomposition and dedicated sections for strengths, weaknesses, and suggestions.

**SCHOLAREVAL instantiation.** We evaluate three variants of SCHOLAREVAL: SCHOLAREVAL$_{\text{Llama}}$, SCHOLAREVAL$_{\text{GPT}}$, and SCHOLAREVAL$_{\text{Claude}}$, which use Llama-3.3-70B, GPT-4.1, and Claude-4-Sonnet as backbones, respectively.

Full prompts and additional experimental setup details are provided in Appendix G.

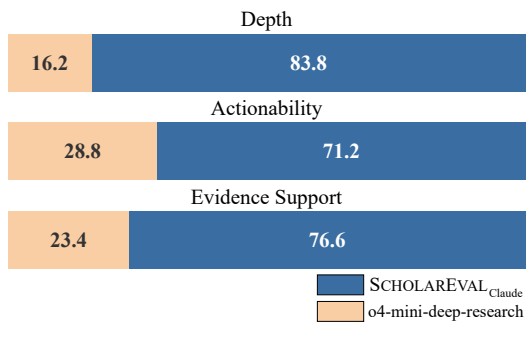

Figure 4: Win rate based on Depth , Act , and Evidence between SCHOLAREVAL Claude and o4-mini-deep-research using LLM judge.

Table 2: Rate of reference invalidity across all systems. Baseline values are lower bounds; actual invalidity is higher, especially for non-retrieval systems. Reference invalidity is not an issue in SCHOLAREVAL.

| | Reference Inv. ↓ |
|---|---|
| Llama-3.3-70B | 19.07% |
| GPT-4.1 | 15.22% |
| Claude-4-Sonnet | 13.90% |
| GPT-4o-search-preview | 1.66% |
| o4-mini-deep-research | 1.07% |
| SCHOLAREVAL Llama | 0% |
| SCHOLAREVAL GPT | 0% |
| SCHOLAREVAL Claude | 0% |

## 4.2 RESULTS AND ANALYSIS

**SCHOLAREVAL outperforms baselines in rubric coverage across disciplines.** As shown in Table 1, SCHOLAREVAL_GPT and SCHOLAREVAL_Claude achieve statistically significant gains on Coverage ↑ over all baselines overall and in every discipline. SCHOLAREVAL_Llama also surpasses its backbone (Llama-3.3-70B) in overall coverage and in AI and Neuroscience, indicating that SCHOLAREVAL delivers improvements across backbones. Importantly, these gains are not explained by emphasizing minor rubric items only; SCHOLAREVAL also provides better coverage of expert-annotated *major* points (see Table 13 in Appendix).

**SCHOLAREVAL eliminates reference invalidity.** As shown in Figure 2, only SCHOLAREVAL variants achieve 0% Reference Inv. ↓ in our evaluation. As previously stated, baseline rates represent lower bounds. However, a manual audit of a sample of outputs (see subsection H.2) indicates that up to 80% of Llama-3.3-70B citations are hallucinated in certain evaluations. Even GPT-4o-search-preview and o4-mini-deep-research are not immune, with at least 1% of cited references not existing. Our audit also uncovers subtler failures, including misattributed authors and claims that are inconsistent with the cited sources. These patterns align with findings from recent evaluations of deep research systems (Li et al., 2025a). SCHOLAREVAL eliminates these issues, ensuring that all citations resolve to valid, traceable sources.

**Quality of SCHOLAREVAL evaluations exceeds that of deep research.** We compare SCHOLAREVAL_Claude to the strongest baseline, o4-mini-deep-research based on the metrics Evidence , Act and Depth . Results in Figure 4 indicate that SCHOLAREVAL Claude is consistently better at giving evidence-based evaluations, making actionable suggestions to improve the research idea, as well as giving sufficient details and deeply engaging with the literature it cites. Our user study detailed in §5 further corroborates these results by the preference of human users.

**Ablations studies.** We conduct ablations to assess the effectiveness of individual components of SCHOLAREVAL. Namely, we remove each of the methods and results extraction (MRE), paper augmentation (PA), and pairwise comparison (PC). Details about the setup for each ablation experiment can be found in Appendix G. Results in Table 3 indicate that the removal of each of these components leads to degradation in performance of SCHOLAREVAL on Coverage ↑ . This is likely due to the reduced information given to the model (i.e. the effectiveness of the methods from prior work based on the reported results, the dense list of relevant papers, and the granular dimension-level paper comparisons). These results showcase the importance of these steps in providing SCHOLAREVAL with essential context from the literature to generate effective idea evaluations.

Table 3: Ablations of different components of SCHOLAREVAL on SCHOLARIDEAS-AI.

| | Coverage ↑ |
|---|---|
| SCHOLAREVAL Claude | 2.91 ± 1.34 |
| -MRE | 2.47 ± 1.42 |
| -PA | 2.42 ± 1.28 |
| -PC | 2.39 ± 1.23 |

## 5 EXPERT USER STUDY

**Design.** We gauge the real-world usefulness of SCHOLAREVAL by recruiting 18 experts for 46 total evaluations in a blind experiment with OpenAI's o4-mini-deep-research OpenAI (2025b). Each expert had an education level of PhD student or beyond and was verified to have at least one paper published in their field (see Table 18). We create a blind interface for experts to interact with SCHOLAREVAL (see Figure 6 and Figure 7).

**Procedure.** We ask each expert participant to first write a research idea that includes the problem they aim to tackle and the suggested methodology. This can be one they have yet to experiment with, or an idea they have already published. After receiving the soundness and contribution evaluation from a single system (either SCHOLAREVAL or OpenAI deep research), they complete a detailed rubric related to the core components of idea evaluation (e.g. the usefulness of the suggestions, the engagement with literature, etc.). The entire process for creating an idea, generating the feedback, and scoring the rubric took an average of 1 hour across participants, and we allow up to 4 unique idea submissions. The exact questions, user demographics, blindness and randomization validity, and compensation details can be found in Appendix I.

**Evaluation.** We group questions into six dimensions. Citations captures the number of references experts would actually use in their research. Because research ideas can be complex and domain-specific, we use Faithful to measure the extent to which the feedback aligns with nuances of the research idea. Useful reflects the helpfulness of each system and the expert's enthusiasm for future use. Focus measures the degree to which each system targets the most important factors of each research idea. LitEngage indicates the depth each system uses when making detailed comparisons with specific components of relevant literature. Refine gauges whether the system provides valuable, targeted, feasible suggestions for improvement that experts believe would actually improve their research idea. The questions related to each dimension are available in Table 17. We refer readers to subsection I.2 for details on linear mixed effects model, which we adopt as our statistical method.

**Results and analysis.** Results in Table 4 show a statistically significant preference for SCHOLAREVAL over o4-mini-deep-research across all six dimensions measured in our expert user study. Experts found 1.5 more useful Citations using SCHOLAREVAL and scored LitEngage 1.2 higher, demonstrating the effectiveness of our multi-stage literature retrieval pipeline. A strong effect on Faithful and Focus underscores the benefit of systemically breaking down the research plan into smaller, controlled comparisons with relevant

Table 4: Mixed effects modeling results from the expert user study. We report the standardized regression coefficient ($\beta$) and statistical significance (* $p < .05$, *** $p < .001$). Deep Research refers to o4-mini-deep-research.

| Dimension | $\beta$ | SCHOLAREVAL | Deep Research |
|---|---|---|---|
| Citations | $0.81^{***}$ | $2.66 \pm 2.43$ | $1.09 \pm 1.28$ |
| Faithful | $0.69^{*}$ | $3.88 \pm 1.50$ | $2.91 \pm 1.19$ |
| Useful | $0.63^{*}$ | $7.80 \pm 1.71$ | $6.56 \pm 2.05$ |
| Focus | $0.61^{*}$ | $7.48 \pm 1.26$ | $6.46 \pm 2.09$ |
| LitEngage | $0.61^{*}$ | $7.62 \pm 1.75$ | $6.41 \pm 1.72$ |
| Refine | $0.60^{*}$ | $7.31 \pm 1.62$ | $6.12 \pm 2.32$ |

literature. The actionable feedback generated by SCHOLAREVAL had clear advantages as well, with experts reporting a 1.2 increase in Refine . The higher score on Useful further underscores the overall usefulness of SCHOLAREVAL as research evaluation framework as judged by experts.

## 6 CONCLUSION

We introduced SCHOLAREVAL, a framework for research idea evaluation that assesses ideas based on their soundness and contribution, grounded in scholarly literature. We also presented SCHOLARIDEAS, a multidisciplinary dataset of research ideas paired with expert-annotated review rubrics. Our experiments demonstrate that SCHOLAREVAL achieves higher coverage of points raised by human reviewers and consistently delivers higher-quality evaluations compared to baselines. Moreover, our user study shows a strong preference for SCHOLAREVAL as a useful idea evaluation tool, providing a confident indication of its potential to augment the ideation process in both AI and human workflows.

## REPRODUCIBILITY STATEMENT

To ensure the reproducibility of our work, we provide full prompts and implementation details of SCHOLAREVAL (Appendix C), the detailed process of constructing SCHOLARIDEAS (Appendix E), full information about our evaluation (Appendix F), experimental setup (Appendix G), and expert user study (Appendix I). Upon acceptance, we will also openly release our code and dataset.

## ETHICS STATEMENT

SCHOLARIDEAS is collected from papers on Openreview and eLife licensed under a Creative Commons Attribution (CC BY) license, which allows reuse with attribution. We provide the full list of papers used to create SCHOLARIDEAS in Appendix E.

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

## APPENDIX

We include herein details omitted from the main text as follows:

- Limitations and Future Work
- Related Work
- SCHOLAREVAL details and prompts
- SCHOLAREVAL interface
- SCHOLARIDEAS curation details
- Evaluation metrics details

- Experimental setup details

- Evaluation results

- User study details

- SCHOLAREVAL output examples

- LLM Usage

## A  LIMITATIONS AND FUTURE WORK

We recognize the following limitations and future work directions:

**Limitations of SCHOLAREVAL.** First, as a literature-grounded framework, SCHOLAREVAL relies heavily on the retrieved literature to evaluate research ideas. Hence, it is possible that it might misjudge a method's effectiveness if it is not yet proven in the existing literature. Similarly, the contribution of a research idea might be misrepresented if some relevant papers are not retrieved for comparison. A meaningful direction for future work would be to improve the retrieval capabilities of SCHOLAREVAL to reduce such occurrences. Second, as we mention in Appendix H, depending on the choice of the model backbone, running an idea evaluation using SCHOLAREVAL can take around 12 *min* and cost up to $3. Although not too high in absolute terms, these costs can add up if SCHOLAREVAL is used to evaluate a large batch of ideas. Future work may explore more lightweight versions of SCHOLAREVAL to enhance its usability in such cases.

**Limitations of SCHOLARIDEAS and automatic evaluation.** We have made a considerable effort to select papers with high-quality reviews. However, idea evaluation remains a subjective task and human reviews cannot always be considered as ground-truth. Although the degree of coverage of human reviews can be one signal of evaluation comprehensiveness, it is not a definitive judge of overall quality. To that end, we have included other metrics in our setup to give a more holistic assessment, but the evaluation of open-form responses remains challenging, and there are other facets of evaluation that we have considered but that were challenging to scale or automate (e.g. citation factuality and relevance, usefulness, etc.). Additionally, our evaluation is limited to the four disciplines included in SCHOLARIDEAS, and although our framework is discipline-agnostic, our results might not necessarily generalize to other disciplines. Upon acceptance, we will release our entire dataset and evaluation pipeline, including the UI we used for the user study, seeking feedback from the wider scientific community on SCHOLAREVAL's utility.

**Limitations of Expert User Study.** Some measures, such as those related to Focus , are inherently subjective to the user, and could be influenced by stylistic factors rather than actual improved quality. Additionally, resarch idea quality is a source of variance, as participants were only allowed to submit unique research ideas to a given system. However, we collect a self-report of research idea detail, which is not statistically significantly different according to a Welch's t-test. Additionally, the overall scores of our results may be inflated by our expert sample, who report a 7.2/10 on AI use for research. Our results may be different in a sample who use AI less often. This does not affect our relative improvement over o4-mini-deep-research, as we show concrete evidence that the study was adequately blinded (see Appendix I). Future work should consider the barriers to adoption for automated research idea evaluation, especially for those who are less keen on using AI for their research cycle. Another strong direction would study how use of SCHOLAREVAL impacts short and long-term research success.

## B  RELATED WORK

**Literature Grounded Systems for Research.** Multiple works have been proposed that use LLMs in literature grounded systems to assist in research. A commonly targeted use case for these systems is literature synthesis and related work generation (Asai et al., 2024; Kang et al., 2023; Agarwal et al., 2025). In addition, other systems have been developed for literature understanding and question answering (Skarlinski et al., 2024; Singh et al., 2025). SCHOLAREVAL builds on techniques inspired by these works but focuses on the underexplored problem of literature-grounded research idea evaluation.

**Research Ideation.** Recently, there has been growing interest in using LLMs for generating research ideas and scientific hypotheses, often from a literature-driven perspective. For example, Wang et al. (2024) proposed SciMON, a framework that retrieves inspirations from past scientific papers to generate novel ideas. Similarly, Baek et al. (2025) employ an iterative approach over related papers and a knowledge store to generate scientific hypotheses. Other works, such as (Radensky et al., 2025; Garikaparthi et al., 2025; Pu et al., 2025), emphasize human-LLM collaboration for idea generation, while Gottweis et al. (2025) use a multi-agent approach to generate and refine hypotheses. Furthermore, Si et al. (2025b) conducted a large-scale study assessing both human- and LLM-generated ideas in the AI field, finding that LLMs are capable of generating more novel ideas based on human evaluation.

**Research Idea Evaluation.** Many of the aforementioned ideation systems also incorporate modules for idea review and refinement (Baek et al., 2025; Radensky et al., 2025). For instance, Baek et al. (2025) use a review agent that is prompted with a rubric induced from human preference. Similarly, Shahid et al. (2025); Radensky et al. (2025) introduce a retrieval-augmented framework that generates novelty classifications with brief reasoning. Additionally, Feng et al. (2025) introduced a graph-based LLM framework to score research ideas. A concurrent work by Afzal et al. (2025) proposes an LLM framework specifically for paper novelty evaluation. Wen et al. (2025) tackle the problem of idea evaluation by building a system that predicts empirical outcomes in AI research, choosing the most promising idea among a pair. SCHOLAREVAL addresses many of the limitations of these systems. First, rather than relying on the parametric knowledge of LLMs for idea evaluation as in Baek et al. (2025), SCHOLAREVAL incorporates a carefully designed retrieval pipeline to ground evaluation in the most recent literature. Second, it goes beyond one-dimensional evaluation (e.g., novelty) (Shahid et al., 2025; Afzal et al., 2025) to also assess the validity of proposed methods, since novelty alone is not a sufficient condition for successful research execution (Si et al., 2025a). Finally, SCHOLAREVAL places strong emphasis on generating dense and actionable feedback for idea refinement, in contrast to systems that only produce scores (Feng et al., 2025; Wen et al., 2025) or sparse feedback (Shahid et al., 2025).

## C    SCHOLAREVAL DETAILS AND PROMPTS

In this section we elaborate on implementation details omitted from the main text due to space limitations and present our full system prompts.

### C.1    SOUNDNESS DETAILS

#### C.1.1    ADDITIONAL IMPLEMENTATION DETAILS

**Snippet Search.** We employ the snippet search endpoint from Semantic Scholar to get paper snippets (from the title, abstract, or body) that are relevant to the method being evaluated. To retrieve all papers referenced in the snippet, we use the field `refMentions` from the returned snippet data, which allows us to match every referenced paper to its Semantic Scholar `corpusID`.

**Paper Downloading.** Once all referenced papers are identified, we use their `corpusID` to retrive their Semantic Scholar entry and use the field `OpenAccessPDF` to get the url to download the full text. Since there are cases where the paper might be behind a paywall, we use Unpaywall with our institution emails to retrieve open access version of the papers, if available.

**Methods and Results Extraction.** To extract key sections from each paper, we first use GROBID to parse the paper PDF into its XML representation. Then, we search for the methods and results section by matching the section names to an exhaustive list of section titles that could potentially be used to reference the methods and results sections (e.g. Methods, Methodology, Protocol etc.).

**TL;DR Summary.** Since the method-level soundness review can be lengthy, we also include a summarization step where we prompt an LLM to generate a TL;DR summary highlighting the top three most important strengths, weaknesses, and suggestions to address. In our SCHOLAREVAL user interface (Appendix D), both this summary and the method-level evaluation are shown to the user, with the latter being expandable.

**Citation Checking.** As a post-processing step, we call a citation checking module that uses an LLM to perform the following functionalities: ensures that all citation worthy statements are followed by relevant citations and formats the bibliography section of the evaluation report.

### C.1.2 PROMPTS

In this section we provide all the prompts used in the Soundness Module.

---

**Methods Extraction**

You are an expert research assistant. You are skilled at reading research ideas and identifying the methods that are being proposed to solve the research problem. Methods can be planned system designs, experiments, human studies, analyses, ablations, etc.

Given a research idea, you should extract all methods as a Python list, such that each method is a separate item in the list. Each item should be a word-for-word copy of a method, along with a short synthesis that grounds the method in the context of the overall research idea. The extracted methods should be interpretable on their own.

Ensure that the methods you extract address different aspects of the research idea and are non-redundant.

The method list you return should be ranked by importance to the research idea, with the most important methods first.

[start research idea] {research_idea} [end research idea]

Please output a parseable Python block as follows:

```python
plans = ["context + method", ...]
```

---

**Query Generation**

You are an expert research assistant. Given a method (i.e., one approach that researchers are adopting to execute their idea) extracted from a research idea, please construct a singular query that will be used to search for paper snippets using the Semantic Scholar API.

Use JSON format with 70 words or less per query. Do not include any text in the query besides the query itself. Do not include text like "semantic search query about ..." or "papers related to ...". Just the actual query text. No operators such as AND, OR should be used. Just a query in natural language that is relevant to the method.

[start extracted method] {clean_method} [end extracted method]

In case the method does not have enough context to construct an effective snippet search query, you can use the research idea to understand the overall research direction and inject useful context.

[start research idea] {research_idea} [end research idea]

Please output a parseable JSON block as follows, being especially careful to use the correct number of escape characters:

```json
{
    "query": "Your search query here (IN 70 WORDS OR LESS)"
}
```

---

## Method and Result Synthesis

You are an expert research assistant knowledgeable in many domains. You are extremely critical and observant, and do not overgeneralize findings. You are given a proposed research method and the methods/results section from a paper.

[start paper] {paper_text} [end paper]

[start proposed research method] {proposed_method} [end proposed research method]

To further understand the scope of the proposed research method, here is the entire research idea that it is extracted from — a method is a single approach that researchers are adopting to execute their research idea:

[start research idea] {research_idea} [end research idea]

For any method in the paper that is related to the proposed research method and the overall research idea, please summarize the method used in the paper, report the experimental outcome from using the method, and provide some context for experimental conditions.

Do not use any in-text citations. Ensure that the method, results, and context you provide are specific and detailed, and that they mention how it relates to the proposed method and research idea.

If the proposed research method does not relate to any methods in the paper, please return an empty dictionary.

Strictly follow the output format displayed below.

JSON formatting requirements: - Must be a complete, valid JSON object - Start with an open bracket and end with closed bracket - No trailing commas after the last property - Validate JSON structure before output

```json
{
    "method": "Description of experimental approach including:
    algorithm/technique, datasets/inputs, computational
    resources, implementation/experimentation details, and
    evaluation setup, and metrics/instruments used, etc.",
    "results": "Quantitative outcomes with specific values,
    comparisons to baselines, statistical significance where
    applicable",
    "context": "Key experimental conditions:
    dataset/population size, hardware/system/instrument
    specs, hyperparameters, or other domain-specific
    constraints that affect reproducibility"
}
```

## Method-Level Soundness Review Synthesis

You are an expert research assistant knowledgeable in many domains. You are extremely critical and observant, and do not overgeneralize findings.

You are given a proposed research method and a list of related work.

Your objective is to create a meta-review of the related work in the context of the proposed research method. Point out any evidence that supports or contradicts the proposed method. Make sure to contrast the related work as a series of iterative scientific work, where newer work can provide evidence that supports or contradicts older work.

It is important that the meta-review you generate always ties back to the original research idea. Judge the support and contradictions as well as suggested actions for each method within the general context of the research idea to ensure that your review is highly relevant and precise.

Be granular, making sure to reference specific details such as:

- algorithm/technique, datasets/inputs/population, computational resources, statistical methods, implementation details, and evaluation setup, and metrics/instruments used, etc.
- quantitative outcomes, comparisons to baselines, statistical significance
- dataset/population size, hardware/system specs, hyperparameters, or other domain-specific constraints that affect reproducibility

It is important that for each method-level meta-review that you generate, your review of the support and contradictions should be ordered starting from strongest evidence of support/contradiction to the weakest. Likewise, the suggested actions should be ordered from most important to least important. This does not mean that you will generate these as bullet points, but rather detailed, coherent paragraphs that are logically ordered.

It is required to copy the in-text citations with their links in Markdown format `[(author, YYYY-MM)](link)` when referring to related work.

**Related Work:** {`related_work`}

[start proposed research method] {`pm`} [end proposed research method]

[start research idea] {`rp`} [end research idea]

**Output format:** Please output a parseable JSON block as follows:

```json
{
    "support": "evidence that supports the proposed method",
    "contradictions": "evidence that contradicts the proposed
                       method",
    "suggested_action": "how can the proposed method be improved
                         based on the related work",
    "soundness_score": "int score 0 to 10 based on the evidence
                        for and against the proposed method"
}
```

## C.2 CONTRIBUTION DETAILS

### C.2.1 ADDITIONAL IMPLEMENTATION DETAILS

**Paper Search.** To retrieve relevant papers for contribution analysis, we use the Semantic Scholar paper relevance search which returns the top n most relevant papers based on a query. For subsequent processing in the contribution module (i.e. relevance assessment and pairwise comparison) we retrieve the paper abstract from the returned `abstract` field.

**Paper Downsampling.** Before the pairwise comparison step, we sample 25 papers from the final list. This downsampling serves two functions: first it reduces the latency and computational cost of the pairwise comparison step, and it reduces the overall context forwarded to the final synthesis step to avoid prohibitive context lengths.

**Citation Checking.** Similar to the Soundness module, we also apply a final post-check on the citations to ensure proper attribution and bibliography formatting.

### C.2.2 PROMPTS

In this section we provide all the prompts used on the Contribution Module.

**Dimension Extraction Prompt**

You are a helpful assistant that reads scientific research ideas and extracts a structured summary of their contributions.

Your task is to identify a small number of high-level contribution dimensions, and for each dimension, extract one or more specific contribution statements that are faithful to the research idea.

Contribution dimensions should represent general categories of scientific contribution that are meaningful and comparable across research ideas, regardless of the field. These might include: - methodology (e.g., proposing a new method, model, or procedure) - application (e.g., applying existing methods to a new problem or domain) - theoretical contribution (e.g., proving a new result, deriving a new model) - data (e.g., constructing a new dataset, conducting original measurements or surveys) - evaluation (e.g., designing an experimental protocol, benchmarking a technique) - tool or system design (e.g., building software, devices, or infrastructure to support research) - conceptual framework (e.g., introducing a new taxonomy or way of thinking about a problem)

Do not limit your output to the examples above but rather generate suitable dimensions for the research idea given to you. Only include dimensions that are actually reflected in the research idea — do not add generic or speculative categories.

For each dimension, write one or more contribution statements that clearly explain what the research is proposing. These statements should be precise, self-contained, and informative — make sure they include enough context as they will be used as the basis to generate search queries later on.

Each dimension may include multiple contribution statements, but do not repeat the same idea across dimensions. Avoid redundancy, and keep the summary compact and informative.

Please output a parseable JSON block as follows:

```json
{
  "<dimension_name_1>": [
    "<contribution_statement_1>",
    "<contribution_statement_2>"
  ],
  "<dimension_name_2>": [
    "<contribution_statement_3>"
  ]
}
```

## Query Generation (Contribution)

You are an expert at writing highly targeted search queries for retrieving academic papers using the Semantic Scholar API.

You will be given a full research idea and one specific contribution from that idea.

Your task is to generate up to {n_queries} short, diverse, and high-quality search queries that are focused on the given contribution and consistent with the overall research context.

These queries will be used to search for paper abstracts. They must be semantically rich and optimized to retrieve papers that are directly relevant to the contribution.

Avoid generating queries that are too general, overly broad, or likely to return unrelated results. Also avoid generating queries that are just broadly related to the research idea but not specifically tailored to the contribution.

**Guidelines:**

- Each query should be brief and focused (as if typed into a search bar; as a rule of thumb do not exceed 7 words per query).
- Queries must stay tightly aligned with the core idea of the contribution.
- Incorporate key methods, problems, domains, or goals described in the contribution.

- Reflect an understanding of the broader research idea, but do not drift away from the specific contribution.
- Use natural phrasing (no Boolean operators like AND, OR, etc.).
- Ensure queries are meaningfully different from one another while remaining on-topic.

### Relevance Assessment Prompt

You are an expert at evaluating whether a paper should be considered relevant for assessing the scientific contribution of a research idea.

You will be given a research idea and a paper abstract retrieved from Semantic Scholar. Your task is to thoroughly understand the scientific contributions of each of the research idea and the paper, and to output a score from 0 to 5 indicating how similar the contributions of the paper are to those of the research idea.

**Scoring Rubric:**
**Score 5 — Highly Similar Contributions:**

- The paper addresses the exact same research question or hypothesis as the idea
- Uses identical or very similar methodological approaches
- Targets the same specific population, system, or domain
- Would directly compete with or overlap significantly with the proposed research
- The paper's findings would substantially impact the novelty of the proposed work

**Score 4 — Very Similar Contributions:**

- The paper addresses a closely related research question with significant overlap
- Uses similar methodological approaches with minor variations
- Targets a very similar population, system, or domain
- Shares most key variables, measurements, or outcomes of interest
- The paper's contributions would moderately impact the proposed research's novelty

**Score 3 — Moderately Similar Contributions:**

- The paper addresses a related research question within the same broad area
- Uses some similar methods or approaches but with notable differences
- Targets a related but distinct population, system, or domain
- Shares some key concepts, variables, or theoretical frameworks
- The paper provides useful context but doesn't directly threaten novelty

**Score 2 — Somewhat Similar Contributions:**

- The paper is in the same general field or discipline
- Uses different methods but addresses conceptually related problems
- Limited overlap in specific research focus or target populations
- Shares broad theoretical background but differs in specific contributions
- The paper is peripherally relevant for background or context

**Score 1 — Minimally Similar Contributions:**

- The paper is tangentially related to the research area
- Very limited overlap in methods, populations, or specific research questions
- May share some terminology or broad field classification
- Provides minimal insight relevant to the proposed research
- Connection is primarily at the disciplinary level

**Score 0 — No Similar Contributions:**

- The paper addresses completely different research questions

- No meaningful overlap in methods, populations, or domains
- Different field or discipline entirely
- No relevant insights for the proposed research
- No discernible connection between the contributions

Respond with a JSON object containing two fields `rationale` and `score`. Please output a parseable JSON block as follows:

```json
{
  "rationale": "<your detailed reasoning for the score>",
  "score": <an integer from 0 to 5 based on the rubric>
}
```

---

**Pairwise Comparison Prompt**

You are an expert in assessing the novelty of scientific research ideas relative to existing work. Your job is to compare a full research idea to a paper abstract based on novelty and originality. You will be given a research idea, a paper abstract, and a comma-separated list of contribution dimensions.

Produce a structured output consisting of:

1. `overall_comparison`: a broad yet precise summary of the novelty of the research idea versus the paper, i.e., whether the idea proposes ideas/angles/uses that appear original relative to what the abstract claims; identify overlap vs. originality explicitly.

2. `dimension_comparisons`: for each provided dimension, a novelty comparison that states whether the idea is doing something not present in the paper for that dimension (or vice versa), and a numeric score:

   - 1 = The research idea is more novel under this dimension (adds ideas/angles/uses not present in the paper abstract)
   - 0 = Neither appears more novel or the paper does not address this dimension (tie/insufficient evidence)
   - -1 = The paper appears more novel or the idea largely replicates what the paper already presents under this dimension

Return your output as a parseable JSON block with exactly this structure:

```json
{
  "overall_comparison": "<novelty-focused summary>",
  "dimension_comparisons": {
    "<dimension_1>": {
      "comparison": "<comparison for this dimension>",
      "score": <1 | 0 | -1>
    },
    "<dimension_2>": {
      "comparison": "<comparison>",
      "score": <1 | 0 | -1>
    }
  }
}
```

---

**Dimension-level Contribution Synthesis Prompt**

You are reviewing a research idea for novelty and originality of its contributions and their impact.

You are critical, knowledgeable, precise, and not afraid to give truthful assessments of the idea's novelty. Do not discuss methodology soundness, feasibility, correctness, or experimental quality — those are out of scope here.

You will be given a full research idea and a JSON file of pairwise comparisons between the idea and existing papers. Each comparison includes a paper reference (authors, title, year, URL), an `overall_comparison` (novelty-focused), and `dimension_comparisons` comparing the research idea to a related work based on a specific contribution dimension.

Your task is to synthesize an overall evaluation of the idea's novelty and impact based on all pairwise comparisons.

**Structure:**
Produce a novelty and impact assessment for each contribution dimension found in the comparisons. Each paragraph focuses on one dimension and contains three subsections:

- **Strengths:** Precisely explain where the idea is novel under this dimension versus the papers and makes an impact; back claims with in-text citations.
- **Weaknesses:** Precisely explain where the idea lacks novelty or is already covered by prior work, and hence lacks impact; back claims with in-text citations. Be exhaustive with identifying weaknesses.
- **Suggestions:** Give actionable, feasible, and useful suggestions to improve the novelty and impact of the work, if needed, based on the evidence from the Strengths and Weaknesses sections; back claims with in-text citations.

Use in-text citations with links in Markdown format: `[Author et al., Year](URL).`

---

## D    SCHOLAREVAL INTERFACE

To conduct the user study and for wider feedback from the community (upon acceptance), we create a SCHOLAREVAL user interface. As shown in Figure 5, our interface includes an input box for the user to enter their research idea or upload it from a file. Optionally, the user can specify a literature search cutoff date, This is especially useful if SCHOLAREVAL is used to evaluate an already published research idea. The user can then generate a Soundness review or switch tabs to generate a Contribution review.

## E    SCHOLARIDEAS CURATION DETAILS

### E.1    PAPER LIST

We provide the full list of the 117 papers that were used to construct SCHOLARIDEAS, at the end of the manuscript in Tables 19, 20, 21, and 22. Each entry in the table links to the paper on OpenReview or eLife.

### E.2    LLM-BASED EXTRACTION

### E.2.1    DETAILS

For each paper in our dataset, we use Claude 4 Sonnet to extract the research idea and review rubrics. The extraction of the review rubrics is done in two stages: first, we instruct the LLM to extract the verbatim excerpts from the review that give assessments about the research idea. In the second step, we instruct the LLM to remove redundancies and format the extracted text into standalone statements classified based on type, axis, and severity. The full prompts used in this process are provided in section E.2.2.

# 🔍 ScholarEval: Research Idea Evaluator

ScholarEval is literature-grounded research idea evaluator. Upload your research idea and get in-depth feedback based on Soundness and Contribution analysis.

## Input Research Idea

Paste your research idea here:

Enter your research idea including problem statement, methodology, and experimental design...

**OR**

Upload research plan file ⑦

☁ Drag and drop file here
Limit 200MB per file • TXT, MD, PDF      **Browse files**

**Optional: Literature Search Cutoff Date**

Cutoff date (YYYY-MM-DD) ⑦

YYYY-MM-DD

## Evaluation Results

Soundness Review    Contribution Review

### Soundness Review

Generate Soundness Review

Figure 5: SCHOLAREVAL user interface.

## E.2.2  PROMPTS

---

**Research Idea Extraction**

You are a highly skilled assistant tasked with thoroughly reading the content of a research paper and extracting its research idea. The research idea should reflect the state of the research at the time it was proposed, before any experiments or conclusions were drawn.

A research idea consists of the following key sections:
**Problem:**
  • This section provides the background of the problem being addressed, the motivation for pursuing it, and any hypotheses that the authors have formed at the start of their research.
  • Focus on the initial framing of the problem, why it's important, and the research questions that guide the study.
**Method:**
  • Summarize the methodology that the authors will follow to address the problem. This includes the overall approach, any theoretical frameworks, models, or algorithms to be used, and the rationale behind the chosen methodology.
  • Do not include any references to specific results or findings — only the proposed methods and strategies.
**Experiment Design:**
  • Describe the experiments that the authors will conduct to test their hypotheses. This should include the design, variables, procedures, and any tools or techniques that will be employed.
  • Provide details about how the authors plan to gather data, measure outcomes, and assess the effectiveness of the methods, but do not include any results.

---

- However, make sure not to make any reference to experimental setups that the authors could not have known before conducting the experiments (e.g., specific values for hyperparameters or other design choices that were made through trial and error).

**Key Constraints:**

- **No results or conclusions:** Exclude any findings, outcomes, or conclusions that were reached after conducting the experiments. The research idea should represent the research at the proposal stage, not the completion stage.
- **First-person perspective:** Write the research idea from the authors' point of view, as if the authors themselves are outlining their proposed work. Avoid referring to the authors in the third person.
- **Concise and focused:** The research idea should be clear, concise, and direct. Include only the information necessary to describe the research as it was proposed, not the outcomes.
- **Represent the status before experimentation:** Ensure the idea reflects the status of the project when it was still in the idea or proposal phase, prior to conducting any experiments.

**General Writing Guidelines:**

- Use active voice as if it's written by the authors themselves.
- Maintain a formal yet straightforward tone typical of research ideas.
- Keep the text brief and to the point, and avoid extraneous details or unnecessary elaboration.

The research idea should be faithful to the original research paper, concise, and aligned with the intent of the authors at the time of the project ideation phase.

Generate the research idea for the following paper:

`{paper_content}`

---

**Verbatim Review Extraction**

You are an expert assistant skilled at extracting key components of scientific paper reviews containing evaluations of the underlying research idea and discarding all other criticism of other aspects of the scientific paper, including results and presentation.
Namely, you will be given a text containing all the reviews that the scientific paper received as well as a research idea detailing the main research idea contained in the scientific paper, and your job is to extract sentences from the review that address components mentioned in the research idea. Any sentences that contain criticism of aspects of the paper not mentioned in the research idea should not be extracted. This is an important guideline to follow.

**Reviews contain the following components:**

- **Summary:** A brief overview of the paper's contributions, main ideas, and objectives.
- **Strengths:** Positive aspects highlighted by the reviewer, such as innovative approaches, solid methodology, or valuable contributions.
- **Weaknesses:** Negative aspects or concerns raised by the reviewer, such as lack of clarity, methodological flaws, or unanswered questions.
- **Questions:** Queries or points of clarification raised by the reviewer, often regarding experimental design, assumptions, or aspects of the research idea.

**The research idea corresponding to the research paper includes:**

- **Problem:** Background information on the problem the paper addresses, the motivation for pursuing it, and any hypotheses formed by the authors.
- **Method:** The methodology proposed by the authors to address the problem.
- **Experiment Design:** Details on the experiments, data collection, and evaluation techniques that the authors plan to employ.

**Review:**

`{review_text}`

**Research Idea:**

`{research_idea_text}`

**Output format:** This is an extraction task, not a generation task. Your job is exactly to extract sentences from the reviews which address the ideation behind the research (i.e., the research at the idea stage, imagine that no results have been obtained or paper written yet).

These should only include criticism related to the problem that the scientists are trying to solve, the methodology they are employing, or the experiments they plan to conduct. Any other content related to typographical errors, presentation or formatting issues, results, comments on figures and tables, artifacts, or conclusions derived from executing the experiments should absolutely not be included.

Only points related to the underlying research idea as specified in the research idea should be kept. Any point not related to an aspect explicitly mentioned in the research idea should not be included.

Return the extracted sentences (**verbatim! no changes made**) as follows:

`Extracted Review: <verbatim extracted sentences>`

Nothing else besides the extracted review should be contained in the response.

---

### Review Formatting Into Rubrics

You are an expert assistant skilled at extracting clear and non-redundant evaluation statements from a review summary. In this task you will be given a summary that was created by merely concatenating extracted sentences from a longer review. You will be given this extractive review summary along with the research idea that it reviews and you have to do the following tasks while taking into account the constraints.

**Tasks:**

- **Removing redundancies:** The review summary was created by concatenating the extracted sentences from multiple reviews written by different reviewers about the same paper. Thus, some of the sentences can be redundant. You should remove all redundancies. If a point is mentioned more than once, drop the other mentions and only keep one.

- **Express each evaluation** (weakness or strength mentioned by the reviewer) as a clear statement. Hence your final output should be a list of such statements, each one addressing a specific point made by the reviewer(s).

- **Adding context:** The review summary might refer to some aspects of the research idea without sufficient context. You should aim to make each statement in the list clearer by adding context from the research idea that would enhance the understanding of the review.

- **Remove mentions of the final paper:** These statements are extracted from the reviews of a paper submitted to a conference or journal. Hence, some statements might have mentions of the final paper (e.g., "in line so and so ..." or "in Figure x ..."). When such statements are encountered, you should slightly rephrase them — without alterations to the meaning — to ensure that they do not refer to the paper but rather to the corresponding component in the research idea. Ensuring this consistency between the statements and the research idea content is crucial.

- For each statement (item in the list), you should also annotate it with whether it is a `"strength"` or a `"weakness"`, whether it is related to the paper's `"soundness"` (correctness of the methods, etc.) or `"contribution"` (novelty and contribution to the field), and finally annotate whether it represents a `"major"` or `"minor"` point. Major points are those that reviewers highly stress and that affect the overall evaluation of the paper, while minor points are those that are mentioned but not stressed as much.

**Constraints:**

- Make sure that the changes you make are minimal and only aim to remove redundancy, express the statements as a list, add context, and ensure consistency with the research idea content.

- Do not completely paraphrase the review summary. Try to keep the original wording and sentences as much as possible. Your changes should not lead to a significant overhaul of the review summary.

- Do not change the tone of the review. It should still be written as if produced by an actual reviewer.

**Extractive review summary:**

{intermediate_text}

**Research Idea:**

{research_idea_text}

Please output a parseable JSONL (JSON Lines) block where each item in the JSONL is a JSON object on a single line as follows:

```jsonl
{
  "statement": "<one clear, self-contained statement>",
  "type": "strength",
  "axis": "soundness",
  "severity": "major"
}
{
  "statement": "<another clear, self-contained statement>",
  "type": "weakness",
  "axis": "contribution",
  "severity": "minor"
}
{
  "statement": "<another clear, self-contained statement>",
  "type": "strength",
  "axis": "soundness",
  "severity": "minor"
}
```

### E.3 EXPERT VALIDATION

#### E.3.1 DETAILS

We invited six experts of Ph.D. students, postdoctoral researchers, and professors (1 from computer science, 2 from biochemistry, 2 from neuroscience, and 1 from ecology). We first conducted a training session to explain the validation task and we have provided the annotators with detailed instructions, shown in section E.3.2. We made an effort to match every annotator to research ideas in their specific area of specialty, and annotators were instructed to skip a research idea if they felt that it fell out of their area of expertise.

#### E.3.2 VALIDATION INSTRUCTIONS

SCHOLARIDEAS Annotation Guidelines

**Get familiar with the data**
Each task contains a research idea (.txt file) and a review summary (.jsonl file).
The research idea is composed of three main sections:

- **Problem:** the problem that the research idea tackles

- **Methodology:** the adopted methodology

- **Experiments:** the planned experimental setup

On the first line of the research idea, you will find a link to the original publication and public reviews on eLife. This is the link to the first version of the publication (v1), which contains the publication before any changes or revisions. On the eLife website, the publication is available under the *Full text* tab and the full reviews under the *Peer Review* tab.

The review summary is a JSONL file. Each line in the file is a dictionary structure with the following attributes:

- **Statement:** a statement extracted from the original reviews

- **Type:** whether the statement represents a 'weakness' or 'strength'

- **Axis:** whether the statement is related to the 'soundness' of the work (correctness and validity of the methods) or its 'contribution' (novelty and impact)

- **Severity:** whether this is a 'major' point (emphasized by the reviewer) or a 'minor' one

**Annotate a task**

For each task:

1. **Check the research idea:** Ensure it represents the state of the work at the ideation stage, i.e. before any experiments were conducted or results obtained. It should only contain the problem, methodology, and planned experiments that the authors could have known at the ideation stage. Verify that the content is faithful to the original publication without significant alterations.

2. **Check the review summary:** The statements should only pertain to the underlying research idea. Points about results, details only known after experimentation, figures, tables, or presentation issues should not be included. Verify that:
   - The statements pertain only to the research idea (problem, methodology, planned experiments).
   - All such statements are extracted.
   - Each statement type is correctly categorized as 'strength' or 'weakness'.
   - Each statement axis is correctly categorized as 'soundness' or 'contribution'.
   - Each statement severity is correctly categorized as 'major' or 'minor'.

3. **Make necessary changes:** If the above guidelines are not respected, make modifications (e.g., editing the research idea, adding/removing statements in the review summary, or correcting type/axis/severity labels).

### E.3.3 VALIDATION RESULTS

Table 5 shows the distribution of edits for each of the research idea and review rubrics. Common issues that required edits include the mention of results in the research idea, errors in the classification of statements based on their severity, and the inclusion of rubrics mentioning the results obtained in the paper or the referring to the presentation (tables, figures, etc.).

Table 5: Distribution of edits across research ideas and review rubrics.

| Category | Percentage of Edits |
| --- | --- |
| Research Idea | 13.40% |
| Review Rubrics | 79.10% |

## F EVALUATION METRICS DETAILS

### F.1 COVERAGE METRIC

To compute the coverage metric, we use Prometheus-Eval (Kim et al., 2024) framework with GPT-4 (OpenAI et al., 2024) backbone. We provide Prometheus with the detailed rubric given below to give

a score on the degree of coverage of each rubric of SCHOLARIDEAS. These scores are subsequently averaged to compute the final Coverage score.

---

**Coverage rubric given to Prometheus**

```
rubric_data = {
  "criteria":
  "You are given a single essential statement from
  a research idea review targeting the technical
  soundness and scientific contribution of a research
  idea. This statement was extracted from actual human
  reviews about the research idea and will be used
  as your ground-truth reference. Additionally, you
  will be given a research idea review that aims to
  give a detailed evaluation of the same research idea.
  The review given to you will typically be long and
  composed of multiple sections and references to related
  papers. Your job is to verify that the given review
  has a mention of the reference statement in its content.
  These references can be paraphrased or articulated
  differently, but they should express the same meaning
  as the reference statement. Does the given review
  express the reference statement?",

  "score1_description":
  "The given review does not mention the reference
  statement at all",

  "score2_description":
  "The given review mentions some related concepts
  but does not express the core meaning of the
  reference statement",

  "score3_description":
  "The given review partially expresses the reference
  statement but misses key aspects or nuances",

  "score4_description":
  "The given review expresses most aspects of the
  reference statement with minor omissions or slight
  differences in emphasis",

  "score5_description":
  "The given review clearly expresses the reference
  statement, capturing its core meaning and implications"
}
```

---

F.2   REFERENCE INVALIDITY

We compute reference invalidity as the fraction of non-resolving citations by issuing an HTTP HEAD request for every paper link referenced in the output of SCHOLAREVAL and baselines, and inspecting the returned status code. In our initial implementation, we noticed that this approach undercounts failures: many publishers return 403 (bot blocking) or even 200 (soft-404 pages etc.) for URLs that do not actually resolve to the referenced papers upon manual inspection. We thus resort to reporting a lower bound for reference invalidity; we label a link as invalid only when it returns 404, 410, or any 5xx status. However the true of reference invalidity for baseline systems is higher. We provide failure cases observed in our manual audit in §H.2. Reference invalidity is entirely eliminated in SCHOLAREVAL.

## F.3 LLM METRICS

We use Claude 4 Sonnet as our LLM judge and instruct it to choose the winner between a pair of reports (SCHOLAREVAL and o4-mini-deep-research) using the prompt below.

To validate the reliability of the LLM judgments, we conduct a small scale study with 33 report pairs and ask 6 Ph.D. students in artificial intelligence, neuroscience, and chemistry - who are unfamiliar with our system and would thus not be able to identify it in the pair - to choose the better report based on the three criteria. We provide our annotators with the same instructions given to the LLM judge. Each report pair was rated by 1, 2, or 3 annotators, and we set the final human label for each report pair based on majority vote. Reports that had equally split ratings were discarded, resulting in 18 remaining pairs. We compute the inter-annotator agreement between the human and LLM labels based on percent agreement and Gwet's AC1, which is a suitable inter-annotator agreement metric for cases representing class imbalance. The results shown in Table 6 underscore a high percent agreement for all metrics and a moderate to substantial agreement based on Gwet's AC1.

Additionally, we provide the human preference results based on the full sample of 33 report pairs in Table 7. These preference results showcase that SCHOLAREVAL is substantially preferred over o4-mini-deep-research across all three metrics and corroborate the LLM-judge preference results.

> **LLM judge prompt**
>
> You are an expert scientific reviewer. You will be given two research evaluation reports (Report A and Report B) and you must answer which one is better based on the following criteria:
>
> **Evidence support:** An evidence-supported report is one that grounds its claims in the literature and backs them with relevant citations that improve traceability and its overall trustworthiness and reliability. Which report has more evidence support?
>
> **Actionability:** Actionability is the extent to which the report offers varied, clear, and actionable suggestions that are likely to improve the research idea. Which report is more actionable?
>
> **Depth:** Depth is measured by the degree of engagement with the point being evaluated and the literature cited. A deep report discusses each point from multiple angles and references specific details about the literature it cites, rather than relying on generic statements followed by citations. Which response has greater depth?
>
> You must choose either `A`, `B`, or `Tie` (when both responses are equivalent). Read both responses thoroughly before judging, and give thorough rationales that show that you are a fair and unbiased judge. If you choose that one response is better, clearly rationalize why. If they are equivalent, then you should choose `Tie`.
>
> **Report A:** {`prompt_report_a`} **Report B:** {`prompt_report_b`}
>
> Judge the responses based on the criteria, while respecting the output format below:
>
> ````json
> {
>   "Evidence-support-rationale": "...",
>   "Evidence-support-winner": "<A, B, or Tie>",
>   "Actionability-rationale": "...",
>   "Actionability-winner": "<A, B, or Tie>",
>   "Depth-rationale": "...",
>   "Depth-winner": "<A, B, or Tie>"
> }
> ````

Table 6: Inter-annotator agreement (human vs. LLM) across metrics.

| Metric | Percent Agreement (%) | Gwet's AC1 |
|---|---|---|
| Evidence support | 66.67 | 0.4476 |
| Actionability | 72.22 | 0.1667 |
| Depth | 61.11 | 0.3883 |

Table 7: Human preference results on sample of 34 report pairs.

| Metric | SCHOLAREVAL | o4-mini-deep-research | Tie |
|---|---|---|---|
| Evidence Support | 70.6 | 14.7 | 14.7 |
| Depth | 79.4 | 11.8 | 8.8 |
| Actionability | 82.4 | 11.8 | 5.9 |

# G EXPERIMENTAL SETUP DETAILS

## G.1 BASELINES

In this section we provide the detailed prompts used to generate the Soundness and Contribution evaluation for all baseline systems. To ensure a fair comparison, we use optimized prompts that instruct all baseline systems to generate detailed results in the same format as SCHOLAREVAL.

Similar to Li et al. (2025a), to avoid cases where the retrieval-augmented models (i.e. GPT-4o-search-preview and o4-mini-deep-research) retrieve the paper corresponding to the research idea being evaluated, we include an instruction to limit the retrieved literature to publications released before the cutoff date (i.e. the publication date of the paper that the research idea was extracted from). Although not error proof, in our experimentation we have observed that this greatly reduces the instances where the target paper is retrieved.

---

**Baselines Soundness Evaluation Prompt**

You are an expert research assistant. You are skilled at evaluating the soundness of all methods proposed in a research idea.

Create an evaluation for every method being proposed to solve the research problem. If there are `n` methods there should be `n` evaluations. Methods can be planned system designs, experiments, human studies, analyses, ablations, etc.

You should find related work for each method being proposed, such that it can provide evidence that supports or contradicts the method at hand.

It is important that the evaluation you generate always ties back to the original research idea. Judge the support and contradictions as well as suggested actions for each method within the general context of the research idea.

Be granular, making sure to reference specific details such as:

- algorithm/technique, datasets/inputs, computational resources, implementation details, and evaluation setup, and metrics used
- quantitative outcomes, comparisons to baselines/state-of-the-art, statistical significance
- dataset size, hardware specs, hyperparameters, or other domain-specific constraints that affect reproducibility

It is required to put the in-text citations with their links in Markdown format `[(author, YYYY-MM)](link)` when referring to related work.

Here is the research idea to evaluate:
[start research idea] {`research_idea`} [end research idea]

**The output should be formatted as follows, such that each extracted method from the research idea would be its own section title, in which there are three subsections: strengths, weaknesses, and suggestions.**

```
<method_1>
"support": "evidence that supports the proposed method",
"contradictions": "evidence that contradicts the proposed
method",
"suggestions": "how can the proposed method be improved based on
```

---

```
the related work"

<method_2>
"support": "evidence that supports the proposed method",
"contradictions": "evidence that contradicts the proposed
method",
"suggestions": "how can the proposed method be improved based on
the related work"

<method_n>
"support": "evidence that supports the proposed method",
"contradictions": "evidence that contradicts the proposed
method",
"suggestions": "how can the proposed method be improved based
on the related work"
```

- When searching for related work, restrict your data sources to research papers. Do not use webpages or any other sources.
- All papers that you reference must be published on or before `cutoff_date`. This is absolutely necessary. Double check every publication's date before referencing it.
- The output format needs to be respected strictly. Do not change the format in any way.
- It is required to copy the in-text citations with their links in Markdown format `[(author, YYYY-MM)](link)` when referring to related work.

**Baselines Contribution Evaluation Prompt**

You are an expert research assistant. You are skilled at evaluating the novelty and impact of a research idea.

First, your task is to identify a small number of high-level contribution dimensions. Contribution dimensions should represent general categories of scientific contribution that are meaningful and comparable across research ideas, regardless of the field. These might include:

- **methodology** (e.g., proposing a new method, model, or procedure)
- **application** (e.g., applying existing methods to a new problem or domain)
- **theoretical contribution** (e.g., proving a new result, deriving a new model)
- **data** (e.g., constructing a new dataset, conducting original measurements or surveys)
- **evaluation** (e.g., designing an experimental protocol, benchmarking a technique)
- **tool or system design** (e.g., building software, devices, or infrastructure to support research)
- **conceptual framework** (e.g., introducing a new taxonomy or way of thinking about a problem)

Do not limit your output to the examples above but rather generate suitable dimensions for the research idea given to you. Only include dimensions that are actually reflected in the research idea — do not add generic or speculative categories. Please do not generate redundant dimensions.

After having identified the dimensions, search through related literature for similar work and compare the novelty and impact of the research idea to these works along each dimension. Please summarize the results of your research into separate sections with biggest strengths, weaknesses, and suggestions for improvement according to the novelty and impact of the extracted dimensions compared to related work.

It is required to put the in-text citations with their links in Markdown format `[(author, YYYY-MM)](link)` when referring to related work.

Here is the research idea to evaluate:
[start research idea] {`research_idea`} [end research idea]

```
<dimension_name_1>
"strengths": "direct comparison with literature where the
research idea is MORE novel and/or impactful under this
dimension",
"weaknesses": "direct comparison with literature where the
research idea is LESS novel and/or impactful under this
dimension",
"suggestions": "actionable and useful suggestions to improve
the novelty and impact of the work, if needed, based on the
evidence from the strengths and weaknesses sections"

<dimension_name_2>
"strengths": "direct comparison with literature where the
research idea is MORE novel and/or impactful under this
dimension",
"weaknesses": "direct comparison with literature where the
research idea is LESS novel and/or impactful under this
dimension",
"suggestions": "actionable and useful suggestions to improve
the novelty and impact of the work, if needed, based on the
evidence from the strengths and weaknesses sections"

<dimension_name_n>
"strengths": "direct comparison with literature where the
research idea is MORE novel and/or impactful under this
dimension",
"weaknesses": "direct comparison with literature where the
research idea is LESS novel and/or impactful under this
dimension",
"suggestions": "actionable and useful suggestions to improve
the novelty and impact of the work, if needed, based on the
evidence from the strengths and weaknesses sections"
```

**Guidelines:**

- When searching for related work, restrict your data sources to research papers. Do not use webpages or any other sources.
- All papers that you reference must be published on or before {`cutoff_date`}. This is absolutely necessary. Double check every publication's date before referencing it.
- The output format needs to be respected strictly. Do not change the format in any way.
- It is required to copy the in-text citations with their links in Markdown format `[(author, YYYY-MM)](link)` when referring to related work. Links need to be present.

G.2 SCHOLAREVAL

We use the following parameter values to ensure a balance between the system performance, latency, and context window limit concerns:

- Snippet search queries generated per extracted method: 1.
- Snippets returned per query: up to 20.
- Paper search queries generated per contribution statement: 3.
- Papers returned per query: up to 20.
- Papers returned by the recommendations for each seed paper: up to 8.
- Papers returned via references augmentation for each seed paper: up to 10.
- Papers sampled from the final list for pairwise comparison: up to 25.

Additionally, we implement functionality that restricts the retrieved snippets and papers to those published prior to the cutoff date (i.e. the publication date of the paper that the research idea was extracted from).

### G.3 ABLATIONS

In our ablation experiments, we use the same setup and parameters described in subsection G.2.

To ablate the methods and results extraction (MRE), we remove the steps where we extract references from the snippets, download the papers, and extract the methods and results from each. Instead, the snippets are directly summarized and forwarded for final soundness review synthesis.

The ablation of the paper augmentation (PA) is conducted by removing both the recommendation based augmentation and citation based augmentation. As such, only the seed list of papers identified after the initial paper search and relevance assessment are forwarded for pairwise comparison.

Finally, to ablate the pairwise comparison (PC) step, we sample 25 papers from the final augmented paper list and forward them directly to the contribution review synthesis step.

## H EVALUATION RESULTS

### H.1 ADDITIONAL RESULTS AND SIGNIFICANCE ANALYSIS

In this section, we include detailed evaluation results supplementing the results we presented in the main paper. Namely, Tables 8, 9, 10, 11, and 12 show the pairwise significance analysis of the coverage results between SCHOLAREVAL variants and every baseline overall and for each discipline. Additionally, Table 13 shows the coverage results across all systems by type, axis, severity. Moreover, we conduct evaluation on SCHOLARIDEAS-AI using o4-mini (the underlying model of o4-mini-deep research) as the model backbone for SCHOLAREVAL, shown in Table 14. These results showcase that using the same underlying model, SCHOLAREVAL still achieves higher coverage of the rubrics in SCHOLARIDEAS. We also conduct an evaluation of the related system that is closest in scope to the contribution module in SCHOLAREVAL, Idea Novelty Checker by (Shahid et al., 2025), which outputs novelty verdicts for ideas followed by short rationales (Table 15). This evaluation contextualizes our system against other scholarly work on idea evaluation and confirms our choice to compare SCHOLAREVAL against strong baseline that output detailed feedback at the same scope as SCHOLAREVAL (i.e. strong LLMs and OpenAI deep research). Finally, table 16 outlines the latency and cost of each of SCHOLAREVAL and baseline systems.

Table 8: Pairwise comparisons for Overall coverage: Welch's t-tests comparing SCHOLAREVAL variants to baselines (n=1076 per model). Two-sided p-values shown; Sig.? indicates SCHOLAREVAL > baseline with $p < 0.05$.

| SCHOLAREVAL | Baseline | Mean$_{SE}$ (SD) | Mean$_{BL}$ (SD) | $p$ | Sig.? |
|---|---|---|---|---|---|
| SCHOLAREVAL Llama-70B | Llama-3.3-70B | 2.04 (1.16) | 1.83 (0.89) | 2.6e-06 | Yes |
| SCHOLAREVAL Llama-70B | GPT-4.1 | 2.04 (1.16) | 2.18 (1.08) | 0.004 | No |
| SCHOLAREVAL Llama-70B | Claude-4-Sonnet | 2.04 (1.16) | 2.18 (1.04) | 0.003 | No |
| SCHOLAREVAL Llama-70B | o4-mini-deep-research | 2.04 (1.16) | 2.28 (1.07) | 6.6e-07 | No |
| SCHOLAREVAL GPT-4.1 | Llama-3.3-70B | 2.72 (1.47) | 1.83 (0.89) | 4.4e-60 | Yes |
| SCHOLAREVAL GPT-4.1 | GPT-4.1 | 2.72 (1.47) | 2.18 (1.08) | 8.3e-22 | Yes |
| SCHOLAREVAL GPT-4.1 | Claude-4-Sonnet | 2.72 (1.47) | 2.18 (1.04) | 2.6e-22 | Yes |
| SCHOLAREVAL GPT-4.1 | o4-mini-deep-research | 2.72 (1.47) | 2.28 (1.07) | 3.4e-15 | Yes |
| SCHOLAREVAL Claude-4-Sonnet | Llama-3.3-70B | 2.77 (1.40) | 1.83 (0.89) | 9.5e-71 | Yes |
| SCHOLAREVAL Claude-4-Sonnet | GPT-4.1 | 2.77 (1.40) | 2.18 (1.08) | 4.0e-27 | Yes |
| SCHOLAREVAL Claude-4-Sonnet | Claude-4-Sonnet | 2.77 (1.40) | 2.18 (1.04) | 8.4e-28 | Yes |
| SCHOLAREVAL Claude-4-Sonnet | o4-mini-deep-research | 2.77 (1.40) | 2.28 (1.07) | 1.7e-19 | Yes |

Table 9: Pairwise comparisons for AI coverage: Welch's t-tests comparing SCHOLAREVAL variants to baselines (n=425 per model). Two-sided p-values shown; Sig.? indicates SCHOLAREVAL > baseline with $p < 0.05$.

| SCHOLAREVAL | Baseline | Mean$_{SE}$ (SD) | Mean$_{BL}$ (SD) | $p$ | Sig.? |
|---|---|---|---|---|---|
| SCHOLAREVAL $_{\text{Llama-70B}}$ | Llama-3.3-70B | 2.06 (1.14) | 1.88 (0.92) | 0.011 | Yes |
| SCHOLAREVAL $_{\text{Llama-70B}}$ | GPT-4.1 | 2.06 (1.14) | 2.21 (1.12) | 0.053 | No |
| SCHOLAREVAL $_{\text{Llama-70B}}$ | Claude-4-Sonnet | 2.06 (1.14) | 2.28 (1.02) | 0.003 | No |
| SCHOLAREVAL $_{\text{Llama-70B}}$ | o4-mini-deep-research | 2.06 (1.14) | 2.35 (1.07) | 1.4e-04 | No |
| SCHOLAREVAL $_{\text{GPT-4.1}}$ | Llama-3.3-70B | 2.84 (1.39) | 1.88 (0.92) | 7.2e-30 | Yes |
| SCHOLAREVAL $_{\text{GPT-4.1}}$ | GPT-4.1 | 2.84 (1.39) | 2.21 (1.12) | 8.1e-13 | Yes |
| SCHOLAREVAL $_{\text{GPT-4.1}}$ | Claude-4-Sonnet | 2.84 (1.39) | 2.28 (1.02) | 4.1e-11 | Yes |
| SCHOLAREVAL $_{\text{GPT-4.1}}$ | o4-mini-deep-research | 2.84 (1.39) | 2.35 (1.07) | 1.2e-08 | Yes |
| SCHOLAREVAL $_{\text{Claude-4-Sonnet}}$ | Llama-3.3-70B | 2.91 (1.34) | 1.88 (0.92) | 2.7e-35 | Yes |
| SCHOLAREVAL $_{\text{Claude-4-Sonnet}}$ | GPT-4.1 | 2.91 (1.34) | 2.21 (1.12) | 5.7e-16 | Yes |
| SCHOLAREVAL $_{\text{Claude-4-Sonnet}}$ | Claude-4-Sonnet | 2.91 (1.34) | 2.28 (1.02) | 3.7e-14 | Yes |
| SCHOLAREVAL $_{\text{Claude-4-Sonnet}}$ | o4-mini-deep-research | 2.91 (1.34) | 2.35 (1.07) | 3.2e-11 | Yes |

Table 10: Pairwise comparisons for Neuroscience coverage: Welch's t-tests comparing SCHOLAREVAL variants to baselines (n=314 per model). Two-sided p-values shown; Sig.? indicates SCHOLAREVAL > baseline with $p < 0.05$.

| SCHOLAREVAL | Baseline | Mean$_{SE}$ (SD) | Mean$_{BL}$ (SD) | $p$ | Sig.? |
|---|---|---|---|---|---|
| SCHOLAREVAL $_{\text{Llama-70B}}$ | Llama-3.3-70B | 2.05 (1.18) | 1.82 (0.86) | 0.004 | Yes |
| SCHOLAREVAL $_{\text{Llama-70B}}$ | GPT-4.1 | 2.05 (1.18) | 2.18 (1.10) | 0.107 | No |
| SCHOLAREVAL $_{\text{Llama-70B}}$ | Claude-4-Sonnet | 2.05 (1.18) | 2.15 (1.08) | 0.209 | No |
| SCHOLAREVAL $_{\text{Llama-70B}}$ | o4-mini-deep-research | 2.05 (1.18) | 2.25 (1.04) | 0.011 | No |
| SCHOLAREVAL $_{\text{GPT-4.1}}$ | Llama-3.3-70B | 2.74 (1.55) | 1.82 (0.86) | 2.0e-24 | Yes |
| SCHOLAREVAL $_{\text{GPT-4.1}}$ | GPT-4.1 | 2.74 (1.55) | 2.18 (1.10) | 5.7e-11 | Yes |
| SCHOLAREVAL $_{\text{GPT-4.1}}$ | Claude-4-Sonnet | 2.74 (1.55) | 2.15 (1.08) | 8.2e-12 | Yes |
| SCHOLAREVAL $_{\text{GPT-4.1}}$ | o4-mini-deep-research | 2.74 (1.55) | 2.25 (1.04) | 1.1e-08 | Yes |
| SCHOLAREVAL $_{\text{Claude-4-Sonnet}}$ | Llama-3.3-70B | 2.55 (1.45) | 1.82 (0.86) | 1.4e-17 | Yes |
| SCHOLAREVAL $_{\text{Claude-4-Sonnet}}$ | GPT-4.1 | 2.55 (1.45) | 2.18 (1.10) | 4.3e-06 | Yes |
| SCHOLAREVAL $_{\text{Claude-4-Sonnet}}$ | Claude-4-Sonnet | 2.55 (1.45) | 2.15 (1.08) | 6.0e-07 | Yes |
| SCHOLAREVAL $_{\text{Claude-4-Sonnet}}$ | o4-mini-deep-research | 2.55 (1.45) | 2.25 (1.04) | 1.1e-04 | Yes |

## H.2 MANUAL AUDIT

Our manual audit of papers referenced in the evaluation reports generated by baseline systems underscores various failure modes that undermine the reliability of these systems for literature-grounded research idea evaluation.

First, our inspection of the links indicates a much high reference invalidity rate than the conservative lowerbound reported in §4.2. For Llama-3.3-70B, some our inspected reports had up to 90% invalid references, and stronger LLMs such as Claude 4 Sonnet had up to 50% reference invalidity. We also observed that this issue is not eliminated by retrieval, as 22% of the links included in an inspected report generated by GPT-4o-search-preview were invalid.

We have also noticed subtler failure modes. In the example below, Claude 4 Sonnet references a paper by Gong et al. However, upon inspection, we notice that the linked paper is in fact authored by Bakalarski et al. (2016).

> While [(Gong et al., 2016-08)](`https://www.nature.com/articles/nbt.3621`) demonstrated DNA-barcoded antibodies for multiplexed imaging, their approach relied on conventional chemical conjugation methods that can compromise antibody function.

These errors in attribution are also made by GPT-4o-search-preview. In the example below, it wrongly attributes a publication by Brotherton & Balskus (2013) to Kazane et al.

Table 11: Pairwise comparisons for Biochemistry coverage: Welch's t-tests comparing SCHOL-AREVAL variants to baselines (n=147 per model). Two-sided p-values shown; Sig.? indicates SCHOLAREVAL > baseline with $p < 0.05$.

| SCHOLAREVAL | Baseline | Mean$_{SE}$ (SD) | Mean$_{BL}$ (SD) | $p$ | Sig.? |
|---|---|---|---|---|---|
| SCHOLAREVAL $_{Llama-70B}$ | Llama-3.3-70B | 2.04 (1.19) | 1.81 (0.90) | 0.094 | No |
| SCHOLAREVAL $_{Llama-70B}$ | GPT-4.1 | 2.04 (1.19) | 2.14 (1.04) | 0.460 | No |
| SCHOLAREVAL $_{Llama-70B}$ | Claude-4-Sonnet | 2.04 (1.19) | 2.01 (1.01) | 0.827 | No |
| SCHOLAREVAL $_{Llama-70B}$ | o4-mini-deep-research | 2.04 (1.19) | 2.08 (1.06) | 0.772 | No |
| SCHOLAREVAL $_{GPT-4.1}$ | Llama-3.3-70B | 2.61 (1.42) | 1.81 (0.90) | 1.3e-06 | Yes |
| SCHOLAREVAL $_{GPT-4.1}$ | GPT-4.1 | 2.61 (1.42) | 2.14 (1.04) | 0.003 | Yes |
| SCHOLAREVAL $_{GPT-4.1}$ | Claude-4-Sonnet | 2.61 (1.42) | 2.01 (1.01) | 1.2e-04 | Yes |
| SCHOLAREVAL $_{GPT-4.1}$ | o4-mini-deep-research | 2.61 (1.42) | 2.08 (1.06) | 5.5e-04 | Yes |
| SCHOLAREVAL $_{Claude-4-Sonnet}$ | Llama-3.3-70B | 2.64 (1.40) | 1.81 (0.90) | 3.7e-07 | Yes |
| SCHOLAREVAL $_{Claude-4-Sonnet}$ | GPT-4.1 | 2.64 (1.40) | 2.14 (1.04) | 0.001 | Yes |
| SCHOLAREVAL $_{Claude-4-Sonnet}$ | Claude-4-Sonnet | 2.64 (1.40) | 2.01 (1.01) | 5.2e-05 | Yes |
| SCHOLAREVAL $_{Claude-4-Sonnet}$ | o4-mini-deep-research | 2.64 (1.40) | 2.08 (1.06) | 2.4e-04 | Yes |

Table 12: Pairwise comparisons for Ecology coverage: Welch's t-tests comparing SCHOLAREVAL variants to baselines (n=190 per model). Two-sided p-values shown; Sig.? indicates SCHOLAREVAL > baseline with $p < 0.05$.

| SCHOLAREVAL | Baseline | Mean$_{SE}$ (SD) | Mean$_{BL}$ (SD) | $p$ | Sig.? |
|---|---|---|---|---|---|
| SCHOLAREVAL $_{Llama-70B}$ | Llama-3.3-70B | 1.94 (1.13) | 1.74 (0.87) | 0.054 | No |
| SCHOLAREVAL $_{Llama-70B}$ | GPT-4.1 | 1.94 (1.13) | 2.13 (1.00) | 0.083 | No |
| SCHOLAREVAL $_{Llama-70B}$ | Claude-4-Sonnet | 1.94 (1.13) | 2.11 (1.03) | 0.126 | No |
| SCHOLAREVAL $_{Llama-70B}$ | o4-mini-deep-research | 1.94 (1.13) | 2.35 (1.11) | 3.8e-04 | No |
| SCHOLAREVAL $_{GPT-4.1}$ | Llama-3.3-70B | 2.52 (1.51) | 1.74 (0.87) | 2.2e-09 | Yes |
| SCHOLAREVAL $_{GPT-4.1}$ | GPT-4.1 | 2.52 (1.51) | 2.13 (1.00) | 0.003 | Yes |
| SCHOLAREVAL $_{GPT-4.1}$ | Claude-4-Sonnet | 2.52 (1.51) | 2.11 (1.03) | 0.002 | Yes |
| SCHOLAREVAL $_{GPT-4.1}$ | o4-mini-deep-research | 2.52 (1.51) | 2.35 (1.11) | 0.212 | No |
| SCHOLAREVAL $_{Claude-4-Sonnet}$ | Llama-3.3-70B | 2.90 (1.39) | 1.74 (0.87) | 8.1e-20 | Yes |
| SCHOLAREVAL $_{Claude-4-Sonnet}$ | GPT-4.1 | 2.90 (1.39) | 2.13 (1.00) | 1.6e-09 | Yes |
| SCHOLAREVAL $_{Claude-4-Sonnet}$ | Claude-4-Sonnet | 2.90 (1.39) | 2.11 (1.03) | 9.3e-10 | Yes |
| SCHOLAREVAL $_{Claude-4-Sonnet}$ | o4-mini-deep-research | 2.90 (1.39) | 2.35 (1.11) | 2.6e-05 | Yes |

> These techniques have been widely used to evaluate the impact of conjugation on antibody performance, ensuring that the modifications do not adversely affect antigen-binding capabilities [(Kazane et al., 2013)](`https://pubs.acs.org/doi/10.1021/ja312154m`).

We also observed that o4-mini-deep-research commits similar attribution errors. In the example below, it attributes CyCIF (Lin et al., 2016) and CODEX (Kuswanto et al., 2023) to the wrong authors.

> The proposed system could show improved signal (via HCR) and straightforward reagent generation (via MaMBA). The plan's demonstration of 12-plex imaging is on par with existing methods like CyCIF (Gerdes et al. 2013) or CODEX (Goltsev et al. 2018), indicating strong competitive impact.

These observations showcase the limitations of even strong baselines (web-connected LMs and deep research systems) in generating reliable literature-backed research idea evaluations.

# I  ADDITIONAL DETAILS FROM THE EXPERT USER STUDY

## I.1  SETUP AND MATERIALS

**Rubric** We include our full rubric for the user study in Table 17. Answers were collected via Google Forms.

Table 13: Coverage results of different variants of SCHOLAREVAL and baselines across type, axis, and severity dimensions (mean scores with standard deviations).

| System | Type | | Axis | | Severity | |
|---|---|---|---|---|---|---|
| | Strength | Weakness | Soundness | Contribution | Major | Minor |
| Llama-3.3-70B | 2.63 (0.82) | 1.64 (0.80) | 1.72 (0.84) | 2.14 (0.95) | 1.90 (0.88) | 1.72 (0.90) |
| GPT-4.1 | 3.21 (1.05) | 1.94 (0.95) | 2.00 (0.99) | 2.71 (1.16) | 2.28 (1.05) | 2.01 (1.12) |
| Claude-4-Sonnet | 3.04 (0.97) | 1.98 (0.95) | 2.06 (1.01) | 2.51 (1.08) | 2.28 (1.03) | 2.01 (1.04) |
| GPT-4o-search-preview | 2.83 (0.98) | 1.69 (0.85) | 1.76 (0.91) | 2.32 (1.05) | 1.99 (0.98) | 1.77 (0.98) |
| o4-mini-deep-research | 2.98 (0.90) | 2.12 (1.04) | 2.17 (1.05) | 2.60 (1.05) | 2.41 (1.05) | 2.08 (1.06) |
| SCHOLAREVAL $_{\text{Llama}}$ | 2.93 (1.19) | 1.83 (1.05) | 1.89 (1.10) | 2.46 (1.21) | 2.16 (1.14) | 1.83 (1.15) |
| SCHOLAREVAL $_{\text{GPT}}$ | 4.16 (1.07) | 2.40 (1.34) | 2.51 (1.43) | 3.36 (1.40) | 2.92 (1.44) | 2.41 (1.45) |
| SCHOLAREVAL $_{\text{Claude}}$ | 3.65 (1.21) | 2.56 (1.36) | 2.61 (1.37) | 3.24 (1.37) | 2.93 (1.38) | 2.51 (1.39) |

Table 14: SCHOLAREVAL coverage on SCHOLARIDEAS-AI using o4-mini as backbone

| System | Coverage (mean $\pm$ std) |
|---|---|
| o4-mini-deep-research | $2.35 \pm 1.07$ |
| SCHOLAREVAL $_{\text{GPT}}$ | $2.84 \pm 1.39$ |
| SCHOLAREVAL $_{\text{Claude}}$ | $2.91 \pm 1.34$ |
| SCHOLAREVAL $_{\text{o4-mini}}$ | $2.78 \pm 1.35$ |

**Recruitment, Demographics, and Compensation.** Participants were recruited via X (Twitter) and graduate department emailing lists internationally. Table 18 shows the number of evaluations completed per discipline, as well as years of experience. Experts were paid $25 per research idea evaluated, with a bonus $10 awarded if they completed written feedback.

**User Interface.** Experts were shown the interface shown in Figure 6 and Figure 7. We give experts a unique research id along with 4 unique idea keys, one for each research idea. Each is linked to a replicable, semi-random, and pre-calculated order of systems. The semi-random component comes from the fact that we forced the random order to include at least 2 of each system to ensure each person had the opportunity to contribute evaluations on either system. The script to regenerate these keys will be released with our code. We ensure validity of system blindness by asking users to guess which system they are evaluating, which has a $-0.12$ pearson correlation ($p = 0.44$) with the actual assigned system.

## I.2   STATISTICAL METHODS

Because we collect multiple research ideas per person, our data is no longer independent, and we cannot use a mean-differences t-test. Instead, we use a Linear Mixed-Effects Model (Raudenbush & Bryk, 2002), which models both fixed effects (SCHOLAREVAL vs deep research) and random effects (ideas within and across participants). This helps to stabilize the ratings across our experts, as some may be consistently higher or lower raters. This additionally helps to account for some variability from resarch idea quality. The Linear Mixed-Effects Model is defined as follows:

$$\mathbf{y} = \mathbf{X}\boldsymbol{\beta} + \mathbf{Z}\mathbf{b} + \boldsymbol{\varepsilon} \tag{1}$$

This adds an additional term, $\mathbf{Z}$, on top of the standard Linear Model to capture the variance of random effects, where $\mathbf{Z}$ is a matrix of shape $n \times m$, where $n$ is the total observations and $m$ is the number of unique participants. Each row is one-hot encoded for the participant who contributed the data point.

## J   SCHOLAREVAL OUTPUT EXAMPLES

In this section we provide an example of a neuroscience research idea from SCHOLARIDEAS along with an excerpt of a method soundness review and contribution dimension generated by SCHOLAREVAL.

Table 15: Coverage of contribution rubrics in SCHOLARIDEAS-AI of Idea Novelty Checker (Shahid et al., 2025)

| System | Coverage (mean (std)) |
|---|---|
| Llama-3.3-70B | 2.22 (1.04) |
| GPT-4.1 | 2.78 (1.15) |
| Claude-4-Sonnet | 2.57 (1.05) |
| GPT-4o-search-preview | 2.36 (1.05) |
| O4-mini-deep-research | 2.66 (1.06) |
| ScholarEval-Llama-3.3 | 2.52 (1.17) |
| ScholarEval-GPT-4.1 | 3.45 (1.34) |
| ScholarEval-Claude-4-Sonnet | 3.34 (1.34) |
| Idea Novelty Checker (Shahid et al., 2025) | 2.27 (1.15) |

Table 16: Latency and cost per run (soundness and contribution) for SCHOLAREVAL and baselines.

| System | Latency (mins) | Cost (USD) |
|---|---|---|
| Llama-3.3-70B | 1.23 | 0.00 |
| GPT-4.1 | 2.90 | 0.03 |
| Claude-4-Sonnet | 7.06 | 0.08 |
| GPT-4o-search-preview | 1.94 | 0.15 |
| o4-mini-deep-research | 3.33 | 0.49 |
| ScholarEval-Llama | 6.63 | 0.00 |
| ScholarEval-GPT | 10.23 | 2.54 |
| ScholarEval-Claude | 12.10 | 3.38 |

---

**Research Idea: Neuroscience Research Idea**

**Problem**

We aim to investigate cortical plasticity at its boundaries by studying information processing in individuals who lost vision at birth or early in life. Total loss of a sensory modality provides a unique opportunity to characterize plasticity at its limits.

Prior work shows that cortical structures activated by vision in sighted brains are recruited for cognitive functions, including braille reading, in visually deprived brains. However, overlap of functional responses alone does not reveal what information these activations represent or their cognitive role.

We will study tactile braille reading in blind participants to clarify the transformation of sensory (hand-dependent) to perceptual (hand-independent) braille letter representations.

**Method**

We will combine fMRI and EEG in a multivariate framework to determine the cortical location and temporal emergence of sensory and perceptual representations, relating them to behavioral similarity ratings.

Participants will read braille letters with either hand via piezo-electric refreshable cells, preventing finger movement artifacts. The stimulus set will include ten braille letters (eight for analysis, two as vigilance targets).

*Multivariate Classification:*

- fMRI voxel patterns → spatial location of sensory representations

- EEG electrode patterns → temporal dynamics of representations

- Perceptual representations: across-hand classification (train on one hand, test on other)

- Sensory representations: within-hand classification minus across-hand classification

**Experiment Design**

*Participants and Stimuli:* Blind individuals (vision loss ≤3 years) will take part in:

- fMRI (N=15)

- EEG (N=11)

- Behavioral similarity ratings (N=19)

Table 17: Questions asked during the expert user study, and their respective dimension, module, and scale.

| Dimension | Module | Scale | Question |
|---|---|---|---|
| IdeaFaithful | Soundness | 1-10 | The number of extracted methods/experiments that are faithful to my original intention |
| IdeaFaithful | Contribution | 1-10 | The number of dimensions that are faithful to my original intention |
| Focus | Soundness/Contribution | 1-10 | The strengths reinforce the most valuable aspects of the proposal |
| Focus | Soundness/Contribution | 1-10 | The weaknesses highlight the most deficient aspects of the proposal |
| Focus | Soundness/Contribution | 1-10 | The suggestions for improvement are the most important aspects to address |
| Focus | Contribution | 1-10 | The criteria used to compare your proposal to related work are based on relevant metrics and/or frameworks and are NOT arbitrary |
| Refine | Soundness/Contribution | 1-10 | The top suggestions for improvement offer valuable, targeted, feasible modifications to your experiment |
| LitEngage | Soundness/Contribution | 1-10 | The evaluation uses detailed comparisons with specific components of relevant literature, rather than providing superficial citations |
| Citations | Soundness/Contribution | YYYY | Please estimate how many citations from your MAIN FIELD of study were new to you and you would use in your work |
| Citations | Soundness/Contribution | YYYY | Please estimate how many citations from OUTSIDE your main field of study were new to you and you would use in your work |
| Useful | Soundness/Contribution | 1-10 | I found the [Module] evaluation useful |
| Useful | Soundness/Contribution | 1-10 | I would use the [Module] evaluation again in the future |

Table 18: User study demographics grouped by domain. This includes number of experts, evaluations, and years of experience per discipline.

| Domain | Experts | Evaluations | 2-4 YoE | 4+ YoE |
|---|---|---|---|---|
| Computer Science | 8 | 21 | 5 | 16 |
| Neuroscience | 4 | 12 | 6 | 6 |
| Chemistry | 4 | 8 | 1 | 7 |
| Ecology | 2 | 5 | 4 | 1 |
| Total | **18** | **46** | 16 | 30 |

Letters (B, C, D, L, M, N, V, Z) presented to left/right index fingers; E, O serve as catch stimuli. *fMRI:* 500ms letter presentations, 2500ms ISI, 16 conditions (8 letters × 2 hands), repeated per run with catch and null trials. A localizer (letters vs. fake letters, both hands) will define ROIs along tactile and sighted reading pathways.

*EEG:* Similar design with 500ms presentations, 500ms ISI, longer catch-trial intervals. 64 channels, standard 10-10 placement.

*Behavioral:* Participants will rate perceived similarity of braille letter pairs (1=similar, 7=different) using adjacent braille cells.

**Analysis Plan**

We will focus on tactile areas (S1, S2, intra-parietal cortex, insula) and sighted reading areas (early visual cortex, V4, lateral occipital complex, letter form area, VWFA).

*Hypotheses:*

1. Sensory representations in tactile processing areas; perceptual representations in sighted reading areas.

2. Sensory representations emerge earlier in time than perceptual representations.

3. Representations correlate with behavioral similarity ratings, showing behavioral relevance.

*Analysis Methods:*

- Region-of-interest and searchlight analyses for fMRI
- Time-resolved classification for EEG
- Non-parametric statistical tests with multiple comparison correction

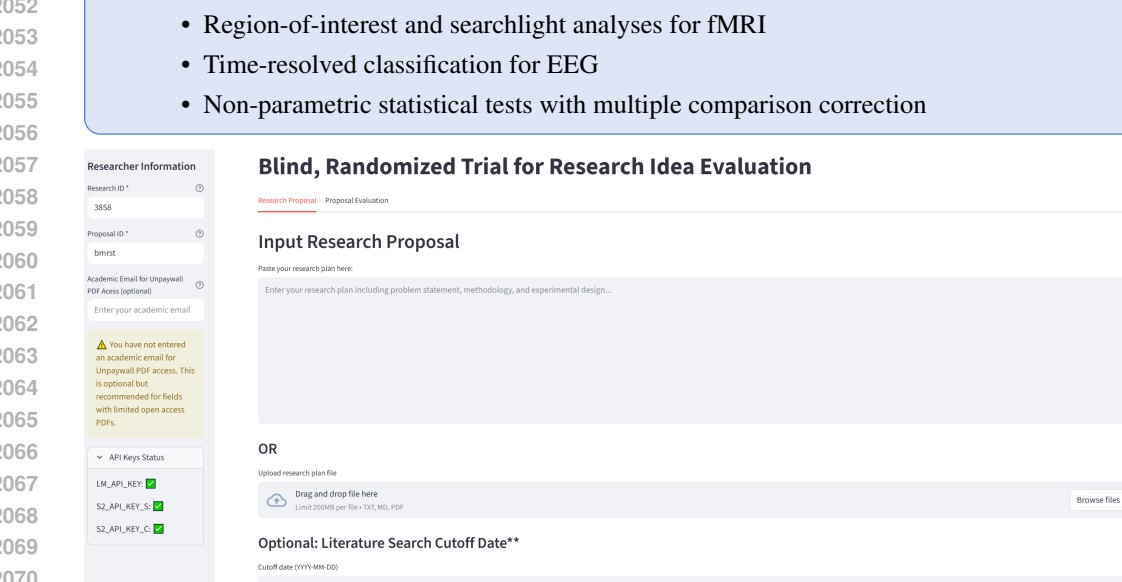

Figure 6: User interface for the Expert User Study. The landing page includes inputs for a research id, proposal id, email for unpaywall, research proposal text box or file upload, and a literature cutoff date.

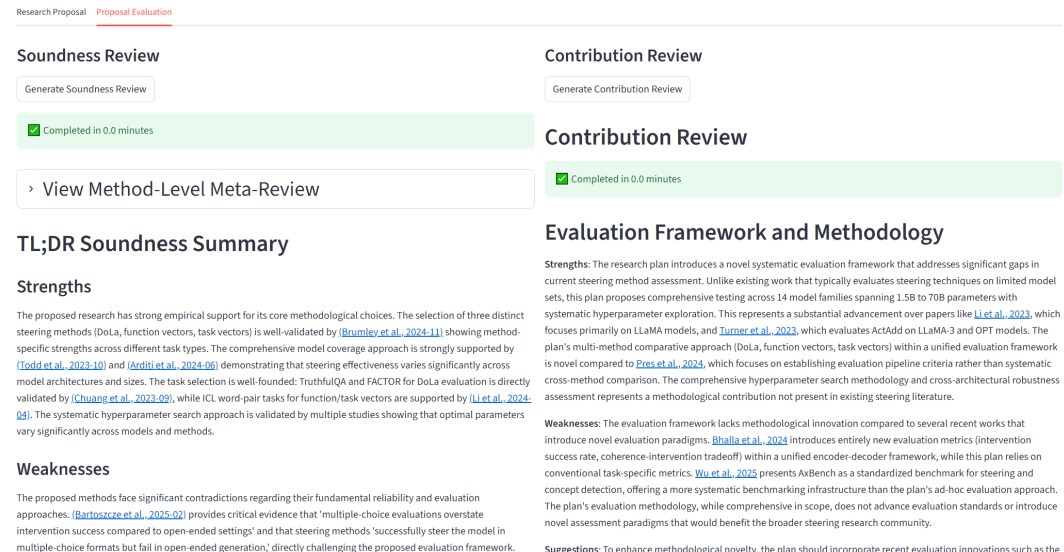

Figure 7: User interface for the Expert User Study, with an example from a user who gave permission to share their output. Reviews for both soundness and contribution are displayed in markdown format. There is a collapsible text box to show the method-level evaluations.

SCHOLAREVAL Example of a Method Soundness Evaluation

[...More method-level reviews...]

**Method: To isolate sensory representations from perceptual ones, we will use a two-step procedure for sensory representations: first conducting within-hand classification (training and testing on the same hand), then subtracting across-hand from within-hand classification results to isolate sensory from perceptual representations.**

*Support*

The proposed two-step subtraction method (within-hand minus across-hand classification) receives strong empirical validation from multiple studies that successfully implemented nearly identical approaches. (Pietrini et al., 2004-04) provides the most direct precedent, employing the exact same methodology with multivariate pattern analysis using support vector machine classifiers to distinguish sensory from perceptual tactile representations in blind participants using refreshable braille cells. Their results demonstrated that within-hand classification captured both sensory and perceptual information while across-hand classification isolated perceptual representations, with the subtraction revealing hand-specific sensory information primarily in somatosensory cortices. (Merabet et al., 2008-08) corroborates this approach with quantitative validation, showing within-hand classification accuracies of 65-72% versus across-hand accuracies of 55-62% in somatosensory regions, yielding meaningful difference scores of 8-15% that were functionally validated through TMS disruption experiments causing 12-18% decreases in tactile discrimination performance. (Grigoryan et al., 2020-06) extends this validation to the temporal domain using EEG, demonstrating that the within-hand versus across-hand distinction emerges early (100-200ms) for sensory representations and later for perceptual ones, directly supporting the proposed method's theoretical foundation. The methodological framework receives additional support from (Amedi et al., 2002-11), which successfully used cross-modal classification to isolate shared representational content, achieving significant above-chance performance in lateral occipital regions and demonstrating the feasibility of using classification approaches to dissociate different types of neural representations. The technical implementation is well-supported by multiple studies using refreshable braille cells with fMRI, including (Burton et al., 2002-12) and (Kim et al., 2017-11), confirming the viability of piezo-electric braille stimulation during neuroimaging. The theoretical rationale is strengthened by extensive evidence for cross-modal plasticity in blind individuals, with (Büchel et al., 1998-03) and (Sadato et al., 1998-07) demonstrating that visual cortical areas are recruited for tactile processing in blind individuals, providing the neurobiological foundation for expecting both sensory and perceptual representations in repurposed visual regions.

*Contradictions*

The proposed subtraction method faces several significant methodological and theoretical challenges that could undermine its validity. Most critically, the fundamental assumption that within-hand classification captures both sensory and perceptual information while across-hand classification captures only perceptual information may be oversimplified and potentially incorrect. The subtraction logic assumes these components are linearly additive and independent, but neural representations are likely to interact nonlinearly, making simple subtraction mathematically invalid. (Rothlein, and Rapp, 2014-04)demonstrates more sophisticated approaches to isolating specific representational features using selectivity analyses, ANOVA comparisons, and regression approaches that account for multiple interacting factors simultaneously, suggesting the proposed binary subtraction approach may be too crude. The method assumes that sensory representations are purely hand-specific and perceptual representations are purely hand-independent, but this dichotomy may not reflect the true complexity of neural coding. Intermediate representations could exist that are partially hand-dependent due to factors like hand-specific motor learning, tactile sensitivity differences, or hemispheric processing asymmetries that are not purely sensory. (Grant et al.) illustrates how discrimination thresholds vary systematically across conditions, suggesting that what appears to be 'sensory' versus 'perceptual' may actually reflect a continuum of processing stages rather than discrete categories. The statistical validity is questionable because the subtraction of two noisy classification accuracies will amplify noise and reduce statistical power, potentially leading to unreliable difference scores. (Siuda-Krzywicka et al., 2016-03) used correlation-based representational similarity analysis with proper statistical controls ($F(2,56)=14.53$, $p¡0.001$), demonstrating more rigorous approaches to comparing representational similarities that account for statistical dependencies.

The method lacks validation against ground truth - there is no independent way to verify that the residual from subtraction actually represents 'pure' sensory information rather than artifacts, noise, or other confounding factors. The approach also fails to account for potential differences in classification difficulty between within-hand and across-hand conditions that could arise from factors unrelated to sensory versus perceptual distinctions, such as differences in signal-to-noise ratios, statistical power, or systematic biases in the classification algorithms.

*Suggested Action*

The proposed method requires substantial methodological improvements to address its fundamental limitations. First, implement multiple complementary analysis approaches rather than relying solely on subtraction, including the selectivity analyses demonstrated by (Rothlein, and Rapp, 2014-04) such as ANOVA selectivity comparisons and regression analyses that can isolate specific representational features while controlling for confounding factors. Second, establish proper statistical validation by implementing permutation testing with scrambled labels (as in the 1000-iteration approach used by (Merabet et al., 2008-08)) to determine chance-level performance for the subtraction scores, and use cluster-based correction for multiple comparisons as demonstrated by (Grigoryan et al., 2020-06). Third, incorporate representational similarity analysis following (Siuda-Krzywicka et al., 2016-03) to examine correlations between neural patterns and behavioral similarity ratings, providing independent validation of the proposed sensory versus perceptual distinction. Fourth, add control analyses to test the method's assumptions, including examining whether the subtraction approach yields consistent results when applied to different classification algorithms, testing whether intermediate levels of hand-dependence exist by examining classification performance across different finger positions or stimulation intensities, and validating that observed differences are not due to systematic biases in classification difficulty. Fifth, implement functional validation similar to (Merabet et al., 2008-08) by using TMS or other perturbation methods to test whether regions identified as containing 'sensory' representations actually causally contribute to hand-specific tactile processing. Sixth, expand the analysis beyond binary classification to examine the full representational geometry using techniques like multidimensional scaling or hierarchical clustering to better understand the relationship between sensory and perceptual representations. Finally, include proper control conditions such as non-letter tactile patterns or scrambled braille stimuli to ensure that observed effects are specific to meaningful braille processing rather than general tactile stimulation differences between hands.

[...More method-level reviews...]

---

**SCHOLAREVAL Example of a Dimension Contribution Evaluation**

[...More dimension-level reviews...]

**Dimension: Experimental Design**

*Strengths:* The experimental design presents novelty through its systematic operationalization of sensory versus perceptual representations via hand-dependency. This approach is entirely absent from existing braille research, which typically focuses on stimulus categories Tian et al., 2022, orthographic contractions Liu et al., 2022, or priming effects [Raczy et al., 2019]. The three-experiment structure combining fMRI spatial localization, EEG temporal dynamics, and behavioral similarity ratings provides comprehensive coverage of the research question that surpasses single-modality studies in the field. The specific focus on individual braille letters with controlled hand presentation using piezo-electric cells represents a novel experimental paradigm not found in existing literature, which typically examines words or uses traditional tactile presentation methods.

*Weaknesses:* The experimental design, while novel in its specific implementation, addresses questions that overlap substantially with existing cross-modal plasticity research. Studies already examine how visual cortex processes tactile information in blind individuals ([de Borst de Gelder, 2018]; [Silson et al., 2021], and the basic framework of studying braille processing across different brain regions is well-established (Liu et al., 2023). The design's focus on letter-level processing, while providing experimental control, may limit ecological validity compared to word-level studies that better reflect natural braille reading behavior.

*Suggestions:* The design could be strengthened by including a direct comparison condition with sighted individuals performing analogous visual letter discrimination tasks to better isolate the effects of visual deprivation. Adding a longitudinal component to track how hand-independence develops with braille reading experience could enhance the design's theoretical contribution. The authors should also consider including more complex braille stimuli (e.g., contracted braille) to bridge the gap between their controlled letter-level approach and real-world braille reading applications demonstrated in other studies (Liu et al., 2022).

[...More dimension-level reviews...]

---

## K LLM USAGE

The authors would like to declare the usage of LLMs for some aspects of code generation, LaTeX formatting and minor cosmetic improvements to the manuscript writing. However, LLMs were not used in the ideation of the project, designing the SCHOLAREVAL framework, or major content writing.

Marwa Abdulhai *et al.*, "Defining Deception in Decision Making"

Philipp Guevorguian *et al.*, "Exploring the Recall of Language Models: Case Study on Molecules"

Seungwon Oh *et al.*, "Recovering Plasticity of Neural Networks via Soft Weight Rescaling"

Yuwei Yan *et al.*, "OpenCity: A Scalable Platform to Simulate Urban Activities with Massive LLM Agents"

Moritz Glaser *et al.*, "ESMGain: Effective and Efficient Prediction of Mutation's functional Effect via ESM2 Transfer Learning and robust Benchmarks"

Svetlana Pavlova *et al.*, "Flow Matching for One-Step Sampling"

Sasan Tavakkol *et al.*, "Less is More: Adaptive Coverage for Synthetic Training Data"

Lorenzo Pacchiardi *et al.*, "100 instances is all you need: predicting LLM success by testing on a few instances"

Zhihan Zhou *et al.*, "GenomeOcean: Efficient Foundation Model for Genome Generation"

Eduardo Sánchez *et al.*, "Linguini: A benchmark for language-agnostic linguistic reasoning"

Zekun Li *et al.*, "MMSci: A Dataset for Graduate-Level Multi-Discipline Multimodal Scientific Understanding"

Alexander Shypula *et al.*, "Does Instruction Tuning Reduce Diversity? A Case Study Using Code Generation"

Naman Gupta *et al.*, "MAC-CAFE: Multi-actor, Centralized Critic Architecture for Feedback-driven Editing"

Magnus Müller *et al.*, "Large-Scale Multi-Agent Reinforcement Learning for Traffic Signal Optimization"

Aohan Sun *et al.*, "Mitigating Privacy Risk of Adversarial Examples with Counterfactual Explanations"

Gwok-Waa Wan *et al.*, "GenBen: A Genarative Benchmark for LLM-Aided Design"

Kyeongrok Park *et al.*, "Toward Human-Interpretable Explanations in a Unified Framework for GNNs"

Wenjie Tang *et al.*, "StarCraft II Arena: Evaluating LLMs in Strategic Planning, Real-Time Decision Making, and Adaptability"

Chen Gao *et al.*, "EmbodiedCity: A Benchmark Platform for Embodied Agent in Real-world City Environment"

Sungmin Han *et al.*, "Improving Transformer Interpretability with Activation Contrast-Based Attribution"

Hong Xie *et al.*, "Multiple-play Stochastic Bandits with Prioritized Resource Sharing"

Santiago Yeomans *et al.*, "From Abstract Noise to Architectural Form: Designing Diffusion Models for Efficient Floor Plan Generation"

Chinmay Mittal *et al.*, "FCoReBench: Can Large Language Models Solve Challenging First-Order Combinatorial Reasoning Problems?"

Haihong Yang *et al.*, "scKGOT: Intercellular Signaling Inference with Knowledge Graph Optimal Transport for Single-cell Transcriptomics"

Changliang Zhou *et al.*, "ICAM: Rethinking Instance-Conditioned Adaptation in Neural Vehicle Routing Solver"

Yisheng Xiao *et al.*, "Path Selection Makes BERT-family Good Generators"

Xiang Liu *et al.*, "ChunkKV: Semantic-Preserving KV Cache Compression for Efficient Long-Context LLM Inference"

Teng Yan *et al.*, "Vision-Based Pseudo-Tactile Information Extraction and Localization for Dexterous Grasping"

Pavel Strashnov *et al.*, "Towards Robust Evaluation of Protein Generative Models: A Systematic Analysis of Metrics"

Zhenlei Wang *et al.*, "Robust Heterogeneous Treatment Effect Estimation under Covariate Perturbation"

Yao Shiyi *et al.*, "BID: Broad Incremental for Android Malware Detection"

Leo McKee-Reid *et al.*, "Honesty to Subterfuge: In-Context Reinforcement Learning Can Make Honest Models Reward Hack"

Tingzhou Wei *et al.*, "Multivariate Time-series Forecasting with SPACE: Series Prediction Augmented by Causality Estimation"

Anuradha Kumari *et al.*, "ZEPHYR GAN: REDEFINING GAN WITH FLEXIBLE GRADIENT CONTROL"

Andrei Chertkov *et al.*, "Tensor Train Decomposition for Adversarial Attacks on Computer Vision Models"

Zhenghan Chen *et al.*, "Advancing Drug-Target Interaction Prediction via Graph Transformers and Residual Protein Embeddings"

Zhenghan Chen *et al.*, "Non-Commutative Spectral Geometry for Adaptive Quantum-Classical Drug-Target Interaction Prediction"

Chris Cameron *et al.*, "Foundation Models for Boolean Logic"

Haoxuan Li *et al.*, "Principle Counterfactual Fairness"

Yuntian Wu *et al.*, "Invariant Spatiotemporal Representation Learning for Cross-patient Seizure Classification"

Benas *et al.*, "Modeled grid cells aligned by a flexible attractor"

Wittkamp *et al.*, "The neural dynamics of positive and negative expectations of pain"

Cui *et al.*, "Dysfunctional S1P/S1PR1 signaling in the dentate gyrus drives vulnerability of chronic pain-related memory impairment"

O'Leary *et al.*, "Natural forgetting reversibly modulates engram expression in hippocampal feed-forward circuits"

Klaassen *et al.*, "Basolateral amygdala inhibition impairs updating of appetitive and aversive values by interacting with the prefrontal cortex"

Haupt *et al.*, "The transformation of sensory to perceptual braille letter representations in the visually deprived brain"

Liu *et al.*, "Cell class-specific long-range axonal projections of neurons in mouse whisker-related somatosensory cortices"

Campbell *et al.*, "Human single-neuron activity is modulated by intracranial theta burst stimulation of the basolateral amygdala"

Derkaloustian *et al.*, "Fine Touch Perception Relies on Frictional Instabilities"

Lee *et al.*, "The influence of temporal context on vision over multiple time scales"

Setogawa *et al.*, "Acquisition of auditory discrimination mediated by different processes through two distinct circuits linked to the lateral striatum"

Cooper *et al.*, "Ultraslow serotonin oscillations in the hippocampus delineate substates across NREM and waking"

Mollá–Albaladejo *et al.*, "Molecular characterization of gustatory second-order neurons reveals integrative mechanisms of gustatory and metabolic information"

Bloem *et al.*, "Dynamic estimation of the attentional field from visual cortical activity"

Xu *et al.*, "Neural Representation of Time across Complementary Reference Frames"

Rieser *et al.*, "Multifaceted Role of Galanin in Whole Brain Excitability"

Lu *et al.*, "The interplay between homeostatic synaptic scaling and homeostatic structural plasticity maintains the robust firing rate of neural networks"

Ecker *et al.*, "Assemblies, synapse clustering and network topology interact with plasticity to explain structure–function relationships of the cortical connectome"

Dash *et al.*, "Rules for reactivation across REM sleep microstates following sensory fear learning"

March *et al.*, "The Hungry Lens: Hunger Shifts Attention and Attribute Weighting in Dietary Choice"

Molkov *et al.*, "Introducing perturbations in point-process models of excitable systems"

Huang *et al.*, "Neural coding of multiple motion speeds in visual cortical area MT"

Liu *et al.*, "Striatal Crosstalk Between Dopamine and Serotonin Systems"

Kang *et al.*, "Rapid rebalancing of co-tuned ensemble activity in the auditory cortex"

Praegel *et al.*, "Age and Learning Shapes Sound Representations in Auditory Cortex During Adolescence"

Zhang *et al.*, "Oxytocin restores context-specific hyperaltruistic preference"

Hall *et al.*, "A cortical–hippocampal communication undergoes rebalancing after new learning"

Zhang *et al.*, "Humans underestimate their body mass in microgravity"

Barnby *et al.*, "Self–other generalisation shapes social interaction and is disrupted in borderline personality disorder"

Tardiff *et al.*, "Normative evidence weighing and accumulation in correlated environments"

Wu *et al.*, "The Self-Interest of Adolescents Overrules Cooperation in Social Dilemmas"

Chen *et al.*, "Synchronous Ensembles of Hippocampal CA1 Pyramidal Neurons During Novel Exploration"

Wang *et al.*, "The relationship between cognitive abilities and mental health as represented by cognitive abilities at the neural and genetic levels of analysis"

Zhong *et al.*, "Modular DNA Barcoding of Nanobodies Enables Multiplexed in situ Protein Imaging and High-throughput Biomolecule Detection"

Majhi *et al.*, "Non-autonomous cell redox-pairs dictate niche homeostasis in multi-lineage stem populations"

Krwawicz *et al.*, "Introduction of cytosine-5 DNA methylation sensitizes cells to oxidative damage"

Xiu *et al.*, "Action mechanism of a novel agrichemical quinofumelin against *Fusarium graminearum*"

Chang *et al.*, "Cancer cells differentially modulate mitochondrial respiration to alter redox state and enable biomass synthesis in nutrient-limited environments"

Mohanty *et al.*, "Deep Learning Reveals Endogenous Sterols as Allosteric Modulators of GPCRs"

Wang *et al.*, "Structure and evolution of Alanine/Serine Decarboxylases & S-Adenosylmethionine Decarboxylases in plants"

Wei *et al.*, "Crystal structure and catalytic mechanism of PL35 family glycosaminoglycan lyases with an ultrabroad substrate spectrum"

Govorunova *et al.*, "Blue-shifted ancyromonad channelrhodopsins for multiplex optogenetics"

He *et al.*, "Coordinated regulation of chemotaxis and resistance to copper by CsoR"

Jandu *et al.*, "Membrane mimetic thermal proteome profiling (MM-TPP) enables proteome-wide target engagement in membranes"

Chong *et al.*, "Establishing the foundations for a data-centric AI approach for virtual drug screening"

Schulze *et al.*, "Effects of residue substitutions on the cellular abundance of proteins "

Maus *et al.*, "Screening the MMV Pathogen Box reveals the mitochondrial bc1-complex as a drug target in mature *Toxoplasma gondii* bradyzoites"

Lefroncois *et al.*, "The Role of ATP Synthase Subunit e (ATP5I) in Mediating the Metabolic and Antiproliferative Effects of Biguanides"

Ntourmas *et al.*, "Endogenous oligomer formation underlies DVL2 condensates and promotes Wnt/$\beta$-catenin signaling"

Marks *et al.*, "Determining the off-target activity of antibiotics and novel translation initiation sites in mitochondria"

Leanza *et al.*, "Increased bone inflammation in type 2 diabetes and obesity correlates with Wnt signaling downregulation and reduced bone strength"

Zhang *et al.*, "Distinct mechanisms of inhibition of Kv2 potassium channels by tetraethylammonium and RY785"

Luo *et al.*, "Isobaric crosslinking mass spectrometry technology for studying conformational and structural changes in proteins and complexes"

Antenucci *et al.*, "Reassessing the substrate specificities of the major *Staphylococcus aureus* peptidoglycan hydrolases lysostaphin and LytM"

Zhou *et al.*, "Structural insights into human propionyl-CoA carboxylase . . ."

Liu *et al.*, "Genome-wide mapping of native co-localized G4s and R-loops . . ."

D'Oliveira *et al.*, "Recognition and Cleavage of Human tRNA . . ."

Rucci *et al.*, "Effects of blood meal source and seasonality on reproductive traits of *Culex quinquefasciatus* (Diptera: Culicidae)"

García-Ruiz *et al.*, "Fitness drivers of division of labor in vertebrates"

Nakagawa *et al.*, "An illusion of a macroecological law, abundance–occupancy relationship in birds"

Jiang *et al.*, "Assessing plant phenological changes based on drivers of spring phenology"

Howard–Spink *et al.*, "Old age variably impacts chimpanzee engagement and efficiency in stone tool use"

Rebindaine *et al.*, "Developmental constraints mediate the summer solstice reversal of climate effects on European beech bud set"

Smit *et al.*, "Risk-taking incentives predict aggression heuristics in female gorillas"

Croijmans *et al.*, "Strip cropping shows promising increases in ground beetle community diversity compared to monocultures"

Wang *et al.*, "Loss of olfaction reduces caterpillar performance and increases susceptibility to a natural enemy"

Tao *et al.*, "Partitioning changes in ecosystem productivity by effects of species interactions in biodiversity experiments"

Yang *et al.*, "Interpreting prediction intervals and distributions for biologically meaningful effects"

Gao *et al.*, "Pesticide-induced resurgence in brown planthoppers is mediated by action on a suite of genes that promote juvenile hormone biosynthesis . . ."

Fargeot *et al.*, "Genetic diversity affects ecosystem functions across trophic levels . . ."

Seltzer *et al.*, "Female Moths Incorporate Plant Acoustic Emissions into Their Oviposition Decision-Making Process"

Seguchi *et al.*, "Vasopressin 1a receptor antagonist disrupts male–male affiliative relationships formed by triadic cohabitation in large-billed crows"

Gatt *et al.*, "Integrating microscopy and transcriptomics from individual eukaryotic plankton (Ukiyo-e-Seq)"

Rydhmer *et al.*, "Automating an insect biodiversity metric using distributed optical sensors: an evaluation across Kansas, USA cropping systems"

Diaz–Colunga *et al.*, "Full factorial construction of synthetic microbial communities"

Zhang *et al.*, "Neuropeptide bursicon and its receptor mediate the transition in seasonal polyphenism of *Cacopsylla chinensis*"

Zhang *et al.*, "Birds migrate longitudinally in response to the resultant Asian monsoons of the Qinghai–Tibet Plateau uplift"

