# OpenReview forum: "ScholarEval: Research Idea Evaluation Grounded in Literature"
_ICLR.cc/2026/Conference — ICLR 2026 Conference Withdrawn Submission_

### Official Review · Reviewer_btKk · 2025-10-26

**Soundness:** 2
**Presentation:** 3
**Contribution:** 2
**Rating:** 4
**Confidence:** 3

**Summary:**

This paper presents a retrieval-augmented evaluation framework named SCHOLAREVAL to evaluate the research ideas by the soundness and contribution, and an expert-verified dataset named SCHOLARIDEAS of 117 research ideas and 1076 review rubrics from four disciplines. The framework adopts a multi-stage pipeline, starting with finding relevant papers of a research idea and synthesizing the existing findings, then measuring the soundness and contribution by evaluating the method components in existing work and comparing the contribution to related work. The experiments show that the proposed framework evaluates the ideas more similarly to human reviews on the proposed dataset.

**Strengths:**

1. This paper contributes a framework to evaluate research ideas and a dataset of ideas and reviews verified by human experts. The framework evaluates the ideas from two aspects and also provides feedback. It shows superiority in multiple aspects over several baselines.

2. The dataset includes human reviews collected from public platforms and covers four disciplines: artificial intelligence, neuroscience, biochemistry, and ecology.

3. The experiments evaluate the proposed framework using various LLMs to show its effectiveness across LLMs. The evaluation measures on various metrics and includes human verification.

**Weaknesses:**

1. The evaluation of soundness and contribution is dependent on the performance of LLMs in retrieving and comparing with prior work, which might be tied to LLM capacity and undermine the evaluation of soundness for ideas of new tasks or novel evaluation.

2. The soundness evaluation works by extracting the method components and evaluating their effectiveness in prior work. This module seems to overlook the connections or interplay between the method components or the significance of model components in different contexts or assumptions.

3. It's unclear if unsuccessful ideas are included in the proposed datasets in Sec 3.1 and whether the framework is capable of distinguishing them from successful ones.

**Questions:**

1. How to ensure an exhaustive search of relevant literature in the framework? As AI papers might be open-sourced largely, does it apply to other disciplines to retrieve papers?

2. How is the efficiency of the framework to evaluate a research idea?

3. The standard deviation of coverage for the proposed method in Table 1 seems to be larger than the baselines across disciplines and variants. Does that indicate an unstable performance of the proposed method?

---

> ### Author Response · Authors · 2025-11-20
> **Response to reviewer btKk (Part 1)**
>
> We thank reviewer btKk for taking the time to engage with our work and appreciating our framework, our dataset and the diverse disciplines it includes, as well as our evaluation which includes various metrics and human verification.
>
> ### W1: *"The evaluation of soundness and contribution is dependent on the performance of LLMs in retrieving and comparing with prior work, which might be tied to LLM capacity and undermine the evaluation of soundness for ideas of new tasks or novel evaluation."*
>
> This is similar to W1 from reviewer qnee. We copy our response below and are glad to further engage in discussion on this point.
>
> We openly discuss this in the limitations section of our manuscript (Appendix A: “Limitations of ScholarEval”). ScholarEval is intentionally designed as a literature grounded framework to ensure the traceability of the provenance of all evaluation claims. If no techniques with any margin of similarity have been attempted in literature, the idea is inherently more difficult to evaluate. Any attempt to circumvent literature grounded judgement would put more reliance on the parametric knowledge of the underlying model, which might lead to poor results since current LLMs still do not have the required domain knowledge to judge the soundness and contribution of research ideas without grounding context, as we have discovered in our initial experimentation. Hence, we view this limitation can be addressed in the future at the level of the underlying model (i.e. how can we improve the parametric knowledge of models in specific domains), and this is indeed an exciting direction to pursue in future work. Moreover, one might also argue that it can be hard for humans as well to judge the effectiveness of truly novel ideas at the ideation stage without relevant knowledge of the literature, i.e. before obtaining any results that might affect judgment. And there are examples from the real-world where truly novel ideas were rejected by human reviewers, such as the highly influential paper introducing dropout (Hinton et al. [1]) which was rejected from NIPS in 2012. Additionally, truly novel ideas represent a slim minority of papers (this study [2] shows that only around 11% of papers can be considered truly novel). Therefore, literature grounded evaluation provides improved interpretability, circumvents current limitations in domain knowledge of frontier models, and reflects real human behavior in evaluating novelty.
>
> ### W2: *"The soundness evaluation works by extracting the method components and evaluating their effectiveness in prior work. This module seems to overlook the connections or interplay between the method components or the significance of model components in different contexts or assumptions."*
>
> This is a very interesting comment that we have previously discussed among authors. Currently, our ScholarEval framework grounds each method evaluation to the broader research idea context by incorporating the research idea in the methods and results summarization step (step 3 of the soundness workflow). Therefore, despite the focus on evaluating one method at a time, some interplay is caught given the evaluation is grounded across all methods. More explicitly capturing the complex capability to reason about the interplay between methods is an exciting avenue for future iterations of the system.
>
> [1] Hinton, G. E., Srivastava, N., Krizhevsky, A., Sutskever, I., & Salakhutdinov, R. R. (2012). Improving neural networks by preventing co-adaptation of feature detectors. arXiv preprint arXiv:1207.0580.
>
> [2] Wang, J., Veugelers, R., & Stephan, P. (2016). Bias against novelty in science: A cautionary tale for users of bibliometric indicators (NBER Working Paper No. 22180). National Bureau of Economic Research.
>
> ### *(Rebuttal continued in next response)*

---

> ### Author Response · Authors · 2025-11-20
> **Response to reviewer btKk (Part 2)**
>
> ### W3: *"It's unclear if unsuccessful ideas are included in the proposed datasets in Sec 3.1 and whether the framework is capable of distinguishing them from successful ones."*
>
> Yes, our dataset ScholarIdeas does include unsuccessful ideas. The AI subset was taken from ICLR 2025 rejected submissions, which we have identified in our manual filtering to contain papers in which the reviews pointed to issues in the underlying idea. This signals that the idea is unsuccessful as opposed to accepted submissions which generally are based on sound and novel ideas with poor execution. Our second data source - eLife, which is a journal hosting papers and public reviews in many life sciences, including biochemistry, neuroscience, and ecology - rates submissions based on two criteria: 1) the significance of the findings and 2) the strength of the evidence. We sampled papers with varying ratings along these criteria, which include unsuccessful ideas.
>
> However, our ScholarEval system is not designed to give final verdicts or scores for research ideas. Instead we focus on rich textual evaluations with valuable, targeted, feasible suggestions for improving the research idea. We provide two justifications for posing idea evaluation as dense feedback and refinement, rather than scoring and discriminating.
>
> (1) While scoring and discrimination might provide a preliminary signal for the calibration of our system, we focus on more difficult, fine-grained, and informative evaluation metrics. For example, the coverage of critiques mentioned by human reviewers and multifaceted questions during our expert user study (detailed in Table 15 in Appendix I: Additional Details from the Expert User Study).
>
> (2) Scoring and discrimination invites a needle-in-a-haystack approach, where good ideas can be surfaced amongst a sea of bad ideas. This can be inefficient and disregard potentially good ideas that need slight refinement. Dense, actionable feedback allows any suggested idea to receive refinement, improves efficiency, and paves a path towards iterative improvement and reinforcement learning with dense rewards.
>
> ### Q1:*"How to ensure an exhaustive search of relevant literature in the framework? As AI papers might be open-sourced largely, does it apply to other disciplines to retrieve papers?"*
>
> As mentioned by the reviewer, we overcame two challenges when designing our literature searching modules.
>
> (1) For fields with many open-source documents, how can we retrieve the most exhaustive set of relevant papers?
>
> (a) We illustrate challenge 1 with the contribution module. The retrieval is done in a two step fashion to ensure exhaustiveness: first, we retrieve initial papers from the Semantic Scholar Papers API, which indexes over 200M academic papers across all fields of study. We then construct a set of highly relevant seed papers for a second round of retrieval. The seed papers are augmented with the Semantic Scholar recommendations API and the papers’ citation graph, followed by a final round of relevance assessment. Additional details can be referenced in the “Paper Discovery” paragraph in section 2.2 of the main text.
>
> (2) For fields with less open-access, how can we ensure the most comprehensive retrieval?
>
> (a) We illustrate challenge 2 with the soundness module. After retrieving the initial snippets using Semantic Scholar snippet search, a host of 285.6M passages from 11.7M full-text papers, we use Unpaywall API to find the full text of the papers referenced in the snippets. Unpaywall is the premier API for finding open access versions of a paper based on its DOI, which is critical for domains with less open access manuscripts.. We further expand on this detail in Appendix C.1 and it is among the primary mechanisms to ensure that ScholarEval is able to perform exhaustive search on more closed-source domains, such as Neuroscience.
>
> Finally, no search is truly exhaustive, and in practice including more papers is limited by the difficulty current language models have with reasoning over long contexts. We believe our extensive usage of search APIs, unpaywall open access, paper recommenders, citation traversal, and relevance assessment provide a strong basis for literature retrieval given the limitations mentioned.

---

> ### Author Response · Authors · 2025-11-20
> **Response to reviewer btKk (Part 3)**
>
> ### Q2: *"How is the efficiency of the framework to evaluate a research idea?"*
>
> We believe this question is asking about the cost and latency of the framework. Here is our response if so:
> As we mention in the Limitations section (Appendix A) running an evaluation using ScholarEval can take up to 12 mins and cost up to $3 (due to model API cost). As we point out to in our response to W2 by reviewer qnee, like many extensive literature grounded frameworks, ScholarEval can incur non-trivial API costs and latency. For example, PaperQA [1], a literature synthesis system, also incurs up to 3 USD in a single run, as do multiple literature understanding systems included in the comprehensive Asta benchmark [2] - Asta Scholar QA for example incurs up to 2.93 USD. Furthermore, our framework is model agnostic and can be used with any underlying open-source model with a cost of 0 USD, and 3 USD only represents an upper bound due to the API cost of current frontier proprietary models. As open-source models are catching up, we expect the cost of our framework to drop significantly. In terms of the latency, ~12mins represents a similar waiting time to current deep research systems. Furthermore, although this might limit the use of ScholarEval in settings requiring low latency, we imagine that this is still an acceptable latency to evaluate a research idea and can definitely be used as an evaluator in autonomous scientific agents as it is negligible relative to the overall runtime of such agents that can reach up to 12 hours [3]. Moreover, the latency in our system is due to rate limits of APIs (Semantic Scholar and LLMs), which could be overcome with better serving.
>
> ### Q3: *"The standard deviation of coverage for the proposed method in Table 1 seems to be larger than the baselines across disciplines and variants. Does that indicate an unstable performance of the proposed method?"*
>
> Thank you for the careful observation - we have taken note that the standard deviation of our method’s coverage is higher. Conducting additional error analysis to uncover failure modes will be extremely important for future extensions of this work. However, based on statistical analysis detailed in Appendix H of our manuscript, ScholarEval still significantly outperforms baselines, and our qualitative analysis and expert user study also indicate clear and significant preference of ScholarEval over other baselines in many other aspects including the validity of the reference, the depth and level of literature engagement, feedback usefulness, and overall quality.
>
> **We thank the reviewer again for engaging with our paper and we hope that our responses above address the reviewer's concerns, and are happy to discuss any further concerns or comments!**
>
> [1] Skarlinski et al. Language Agents Achieve Superhuman Synthesis of Scientific Knowledge. https://arxiv.org/abs/2409.13740
>
> [2] Bragg et al. AstaBench: Rigorous Benchmarking of AI Agents with a Scientific Research Suite. https://arxiv.org/abs/2510.21652
>
> [3] Mitchener et al. Kosmos: An AI Scientist for Autonomous Discovery. https://arxiv.org/abs/2511.02824

---

> ### Author Response · Authors · 2025-11-26
>
> **Dear Reviewer btKk, Thank you again for your review of our work. We understand you might be busy, but we hope you will be able to review our rebuttal. We are happy to clarify or elaborate on any points as needed. If our response addresses your concerns and positively influences your view of the work, we would be grateful if you reflect that in an updated score.**

---

### Official Review · Reviewer_uuD6 · 2025-10-31

**Soundness:** 2
**Presentation:** 2
**Contribution:** 2
**Rating:** 2
**Confidence:** 4

**Summary:**

This paper presents ScholarEval, a retrieval-augmented generation (RAG) framework designed to automatically evaluate research ideas. The framework divides evaluation into two dimensions: "Soundness" (the empirical validity of the method) and "Contribution" (progress relative to previous research). To evaluate the system, the authors constructed a new expert-annotated dataset, ScholarIdeas, which contains 117 research ideas (extracted from published papers) from 4 disciplines (including AI and life sciences) and their corresponding review rubrics.

The main advantages and contributions of the paper are claimed to lie in: (1) proposing a multi-stage evaluation framework that can provide dense and actionable feedback; (2) constructing a new evaluation benchmark, ScholarIdeas; (3) experiments show that ScholarEval significantly outperforms baselines including general LLMs and 04-mini-deep-research in both automatic metrics (covering expert review points) and human evaluation (User Research).

**Strengths:**

1. The problem addressed by the paper—automatically evaluating the quality of research ideas—is a core challenge in the field of AI scientific discovery and holds high value.
2. The paper focuses on generating dense, literature-supported, actionable feedback rather than just a single assessment score. This focus on actionability is commendable.
3. Section 5 of the paper, "Expert User Research," is its most rigorous part. Different from the main body of the paper (which is based on ScholarIdeas evaluation), this study adopted the correct methodology (real experts evaluating fundamental pre-execution ideas) and demonstrated that the system outperforms a key baseline in practical applications.

**Weaknesses:**

- The crisis of effectiveness in the core dataset (ScholarIdeas).
    - Task misalignment: The paper claims to evaluate pre-execution creativity, but the dataset was extracted ex post facto from already completed papers. This represents a fundamental task misalignment.
    - Narrow evaluation scope: The automatic evaluation of the paper relies entirely on ScholarIdeas. The lack of testing on other real-world data (such as ACL 2025, NeurIPS 2025, etc.) makes it impossible to prove the generalization ability of this framework.
    - Timeliness issue: How can we ensure that this data will still be valid next year? Also, isn't the evaluation of 117 ideas too few?
- The baseline comparison is weak and misleading.
    - Baseline is seriously insufficient: The experimental comparison in the paper is seriously inadequate. It avoids comparison with real academic competitors (i.e., other published creativity assessment/novelty assessment tools).
    - Omission of key literature: The paper completely overlooks directly relevant contemporary frameworks such as DeepReview (with its 'deep thinking process') or THE-Tree ('historical evolution'), which are designed to deepen evaluation and reasoning.
    - "Straw Man" Argument: The paper chooses to compare with a general LLM and a commercial Black box system (04-mini-deep-research). This is a "straw man" style of argument and does not prove its superiority over other research work in the field.
- Inherent bias against breakthrough innovation: This framework is entirely "literature-based," which means it is methodologically unable to fairly evaluate truly novel, interdisciplinary, or paradigm-challenging ideas that lack literature precedents.
- Invalid and contradictory evaluation protocols.
    - **Automated evaluation of the loop**: The primary metric "Coverage" measures the degree of match between the system (an LLM pipeline) and the "gold standard" (a benchmark extracted by another LLM), rather than aligning with the true human ability to evaluate creativity.
    - **Unreliable LLM-as-judge**: The paper uses LLM-as-judge (Figure 4) to claim its superiority. However, the validation conducted by the authors themselves in the appendix (Table 6) shows that the agreement between LLM-judge and human annotators is extremely low (e.g., Cohen's Kappa for the "Depth" dimension is -0.0161), rendering its conclusion invalid.

**Questions:**

1. Regarding the dataset and evaluation scope:
    * Can the author explain the rationale for retrospectively extracting "ideas" from published papers to construct ScholarIdeas? This seems to be contrary to the goal of evaluating "pre-execution" ideas.
    * Why is the automatic evaluation limited to the ScholarIdeas dataset? Have the authors considered testing on other contemporary, real-world "testbeds" (such as data from ACL 2025, NeurIPS 2025) to verify the generalization ability of the framework?
2. Regarding the baseline comparison: Could the author explain why all relevant academic baselines (such as DeepReview, THE-Tree, etc.) were omitted from the experimental comparison? Without such a comparison, how can the superiority of ScholarEval over the domain SOTA be verified?
3. Regarding the validity of the evaluation protocol: Given that the appendix (Table 6) shows extremely low agreement (Kappa value close to 0) between LLM-as-judge and human evaluators, why do the authors still use it as the core evidence to support the superiority of the system in Figure 4?

---

> ### Author Response · Authors · 2025-11-20
> **Response to reviewer uuD6 (Part 1)**
>
> We truly thank reviewer uuD6 for the many strengths they highlighted in our paper, including that we address a core challenge in the field of AI for scientific discovery, that our focus on generating literature-supported, dense, and actionable feedback is commendable, and for appreciating the rigor of our expert user study.
>
> We address all of the reviewers concerns in the points below:
>
> ### W1/Q1: *"Regarding the dataset and evaluation scope: Can the author explain the rationale for retrospectively extracting "ideas" from published papers to construct ScholarIdeas? This seems to be contrary to the goal of evaluating "pre-execution" ideas. Why is the automatic evaluation limited to the ScholarIdeas dataset? Have the authors considered testing on other contemporary, real-world "testbeds" (such as data from ACL 2025, NeurIPS 2025) to verify the generalization ability of the framework? How can we ensure that this data will still be valid next year? Also, isn't the evaluation of 117 ideas too few?"*
>
> **Regarding "the rationale for retrospectively extracting "ideas" from published papers"**:
>
> The focus in our paper is the evaluation (soundness and contribution) of pre-execution ideas, and we worked carefully with experts to ensure the validity of the extracted research ideas in the ScholarIdeas dataset. We clarify our decisions below (which are also outlined in section 3 of our paper).
>
> (1) The choice to extract from existing research ideas: unfortunately, there is a lack of data-sources for pre-execution research ideas and constructing such a data source from scratch would be excessively laborious, let alone the need to also annotate each research idea with multiple human reviews. However, existing published papers offer a rich resource that can be used to extract research ideas. Moreover, in our initial paper selection phase, we make a serious effort to only include papers from reputable open peer-review platforms (Openreview for AI ideas and eLife for ideas in the life sciences - biochemistry, neuroscience, and ecology) that have multiple reviews showing consensus.
>
> (2) The extraction of the research ideas is not only LLM-based, but we involve six domain experts to ensure the validity of the extracted ideas and corresponding reviews: As highlighted in section 3.1 of our manuscript, after LLM-based extraction of the research ideas, we include an expert validation where the research idea and correspond review are validated by domain experts and corresponding edits are made. Namely, are six domain experts in artificial intelligence, biochemistry, neuroscience, and ecology rigorously checked the following:
>
> (a) That the extracted research idea is a faithful representation of the paper at the ideation/pre-execution phase
>
> (b) That the corresponding review rubrics do not have mentions to execution, results, or paper presentation
>
> (c) That the research idea and review rubrics are consistent (i.e. the review only addresses aspects contained in the research idea).
>
> We provide details about our dataset construction in section 3 of our manuscript and expert validation in Appendix E.3. Namely, our expert validators made changes to the research ideas 13.40% of the time and to the review rubrics 79.10%, highlighting the significant effort put into ensuring the validity of the (research idea, review rubrics) instances in our dataset and our aim to ensure the alignment between actual research ideas and extracted ones.
>
> **Regarding the “narrow evaluation scope”:**
>
> We use ScholarIdeas for the automatic evaluation, and **ScholarIdeas itself is extracted from real-world data**. Namely, as outlined in our response to the previous point, ScholarIdeas is constructed by extracting research ideas from real-world papers that have accompanying high-quality human-written reviews. Specifically, the **artificial intelligence ideas were extracted from ICLR 2025** submissions and their accompanying public reviews.  The 40 AI research ideas included ScholarIdeas cover many subdomains of AI including BioML, Robotics, Interpretability, Optimization, Privacy, Embodied Environments, Foundation Models, and more, suggesting strong coverage of evaluation topics. Moreover, **ACL reviews are not publicly released**, making them impossible to utilize. **NeurIPS reviews are typically released for accepted papers only**, which makes these reviews less suitable for balanced evaluation. For biochemistry, neuroscience, and ecology research ideas, **we use the eLife journal which also hosts real-world submissions in different fields of the life sciences** and gives access to the human written reviews for all submissions. Hence, **our evaluation specifically uses real-world data from four domains and two reviewing platforms, suggesting the generalization ability of our framework.** More details on our dataset creation process can be viewed in section 3 of the main text in our manuscript.
>
> ### *(Rebuttal to W1/Q1 continued in next response)*

---

> ### Author Response · Authors · 2025-11-20
> **Response to reviewer uuD6 (Part 2)**
>
> ### W1/Q1 (Continued): *"Regarding the dataset and evaluation scope: Can the author explain the rationale for retrospectively extracting "ideas" from published papers to construct ScholarIdeas? This seems to be contrary to the goal of evaluating "pre-execution" ideas. Why is the automatic evaluation limited to the ScholarIdeas dataset? Have the authors considered testing on other contemporary, real-world "testbeds" (such as data from ACL 2025, NeurIPS 2025) to verify the generalization ability of the framework? How can we ensure that this data will still be valid next year? Also, isn't the evaluation of 117 ideas too few?"*
>
> **Regarding the “timeliness issue”:**
>
> **We ensure the continued validity of our dataset by implementing a literature search cutoff date** (mentioned in line 268 of our manuscript). Specifically, the publication date of each of the extracted research ideas is used as a literature cutoff date during the execution of ScholarEval. This ensures that ScholarEval only has access to literature prior to the publication date of the paper corresponding to that idea - i.e. it cannot fetch the paper itself or any subsequent papers that refer to it or to its execution results. Concerning the risk of contamination after dataset release, this is a common issue for all datasets and benchmarks, that can only be alleviated by using live benchmarks, and using our dataset creation pipeline, we could potentially update our dataset with newer publications periodically.
>
> **Regarding the size of our dataset:**
>
> Our dataset contains 117 ideas, each with multiple review rubrics representing individual points of criticism expressed by a human reviewer. **Across our whole dataset, we have 1076 review rubrics. Our coverage metric for each baseline and version of ScholarEval is the average coverage score (rated from 1 to 5) across all these 1076 rubrics.** This represents a **large enough sample that yields statistically significant results.** We believe that future efforts should focus on expanding the domain coverage rather than the size of existing domains. Prior work tends to focus on only one domain (AI), and we have expanded this to four, greatly improving coverage and usability for scientists of many backgrounds.

---

> ### Author Response · Authors · 2025-11-20
> **Response to reviewer uuD6 (Part 3)**
>
> ### W2/Q2: *"Regarding the baseline comparison: Could the author explain why all relevant academic baselines (such as DeepReview, THE-Tree, etc.) were omitted from the experimental comparison? Without such a comparison, how can the superiority of ScholarEval over the domain SOTA be verified?"*
>
> We acknowledged multiple academic attempts at research idea evaluation in our related work section, and appreciate the additional sources provided by the reviewer. **Although these works address the general problem of research idea evaluation, the angles that they take and their system outputs make them unsuitable baselines. We discuss each related academic work we are aware of - as well as the ones the reviewer shared - in detail in the points below:**
>
> - **Baek et al. [1]** introduce a review agent as a component in their idea generation system. However, this is a **simple prompting based component based on GPT 3.5 which is similar to the prompting baselines we have already included**.
> - **Shahid et al. [2]** introduce Idea Novelty Checker (which is part of the Scideator ideation system in Radensky et al. [3]). **This idea checker only focuses on novelty evaluation, and only outputs novel/not novel verdicts accompanied by short rationales.** Our ScholarEval, on the other hand, gives long form evaluations of ideas based on soundness and contribution with references to relevant literature.
> - **Feng et al. [4]** introduce GraphEval, which is a graph-based LLM framework that only scores research ideas. Hence **it does not give natural language feedback that we can compare against our system** in terms of coverage, reference invalidity, depth, actionability, and evidence support.
> - **Wen et al. [5]** introduce a **method to predict the most promising idea among a pair, without any feedback**. Hence, just like Feng et al. [4], it is not suitable as a baseline in our setting, where we are evaluating open-form natural language feedback.
> - **DeepReview:** thank you for bringing this paper to our attention. However, **DeepReview focuses on generating reviews of completed papers and not pre-execution ideas as is the case for ScholarEval**. We have not included paper review systems as our baselines since they focus on an entirely different task. Paper review generation systems expect manuscripts representing fully executed ideas, and generate reviews addressing the significance of the results, the paper presentation etc., making them misaligned with our evaluation dataset containing pre-execution ideas. However, **we will add a section in our related work discussing paper review works and cite DeepReview and similar papers**, since it’s a closely related research direction.
> - **THE-tree**:  thank you for bringing this paper to our attention. This is a *concurrent* work that seems more closely aligned with our scope (research idea evaluation). We were curious to experiment with it but realized that the github repository linked in the paper is empty https://github.com/Auto-THE/THE-Tree-show.
>
> Out of all these systems, the most sensible one to include could be the Idea Novelty Checker by Shahid et al. [2], which is similar to our contribution module but gives significantly shorter rationales. We have evaluated it on ScholarEval-AI and measured the contribution coverage (since this is a novelty only system) and we compare it with all other baselines and our system. As shown below, it only performs slightly better than our weakest baseline Llama-3.3-70B, which makes sense since this system is designed to classify ideas along with short rationales, hence it is not suitable to compare alignment with detailed human-written contribution assessments
>
> **New evaluation results: Coverage (based on contribution rubrics only) on ScholarIdeas-AI of our existing baselines, versions of ScholarEval, and Idea Novelty Checker (Shahid et al. [2])**
>
> | System                               | Coverage (mean (std)) |
> |--------------------------------------|------------------------|
> | Llama-3.3-70B                        | 2.22 (1.04)           |
> | GPT-4.1                              | 2.78 (1.15)           |
> | Claude-4-Sonnet                      | 2.57 (1.05)           |
> | GPT-4o-search-preview                | 2.36 (1.05)           |
> | O4-mini-deep-research                | 2.66 (1.06)           |
> | ScholarEval-Llama-3.3                | 2.52 (1.17)           |
> | ScholarEval-GPT-4.1                  | 3.45 (1.34)           |
> | ScholarEval-Claude-4-Sonnet          | 3.34 (1.34)           |
> | **Idea Novelty Checker (Shahid et al. [2])** | 2.27 (1.15)       |
>
> We also encourage the reviewer to see part 3 or our response to reviewer EwMQ, in which we further explain the rationale behind the baselines we chose.

---

> ### Author Response · Authors · 2025-11-20
> **Response to reviewer uuD6 (Part 4)**
>
> ### W2 (Continued): *Concerning ""Straw Man" Argument: The paper chooses to compare with a general LLM and a commercial Black box system (04-mini-deep-research). This is a "straw man" style of argument and does not prove its superiority over other research work in the field."*
>
> We provide the following rationales for choosing our baselines: In our evaluation **we focus on strong systems that are capable of generating detailed feedback in the same form as ScholarEval**. Namely, we only choose the **strongest open-source and closed-source non-reasoning models, which are Llama-3.3-70B, GPT-4.1, Claude-4-Sonnet.** As for the **deep research system, we have opted for o4-mini-deep-research since it is the only suitable deep research system available through API**. Other deep research systems are available through web interface only, meaning we cannot conduct large scale evaluation on them using our ScholarIdeas dataset. Moreover, **strong frontier LLMs and deep research are systems that actual humans are likely to use to evaluate their research ideas, hence showing performance improvements over them highlights the practical utility of our framework.**
>
> ### W3: *"Inherent bias against breakthrough innovation: This framework is entirely "literature-based," which means it is methodologically unable to fairly evaluate truly novel, interdisciplinary, or paradigm-challenging ideas that lack literature precedents."*
>
> We openly discuss this in the limitations section of our manuscript (Appendix A: “Limitations of ScholarEval”). ScholarEval is intentionally designed as a literature grounded framework to ensure the traceability of the provenance of all evaluation claims. If no techniques with any margin of similarity have been attempted in literature, the idea is inherently more difficult to evaluate. Any attempt to circumvent literature grounded judgement would put more reliance on the parametric knowledge of the underlying model, which might lead to poor results since current LLMs still do not have the required domain knowledge to judge the soundness and contribution of research ideas without grounding context, as we have discovered in our initial experimentation. Hence, we view this limitation can be addressed in the future at the level of the underlying model (i.e. how can we improve the parametric knowledge of models in specific domains), and this is indeed an exciting direction to pursue in future work. Moreover, one might also argue that it can be hard for humans as well to judge the effectiveness of truly novel ideas at the ideation stage without relevant knowledge of the literature, i.e. before obtaining any results that might affect judgment. And there are examples from the real-world where truly novel ideas were rejected by human reviewers, such as the highly influential paper introducing dropout (Hinton et al. [6]) which was rejected from NIPS in 2012. Additionally, truly novel ideas represent a slim minority of papers (this study [7] shows that only around 11% of papers can be considered truly novel). Therefore, literature grounded evaluation provides improved interpretability, circumvents current limitations in domain knowledge of frontier models, and reflects real human behavior in evaluating novelty.
>
> ### W4: Concerning "Coverage measures the degree of match between the system (an LLM pipeline) and the "gold standard" (a benchmark extracted by another LLM), rather than aligning with the true human ability to evaluate creativity."
>
> As we have clarified in our response concerning the dataset creation (and in section 3 of our manuscript), the **gold standard is not LLM-generated**. Rather, these are based on the true human ability to evaluate creativity. Specifically, for each paper used to create ScholarIdeas, we also collect the public reviews that the paper received and do the following:
>
> (1) only keep papers for which the reviews are high quality and clearly address aspects related to the idea behind the paper rather than mere criticisms of the paper execution and presentation
>
> (2) only keep instances where the reviewers are showing consensus about the quality of the idea to ensure that these represent reliable ground truths
>
> (3) use a strong LLM (Claude-4-Sonnet) to extract the points in the review that address the research idea verbatim and drop all other points, hence this is just an extractive process and not a generative one
>
> (4) involve six domains experts in the disciplines we cover (artificial intelligence, biochemistry, neuroscience, and ecology) to ensure that the extracted reviews represent what the human reviewers included and nothing more - in addition to ensuring the correctness of the research idea extracted from the paper and the consistency between the research idea and reviews.
>
> So in conclusion, our **coverage metric measures the degree of match between the system outputs and a gold standard representing human expert evaluations of the same idea.**

---

> ### Author Response · Authors · 2025-11-20
> **Response to reviewer uuD6 (Part 5)**
>
> ### W4/Q3: *"The paper uses LLM-as-judge (Figure 4) to claim its superiority. However, the validation conducted by the authors themselves in the appendix (Table 6) shows that the agreement between LLM-judge and human annotators is extremely low (e.g., Cohen's Kappa for the "Depth" dimension is -0.0161), rendering its conclusion invalid."*
>
> Thank you for your comment. We encourage the reviewer to read Response to reviewer EwMQ (Part 2) for a full discussion of how we validate our LLM metrics. Below we discussed the low inter-annotator agreement raised by the reviewer:
>
> The inter-annotator agreement between the LLM and human evaluations, shown in table 6 in our manuscript, show agreement on evidence support and actionability metrics. However, **due the strong class imbalance in the human and LLM evaluations, which both greatly favor ScholarEval, the Cohen’s kappa, specifically for the depth metric, is misleadingly low.** This is the well-documented statistical phenomenon called the kappa paradox [8, 9]. **An agreement metric that is more robust towards class imbalance is Gwet’s Agreement Coefficient (Gwet's AC1) which we now report in the table 6 below** which shows moderate agreement between the human and LLM evaluations (and in the case of actionability close to substantial agreement).
>
> Table 6 (updated with Gwet's AC1 metric): Inter-annotator agreement between human and LLM evaluations. Due to class imbalance resulting from humans and LLMs largely preferring ScholarEval, **we now report Gwet's AC1 metric which is more robust towards class imbalance. The results show moderate to substantial agreement.**
>
> | Metric   |   Percent Agreement | Cohen's kappa | Gwet's AC1 |
> |--------------|-------------------|---------------|------------|
> | Evidence     | 66.67%         | 0.1818       | 0.44757    |
> | Actionability| 72.22%         | 0.1667       | 0.58621    |
> | Depth        | 61.11%         | -0.0161      | 0.38835    |
>
>
> Given the corrected statistical agreement measure (we will update our manuscript to reflect this), and several levels of validation with human judgment, including the strong preference for ScholarEval based on the human annotation in table 7 of our manuscript and the large scale user study with experts showing strong overlap with our automated metrics, we show the reliability of our LLM-based automatic metrics.
>
> **We thank the reviewer again for their careful engagement with our paper. We hope that our responses satisfactorily address the concerns raised, and we would be glad to discuss any remaining questions or comments.**
>
> ### *(References provided in next response)*

---

> ### Author Response · Authors · 2025-11-20
> **Response to reviewer uuD6 (Part 6)**
>
> ### References
>
> [1] Baek et al. ResearchAgent: Iterative research idea generation over scientific literature with large language models. In Luis Chiruzzo, Alan Ritter, and Lu Wang (eds.), Proceedings of the 2025 Conference of the Nations of the Americas Chapter of the Association for Computational Linguistics: Human Language Technologies (Volume 1: Long Papers), pp. 6709–6738, Albuquerque, New Mexico, April 2025. Association for Computational Linguistics. ISBN 979-8-89176-189-6. doi: 10.18653/v1/2025.naacl-long. 342. URL https://aclanthology.org/2025.naacl-long.342/.
>
> [2] Shahid et al.. Literature-grounded novelty assessment of scientific ideas. In Tirthankar Ghosal, Philipp Mayr, Amanpreet Singh, Aakanksha Naik, Georg Rehm, Dayne Freitag, Dan Li, Sonja Schimmler, and Anita De Waard (eds.), Proceedings of the Fifth Workshop on Scholarly Document Processing (SDP 2025), pp. 96–113, Vienna, Austria, July 2025. Association for Computational Linguistics. ISBN 979-8-89176-265-7. doi: 10.18653/v1/2025.sdp-1.9. URL https://aclanthology. org/2025.sdp-1.9/.
>
> [3] Radensky et al. Scideator: Human-llm scientific idea generation grounded in research-paper facet recombination, 2025. URL https://arxiv.org/abs/2409.14634.
>
> [4] Feng et al. Grapheval: A lightweight graph-based LLM framework for idea evaluation. In The Thirteenth International Conference on Learning Representations, 2025. URL https://openreview.net/forum?id=5RUM1aIdok.
>
> [5] Wen et al. Predicting empirical ai research outcomes with language models, 2025. URL https://arxiv.org/abs/2506.00794.
>
> [6] Hinton, G. E., Srivastava, N., Krizhevsky, A., Sutskever, I., & Salakhutdinov, R. R. (2012). Improving neural networks by preventing co-adaptation of feature detectors. arXiv preprint arXiv:1207.0580.
>
> [7] Wang, J., Veugelers, R., & Stephan, P. (2016). Bias against novelty in science: A cautionary tale for users of bibliometric indicators (NBER Working Paper No. 22180). National Bureau of Economic Research.
>
> [8] Dettori et al. 2020. Kappa and beyond: Is there agreement? Global Spine Journal, 10(4), 499–501. https://doi.org/10.1177/2192568220911648
>
> [9] Zec et al. 2017. High agreement and high prevalence: The paradox of Cohen’s kappa. The Open Nursing Journal, 11, 211–218. https://doi.org/10.2174/1874434601711010211

---

> ### Author Response · Authors · 2025-11-26
>
> **Dear Reviewer uuD6, Thank you again for your review of our work. We understand you might be busy, but we hope you will be able to review our rebuttal. We are happy to clarify or elaborate on any points as needed. If our response addresses your concerns and positively influences your view of the work, we would be grateful if you reflect that in an updated score.**

---

### Official Review · Reviewer_EwMQ · 2025-11-01

**Soundness:** 2
**Presentation:** 3
**Contribution:** 2
**Rating:** 2
**Confidence:** 4

**Summary:**

This paper introduces SCHOLAREVAL, a retrieval-augmented framework designed to evaluate the quality of research ideas. The proposed framework consists of two modules to assess ideas: a soundness module for the empirical validity and a contribution module for novelty. Each module synthesizes the review following a retrieval-augmented way. To validate the system, the authors create a new dataset of 117 expert-annotated research ideas and detailed review rubrics from OpenReview and eLife. Experiments show that SCHOLAREVAL outperforms LLM baselines. An expert study is done to highlight its strength.

**Strengths:**

S1: The paper is well written. The structure of the paper is well organized.
S2: The paper contributes a new dataset SCHOLARIDEAS for evaluation of the idea evaluation works.
S3: The proposed framework SCHOLAREVAL outperforms the baselines. An expert study is also conducted and shows a preference for SCHOLAREVAL on all measured dimensions.

**Weaknesses:**

W1: The construction of the dataset needs more consideration. The research ideas used for evaluation are extracted from published papers by LLMs, not preliminary ideas before execution. The extraction is made by removing execution sections of papers to simulate research ideas at an early stage. The gap between the actual ideas and the extracted ones cannot be ignored.
W2: The paper highly relies on LLMs providing evaluation. More is expected on validation of the LLM-based metrics.
W3: The compared baselines are weak. The proposed framework is compared against multiple general LLMs and only one deep research system (o4-mini-deep-research). The Related Work section mentions several papers specifically designed for research idea evaluation but they are not used in the experiments. This makes the experimental results less convincible.

**Questions:**

See weaknesses.

---

> ### Author Response · Authors · 2025-11-20
> **Response to reviewer EwMQ (Part 1)**
>
> We thank reviewer EwMQ for appreciating the contribution of our ScholarEval framework, ScholarIdeas datasets, expert study, and for considering our paper well-written.
>
> We would like to address each of the reviewer’s concerns below:
>
> ### W1: *"The construction of the dataset needs more consideration. The research ideas used for evaluation are extracted from published papers by LLMs, not preliminary ideas before execution. The extraction is made by removing execution sections of papers to simulate research ideas at an early stage. The gap between the actual ideas and the extracted ones cannot be ignored."*
>
> We have made a concerted effort to **ensure the validity of the research ideas in our ScholarIdeas dataset and align the extracted ideas with the actual ones**, which we outline in the following points:
>
> (1) The choice to extract from existing research ideas: unfortunately, there is a lack of data-sources for pre-execution research ideas accompanied by expert reviews. Constructing such a data source from scratch would be excessively laborious, let alone the need to also annotate each research idea with multiple human reviews. However, existing papers, given a careful extraction pipeline (detailed in the point (2) of this response), offer a rich source of research ideas with accompanying peer reviews. Moreover, we only include papers from reputable open peer-review platforms (Openreview (ICLR 2025) for AI ideas and eLife https://elifesciences.org/  for ideas in the life sciences - biochemistry, neuroscience, and ecology), specifically choosing papers where multiple reviews show consensus.
>
> (2) We collaborated with six domain experts to ensure the validity of the LLM-extracted ideas and corresponding reviews. As highlighted in section 3.1 of our manuscript, after LLM-based extraction of the research ideas, the research idea and corresponding review are validated by domain experts and corresponding edits are made. Namely, our six domain experts in artificial intelligence, biochemistry, neuroscience, and ecology rigorously checked the following:
>
> (a) That the extracted research idea is a faithful representation of the paper at the ideation/pre-execution phase
>
> (b) That the corresponding review rubrics do not have mentions to execution, results, or paper presentation
>
> (c) That the research idea and review rubrics are consistent (i.e. the review only addresses aspects contained in the research idea).
>
> Details about our dataset creation process are in section 3 of the main text in our manuscript and we provide full details about the expert validation in Appendix E.3: Expert Validation. Namely, our expert validators made changes to the research ideas 13.40% of the time and to the review rubrics 79.10%, highlighting the significant effort put into ensuring the validity of the (research idea, review rubrics) instances in our dataset and our aim to ensure the alignment between actual research ideas and extracted ones. It’s worth noting that experts were told to only validate ideas and reviews that they felt they were confident about assessing and fell directly in their area of expertise, and ideas that did not satisfy this were eliminated.
>
> ### *(Rebuttal Continued in Next Comment)*

---

> ### Author Response · Authors · 2025-11-20
> **Response to reviewer EwMQ (Part 2)**
>
> ### W2: *"The paper highly relies on LLMs providing evaluation. More is expected on validation of the LLM-based metrics."*
>
> Our paper follows a multifaceted evaluation approach. **In addition to ScholarIdeas, which we release as a replicable evaluation for the community to build on, we additionally ran a large-scale user study involving 18 experts (PhD+) totaling 46 evaluations across four disciplines (artificial intelligence, biochemistry, neuroscience, ecology)**. Aligning with the results of the automatic evaluation, experts significantly preferred ScholarEval over deep research in terms of the usefulness of the citations, the faithfulness to the original research idea, its focus on the most critical aspects of the review, the depth in engaging with literature, and the value of the suggestions to further refine the idea. These results are summarized in table 4 of the main text in our manuscript. We detail the expert user study in section 5 of the main text and provide more information on the expert recruitment and procedure in Appendix I.
>
> **In addition to conducting human evaluation via this user study, the results of which align with and give credence to the LLM-based evaluation, we also ensure the validity of each of the automatic metrics** reported in section 4 in the following ways:
>
> - Coverage metric: To measure the response’s coverage of the expert rubrics, we use Prometheus [1] which is a widely adopted framework in many works that evaluate long form responses [2, 3, 4, 5, etc.] which is shown to effective and aligns with human judgement especially when given references to compare the answer against [1], which we do provide.
>
> - Reference invalidity metric: the reference invalidity metric, i.e. the number of references (citations) in the evaluation report that are hallucinated by the model, is not an LLM-based metric. Is it rather computed by automatically checking the response code returned by the reference links, and we further bolster it with a manual audit of the references which we detail in Appendix H.2.
>
> - Evidence Support, Actionability, and Depth: These metrics use Claude-4-Sonnet as a judge to give pairwise preferences between ScholarEval and our strongest baseline (o4-mini-deep-research). **We have conducted human evaluation on a subset of the data** to validate the LLM-judge, detailed in Appendix F.3. Namely, the human annotators also largely prefer ScholarEval outputs over deep research (in a blinded setting) across the three criteria of Evidence Support, Depth, and actionability, as shown in Table 7, copied below.
>
> Table 7: Human preference results on a sample of 34 report pairs.
>
> | Metric         | ScholarEval | o4-mini-deep-research | Tie  |
> |----------------|---------|------------------------|------|
> | Evidence Support | 70.6  | 14.7                   | 14.7 |
> | Depth            | 79.4  | 11.8                   | 8.8  |
> | Actionability    | 82.4  | 11.8                   | 5.9  |
>
> **We have also reported the inter-annotator agreement between the LLM and human evaluations**, shown in table 6 in our manuscript, which show agreement on evidence support and actionability metrics. However, due the strong class imbalance in the human and LLM evaluations, which both greatly favor ScholarEval, the Cohen’s kappa, specifically for the depth metric, is misleadingly low. This is the well-documented statistical phenomenon called the kappa paradox [6, 7]. An agreement metric that is more robust towards class imbalance is Gwet’s Agreement Coefficient (Gwet's AC1) which we now report in the table below which shows moderate agreement between the human and LLM evaluations (and in the case of actionability close to substantial agreement).
>
> Table 6 (updated with Gwet's AC1 metric): Inter-annotator agreement between human and LLM evaluations. Due to class imbalance resulting from humans and LLMs largely preferring ScholarEval, **we now report Gwet's AC1 metric which is more robust towards class imbalance. The results show moderate to substantial agreement.**
>
> | Metric   |   Percent Agreement | Cohen's kappa | Gwet's AC1 |
> |--------------|-------------------|---------------|------------|
> | Evidence     | 66.67%         | 0.1818       | 0.44757    |
> | Actionability| 72.22%         | 0.1667       | 0.58621    |
> | Depth        | 61.11%         | -0.0161      | 0.38835    |
>
>
> Given the corrected statistical agreement measure (we will update our manuscript to reflect this), and several levels of validation with human judgment, including the strong preference for ScholarEval based on the human annotation and the large scale user study with experts showing strong overlap with our automated metrics, we are confident in the validity of our LLM-based automatic metrics.
>
> ### *(Rebuttal Continued in Next Comment)*

---

> ### Author Response · Authors · 2025-11-20
> **Response to reviewer EwMQ (Part 3)**
>
> ### W3: *The compared baselines are weak. The proposed framework is compared against multiple general LLMs and only one deep research system (o4-mini-deep-research). The Related Work section mentions several papers specifically designed for research idea evaluation but they are not used in the experiments. This makes the experimental results less convincible.*
>
> In our response below **we summarize why each work mentioned in Related Work was not included as a baseline, we provide new evaluation results of the most sensible ones among these showing its performance relative to ScholarEval and other baselines, and rationalize the choice behind our baselines**.
>
> In the Related Work section, we acknowledge the works that have addressed the problem of research idea evaluation and we also mention how their settings and outputs differ from ours, making them unsuitable as baselines. We outline why we did not adopt each of the mentioned works as baselines in the bullet points below:
>
> - **Baek et al. [4]** introduce a review agent as a component in their idea generation system. However, this is a **simple prompting based component based on GPT 3.5 which is similar to the prompting baselines we have already included**.
> - **Shahid et al. [5]** introduce Idea Novelty Checker (which is part of the Scideator ideation system in Radensky et al. [6]). **This idea checker only focuses on novelty evaluation, and only outputs novel/not novel verdicts accompanied by short rationales.** Our ScholarEval, on the other hand, gives long form evaluations of ideas based on soundness and contribution with references to relevant literature.
> - **Feng et al. [7]** introduce GraphEval, which is a graph-based LLM framework that only scores research ideas. Hence **it does not give natural language feedback that we can compare against our system** in terms of coverage, reference invalidity, depth, actionability, and evidence support.
> - **Wen et al. [8]** introduce a **method to predict the most promising idea among a pair, without any feedback**. Hence, just like Feng et al. [7], it is not suitable as a baseline in our setting, where we are evaluating open-form natural language feedback.
>
> Out of all the systems, the most sensible one to include would be the Idea Novelty Checker by Shahid et al. [5], which is similar to our contribution module but gives significantly shorter rationales. We have evaluated it on ScholarEval-AI and measured the contribution coverage (since this is a novelty only system) and we compare it with all other baselines and our system. As shown below, it only performs slightly better than our weakest baseline Llama-3.3-70B, which makes sense since this system is designed to classify ideas along with short rationales, hence it is not suitable to compare alignment with detailed human-written contribution assessments
>
> **New evaluation results: Coverage (based on contribution rubrics only) on ScholarIdeas-AI of our existing baselines, versions of ScholarEval, and Idea Novelty Checker (Shahid et al. [5])**
>
> | System                               | Coverage (mean (std)) |
> |--------------------------------------|------------------------|
> | Llama-3.3-70B                        | 2.22 (1.04)           |
> | GPT-4.1                              | 2.78 (1.15)           |
> | Claude-4-Sonnet                      | 2.57 (1.05)           |
> | GPT-4o-search-preview                | 2.36 (1.05)           |
> | O4-mini-deep-research                | 2.66 (1.06)           |
> | ScholarEval-Llama-3.3                | 2.52 (1.17)           |
> | ScholarEval-GPT-4.1                  | 3.45 (1.34)           |
> | ScholarEval-Claude-4-Sonnet          | 3.34 (1.34)           |
> | **Idea Novelty Checker (Shahid et al. [5])** | 2.27 (1.15)       |
>
> Due to the reasons above, **in our evaluation we focus on strong systems that are capable of generating detailed feedback in the same form as ScholarEval**. Namely, **we only choose the strongest open-source and closed-source non-reasoning models, which are Llama-3.3-70B, GPT-4.1, Claude-4-Sonnet. As for the deep research system, we have opted for o4-mini-deep-research since it is the only suitable deep research system available through API.** Other deep research systems are available through web interface only, meaning we cannot conduct large scale evaluation on them using our ScholarIdeas dataset. **Moreover, strong frontier LLMs and deep research are systems that actual humans are likely to use to evaluate their research ideas, hence showing performance improvements over them highlights the practical utility of our framework.**
>
>
> **We thank the reviewer again for their engagement with our paper. We hope that our responses satisfactorily address the concerns raised, and we would be glad to discuss any remaining questions or comments.**
>
> ### *(References given in next response)*

---

> ### Author Response · Authors · 2025-11-20
> **Response to reviewer EwMQ (Part 4)**
>
> ### References
>
> [1] Kim et al. 2024. Prometheus 2: An Open Source Language Model Specialized in Evaluating Other Language Models. In Proceedings of the 2024 Conference on Empirical Methods in Natural Language Processing, pages 4334–4353, Miami, Florida, USA. Association for Computational Linguistics.
>
> [2] Yifei et al. ResearchQA: Evaluating Scholarly Question Answering at Scale Across 75 Fields with Survey-Mined Questions and Rubrics. https://arxiv.org/abs/2509.00496
>
> [3] Asai et al. OpenScholar: Synthesizing Scientific Literature with Retrieval-Augmented LMs. https://arxiv.org/abs/2411.14199
>
> [4] Shao et al. 2024. Assisting in Writing Wikipedia-like Articles From Scratch with Large Language Models. In Proceedings of the 2024 Conference of the North American Chapter of the Association for Computational Linguistics: Human Language Technologies (Volume 1: Long Papers), pages 6252–6278, Mexico City, Mexico. Association for Computational Linguistics.
>
> [5] Bonomo et al. 2025. LiteraryQA: Towards Effective Evaluation of Long-document Narrative QA . In Proceedings of the 2025 Conference on Empirical Methods in Natural Language Processing (pp. 1729--1731). Association for Computational Linguistics.
>
> [6] Dettori et al. 2020. Kappa and beyond: Is there agreement? Global Spine Journal, 10(4), 499–501. https://doi.org/10.1177/2192568220911648
>
> [7] Zec et al. 2017. High agreement and high prevalence: The paradox of Cohen’s kappa. The Open Nursing Journal, 11, 211–218. https://doi.org/10.2174/1874434601711010211
>
> [8] Baek et al. ResearchAgent: Iterative research idea generation over scientific literature with large language models. In Luis Chiruzzo, Alan Ritter, and Lu Wang (eds.), Proceedings of the 2025 Conference of the Nations of the Americas Chapter of the Association for Computational Linguistics: Human Language Technologies (Volume 1: Long Papers), pp. 6709–6738, Albuquerque, New Mexico, April 2025. Association for Computational Linguistics. ISBN 979-8-89176-189-6. doi: 10.18653/v1/2025.naacl-long. 342. URL https://aclanthology.org/2025.naacl-long.342/.
>
> [9] Shahid et al.. Literature-grounded novelty assessment of scientific ideas. In Tirthankar Ghosal, Philipp Mayr, Amanpreet Singh, Aakanksha Naik, Georg Rehm, Dayne Freitag, Dan Li, Sonja Schimmler, and Anita De Waard (eds.), Proceedings of the Fifth Workshop on Scholarly Document Processing (SDP 2025), pp. 96–113, Vienna, Austria, July 2025. Association for Computational Linguistics. ISBN 979-8-89176-265-7. doi: 10.18653/v1/2025.sdp-1.9. URL https://aclanthology. org/2025.sdp-1.9/.
>
> [10] Radensky et al. Scideator: Human-llm scientific idea generation grounded in research-paper facet recombination, 2025. URL https://arxiv.org/abs/2409.14634.
>
> [11] Feng et al. Grapheval: A lightweight graph-based LLM framework for idea evaluation. In The Thirteenth International Conference on Learning Representations, 2025. URL https://openreview.net/forum?id=5RUM1aIdok.
>
> [12] Wen et al. Predicting empirical ai research outcomes with language models, 2025. URL https://arxiv.org/abs/2506.00794.

---

> ### Author Response · Authors · 2025-11-26
>
> **Dear Reviewer EwMQ,
> Thank you again for your review of our work. We understand you might be busy, but we hope you will be able to review our rebuttal. We are happy to clarify or elaborate on any points as needed. If our response addresses your concerns and positively influences your view of the work, we would be grateful if you reflect that in an updated score.**

---

### Official Review · Reviewer_qnee · 2025-11-04

**Soundness:** 4
**Presentation:** 4
**Contribution:** 4
**Rating:** 6
**Confidence:** 4

**Summary:**

This paper introduces ScholarEval, a novel, retrieval-augmented framework designed for comprehensive evaluation of research ideas, a critical and underexplored area in AI co-science. The system assesses ideas based on two core, literature-grounded criteria: Soundness (empirical validity of proposed methods) and Contribution (advancement across multiple dimensions relative to prior work). The pipeline involves targeted search, information extraction from the full text and abstracts of scholarly literature (Semantic Scholar), and synthesis of dense, actionable, and cited feedback. To benchmark the system, the authors introduce ScholarIdeas, the first expert-annotated, multi-domain dataset for idea evaluation. ScholarEval significantly outperforms strong baselines, including a state-of-the-art agentic system, in terms of review coverage, depth, evidence support, and overall usefulness, as validated by automatic metrics and an expert user study.

**Strengths:**

1. ScholarEval fills a critical gap by offering a system that provides comprehensive, multifaceted evaluation across both methodological soundness and multi-dimensional contribution, moving beyond one-dimensional scoring or sparse feedback.
2. The explicit use of retrieval-augmentation ensures that all claims and suggestions are supported by direct evidence from the literature via citations, resulting in feedback that is demonstrably more actionable, valid, and useful for idea refinement compared to baselines.
3. The system significantly outperforms the strongest baseline (o4-mini-deep-research) by a large margin in coverage of expert review points and is consistently preferred by human experts in terms of depth, actionability, and literature engagement.
4. The introduction of ScholarIdeas (117 ideas, 1076 rubrics across four disciplines) is a major, high-quality contribution to the community, enabling future standardized benchmarking in this area.

**Weaknesses:**

1. The framework's core strength is also its primary vulnerability. It relies entirely on retrieved literature, meaning it may misjudge the effectiveness of a truly novel method not yet represented in the literature or misrepresent the contribution if relevant contrasting papers are missed.
2. The reported high cost (up to $3) and time (around 12 minutes) per evaluation could pose a significant barrier to adoption for large-scale or batch ideation workflows.
3. Dependency and Sensitivity to LLM Backbone: The entire multi-stage process, from method extraction and summarization to synthesis, is highly dependent on the capability of the underlying LLM (M). The paper does not fully isolate the performance gain attributable to the pipeline structure versus the quality of the specific LLM used.

**Questions:**

1. The paper acknowledges that a failure to retrieve relevant papers can misrepresent soundness or contribution. What is the measured recall or F1 score for the crucial "Context Retrieval" step in the Soundness module? Could the authors provide an ablation study that systematically quantifies the impact on final coverage when retrieval is limited (e.g., reducing the number of papers/snippets retrieved) to better understand the system's robustness to retrieval noise or omissions?
2. Given the multi-stage reliance on the LLM, please provide an ablation where the full ScholarEval pipeline is executed using the same (presumably weaker) LLM backbone used for the "deep-research" baseline16. This is crucial to definitively isolate the performance gain stemming from the novel pipeline architecture versus the improved capabilities of the underlying language model (M).

---

> ### Author Response · Authors · 2025-11-20
> **Response to reviewer qnee (Part 1)**
>
> We thank reviewer qnee for the positive feedback and for *finding that our contribution fills a critical gap by offering comprehensive feedback beyond one dimensional scoring or sparse feedback.*
>
> We would like to address each of the reviewer’s comments on weaknesses and questions below:
>
> ### W1: *"The framework's core strength is also its primary vulnerability. It relies entirely on retrieved literature, meaning it may misjudge the effectiveness of a truly novel method not yet represented in the literature or misrepresent the contribution if relevant contrasting papers are missed."*
>
> We openly discuss this in the limitations section of our manuscript (Appendix A: “Limitations of ScholarEval”). ScholarEval is intentionally designed as a literature grounded framework to ensure the traceability of the provenance of all evaluation claims. If no techniques with any margin of similarity have been attempted in literature, the idea is inherently more difficult to evaluate. Any attempt to circumvent literature grounded judgement would put more reliance on the parametric knowledge of the underlying model, which might lead to poor results since current LLMs still do not have the required domain knowledge to judge the soundness and contribution of research ideas without grounding context, as we have discovered in our initial experimentation. Hence, we view this limitation can be addressed in the future at the level of the underlying model (i.e. how can we improve the parametric knowledge of models in specific domains), and this is indeed an exciting direction to pursue in future work. Moreover, one might also argue that it can be hard for humans as well to judge the effectiveness of truly novel ideas at the ideation stage without relevant knowledge of the literature, i.e. before obtaining any results that might affect judgment. And there are examples from the real-world where truly novel ideas were rejected by human reviewers, such as the highly influential paper introducing dropout (Hinton et al. [1]) which was rejected from NIPS in 2012. Additionally, truly novel ideas represent a slim minority of papers (this study [2] shows that only around 11% of papers can be considered truly novel). Therefore, literature grounded evaluation provides improved interpretability, circumvents current limitations in domain knowledge of frontier models, and reflects real human behavior in evaluating novelty.
>
> ### W2: *"The reported high cost (up to 3 USD) and time (around 12 minutes) per evaluation could pose a significant barrier to adoption for large-scale or batch ideation workflows."*
> We have also discussed this in Appendix A: “Limitations of ScholarEval.” in our manuscript. Indeed, like many extensive literature grounded frameworks, ScholarEval can incur non-trivial API costs and latency. For example, PaperQA [3], a literature synthesis system, also incurs up to 3 USD in a single run, as do multiple literature understanding systems included in the comprehensive Asta benchmark [4] - Asta Scholar QA for example incurs up to 2.93 USD. Furthermore, our framework is model agnostic and can be used with any underlying open-source model with a cost of 0 USD, and 3 USD only represents an upper bound due to the API cost of current frontier proprietary models. As open-source models are catching up, we expect the cost of our framework to drop significantly.
> In terms of the latency, ~12mins represents a similar waiting time to current deep research systems. Furthermore, although this might limit the use of ScholarEval in settings requiring low latency, we imagine that this is still an acceptable latency to evaluate a research idea and can definitely be used as an evaluator in autonomous scientific agents as it is negligible relative to the overall runtime of such agents that can reach up to 12 hours [5]. Moreover, the latency in our system is due to rate limits of APIs (Semantic Scholar and LLMs), which could be overcome with better serving.
>
> [1] Hinton, G. E., Srivastava, N., Krizhevsky, A., Sutskever, I., & Salakhutdinov, R. R. (2012). Improving neural networks by preventing co-adaptation of feature detectors. arXiv preprint arXiv:1207.0580.
>
> [2] Wang, J., Veugelers, R., & Stephan, P. (2016). Bias against novelty in science: A cautionary tale for users of bibliometric indicators (NBER Working Paper No. 22180). National Bureau of Economic Research.
>
> [3] Skarlinski et al. Language Agents Achieve Superhuman Synthesis of Scientific Knowledge. https://arxiv.org/abs/2409.13740
>
> [4] Bragg et al. AstaBench: Rigorous Benchmarking of AI Agents with a Scientific Research Suite. https://arxiv.org/abs/2510.21652
>
> [5] Mitchener et al. Kosmos: An AI Scientist for Autonomous Discovery. https://arxiv.org/abs/2511.02824
>
> ### *(Rebuttal continued in next comment)*

---

> ### Author Response · Authors · 2025-11-20
> **Response to reviewer qnee (Part 2)**
>
> ### W3: *"Dependency and Sensitivity to LLM Backbone: The entire multi-stage process, from method extraction and summarization to synthesis, is highly dependent on the capability of the underlying LLM (M). The paper does not fully isolate the performance gain attributable to the pipeline structure versus the quality of the specific LLM used."*
>
> Thank you for expressing the concern that the paper does not fully isolate the performance gain attributable to the pipeline structure versus the quality of the specific LLM used. Although we do not include this as an explicit result in our paper, we include the underlying LLMs that we use as baselines in Table 1 of our manuscript (also copied below). Namely, these are Llama-3.3-70B, GPT-4.1, and Claude-4-Sonnet. As highlighted in our results, using our ScholarEval framework with these backbone models leads to significant coverage improvement compared to each respective model, suggesting that these performance gains are attributable to our pipeline structure. Beyond the coverage metric, ScholarEval is the only system not to suffer from hallucinated citations as shown in Table 2 of our manuscript (also copied below). The barebone models alone suffer from substantial rates of invalid/hallucinated citations. The elimination of this issue is attributable to our pipeline structure as well.
>
> Table 1: Coverage of baselines and variants of ScholarEval overall and per-discipline.  ∗
> and
> †
> indicate significant improvement over 1 or all baselines, respectively. Best results are bolded.
> | Model                     | Overall           | AI                | Neuroscience                 | Biochemistry                 | Ecology                     |
> |---------------------------|-------------------|-------------------|------------------------------|------------------------------|-----------------------------|
> | Llama-3.3-70B             | 1.83 ± 0.89       | 1.88 ± 0.92       | 1.82 ± 0.86                  | 1.81 ± 0.90                  | 1.74 ± 0.87                 |
> | GPT-4.1                   | 2.18 ± 1.08       | 2.21 ± 1.12       | 2.18 ± 1.10                  | 2.14 ± 1.04                  | 2.13 ± 1.00                 |
> | Claude-4-Sonnet           | 2.18 ± 1.04       | 2.28 ± 1.02       | 2.15 ± 1.08                  | 2.01 ± 1.01                  | 2.11 ± 1.03                 |
> | GPT-4o-search-preview     | 1.90 ± 0.98       | 1.95 ± 0.97       | 1.84 ± 0.96                  | 1.86 ± 1.00                  | 1.95 ± 1.02                 |
> | o4-mini-deep-research     | 2.28 ± 1.07       | 2.35 ± 1.07       | 2.25 ± 1.04                  | 2.08 ± 1.06                  | 2.35 ± 1.11                 |
> | ScholarEval$_{\text{Llama}}$  | 2.04 ± 1.16*      | 2.06 ± 1.14*      | 2.05 ± 1.18*                 | 2.04 ± 1.19                  | 1.94 ± 1.13                 |
> | ScholarEval$_{\text{GPT}}$    | 2.72 ± 1.47†      | 2.84 ± 1.39†      | **2.74 ± 1.55†**             | 2.61 ± 1.42†                 | 2.52 ± 1.51†                |
> | ScholarEval$_{\text{Claude}}$ | **2.77 ± 1.40†**  | **2.91 ± 1.34†**  | 2.55 ± 1.45†                 | **2.64 ± 1.40†**             | **2.90 ± 1.39†**            |
>
> Table 2: Rate of reference invalidity across all systems. Values for baselines represent lower bounds, actual reference invalidity is substantially higher, especially for non-retrieval systems. Reference invalidity is not issue in ScholarEval.
> | System                    | Reference Invalidity ↓ |
> |---------------------------|------------------------|
> | Llama-3.3-70B             | 19.07%                 |
> | GPT-4.1                   | 15.22%                 |
> | Claude-4-Sonnet           | 13.90%                 |
> | o4-mini-deep-research     | 1.07%                  |
> | ScholarEval$_{\text{Llama}}$  | 0%                     |
> | ScholarEval$_{\text{GPT}}$    | 0%                     |
> | ScholarEval$_{\text{Claude}}$ | 0%                     |
>
> ### *(Rebuttal continued in next comment)*

---

> ### Author Response · Authors · 2025-11-20
> **Response to reviewer qnee (Part 3)**
>
> ### Q1: *"The paper acknowledges that a failure to retrieve relevant papers can misrepresent soundness or contribution. What is the measured recall or F1 score for the crucial "Context Retrieval" step in the Soundness module? Could the authors provide an ablation study that systematically quantifies the impact on final coverage when retrieval is limited (e.g., reducing the number of papers/snippets retrieved) to better understand the system's robustness to retrieval noise or omissions?"*
>
> In Table 3 of our manuscript (also copied below), we provide the results of ablations of different components of our system, including one that limits the retrieval in the contribution module by removing the paper augmentation phase. We notice performance degradations highlighting the effectiveness and importance of the retrieval steps in our pipeline. Especially salient is a contrast to GPT-4o-search-preview shown in Table 1 (included in our response to W3). Despite having search capabilities, and returning valid citations (as shown in table 2 included in our response to W3), 4o-search performs similar to or worse than non-retrieval baselines. This additionally highlights the importance of not only finding relevant papers, but having a method to condense such extensive context. Furthermore, our LLM baselines in Table 1, which are all outperformed by their ScholarEval counterpart, also represent an ablation of retrieval, since these models did not rely on context retrieval.
>
> Table 3: Ablations of different components of ScholarEval on ScholarIdeas-AI. MRE = Methods and Results Extraction in soundness workflow, PA = Paper Augmentation in contribution workflow, PC = Pairwise Comparison in contribution workflow.
> |                          | Coverage ↑        |
> |--------------------------|-------------------|
> | ScholarEval$_{\text{Claude}}$ | 2.91 ± 1.34       |
> | └─ -MRE                  | 2.47 ± 1.42       |
> | └─ -PA                   | 2.42 ± 1.28       |
> | └─ -PC                   | 2.39 ± 1.23       |
>
>
> ### Q2: *"Given the multi-stage reliance on the LLM, please provide an ablation where the full ScholarEval pipeline is executed using the same (presumably weaker) LLM backbone used for the "deep-research" baseline16. This is crucial to definitively isolate the performance gain stemming from the novel pipeline architecture versus the improved capabilities of the underlying language model (M)."*
>
> Below is the requested ablation where we execute the full ScholarEval pipeline using the deep research model backbone (o4-mini) on ScholarIdeas-AI. We note that using the same backbone (o4-mini) our ScholarEval still significantly outperforms o4-mini-deep-research (p ≪ 0.001), meaning that our performance gains are not merely due to using stronger models (although, GPT-4.1 and Claude-4-Sonnet are not universally better than o4-mini which remains a powerful reasoning model). We had initially not used reasoning models in our pipeline due their higher latency and high cost (although the price per token is cheaper, they generate more reasoning tokens which leads to high execution costs for the large batch). We will include this new ablation in the final paper.
>
> New Ablation: Coverage Results on ScholarIdeas-AI (n = 425 rubrics)
>
> | System                    | Coverage (mean ± std) |
> |---------------------------|------------------------|
> | o4-mini-deep-research     | 2.35 ± 1.07           |
> | ScholarEval-GPT-4.1       | 2.84 ± 1.39           |
> | ScholarEval-Claude-4-Sonnet | 2.91 ± 1.34         |
> | ScholarEval-o4-mini       | 2.78 ± 1.35           |
>
> **We thank the reviewer again for their engagement with our paper. We hope that our responses address the concerns raised, and we are happy to discuss any further questions or comments.**

---

> ### Author Response · Authors · 2025-11-26
>
> **Dear Reviewer qnee,
> Thank you again for your review of our work. We understand you might be busy, but we hope you will be able to review our rebuttal. We are happy to clarify or elaborate on any points as needed. If our response addresses your concerns and positively influences your view of the work, we would be grateful if you reflect that in an updated score.**

---

### Author Response · Authors · 2025-12-04
**Summary of Rebuttals for AC (Part 1)**

First, we would like to thank all reviewers for the strengths they have highlighted in our work, including that our contribution fills a critical gap by offering comprehensive feedback beyond one-dimensional scoring, for appreciating the rigor in our expert user study, and our multifaceted evaluation.

We have provided extensive responses to each concern raised by the reviewers, and were looking forward to further engaging in discussion with them.

We provide the AC with the following summary of the points raised by the reviewers and our responses to them, with pointers to specific parts of the rebuttals where we mention more details.

**Limitations of framework:**

**a. Dependence on Literature and Evaluation of Novel Ideas, raised by reviewers qnee, uuD6, btKk:**

We openly discuss in our limitations that ScholarEval is designed as a literature-grounded framework to ensure the traceability of the provenance of all evaluation claims. Circumventing this would rely on the parametric knowledge of models, which currently lack the required domain knowledge to judge soundness without grounding contexts. We further mentioned that "truly novel" ideas are a minority (approx. 11% of papers), and even humans struggle to judge them without results; for example, the highly influential paper introducing dropout (Hinton et al., 2012) was rejected from NIPS 2012. Therefore, literature-grounded evaluation provides improved interpretability and reflects real human behavior in evaluating novelty.
For further details please refer to our response to Reviewer qnee (Part 1).

**b. Cost and Latency of the Framework, raised by reviewers qnee, btKk:**

This is also a point we openly mention in the Limitations section of our manuscript. While ScholarEval can incur non-trivial API costs and latency, it remains in the same range as other extensive literature-grounded frameworks (e.g., PaperQA and Asta Scholar QA, which incurs up to 2.93 USD). The cost of 3 USD is an upper bound using frontier proprietary models; the framework is model-agnostic and costs 0 USD with open-source models. Regarding latency (~12 mins), this is comparable to current deep research systems and is negligible relative to the runtime of autonomous scientific agents (which can reach up to 12 hours). Furthermore, latency is currently driven by API rate limits, which can be overcome with better serving.
For further details please refer to our response to Reviewer qnee (Part 1).

**Isolation of Pipeline Contribution vs. LLM Capabilities, raised by reviewer qnee:**

To isolate the performance gain attributable to our pipeline structure, we provided a new ablation using the same backbone as the deep research baseline (o4-mini). ScholarEval-o4-mini significantly outperforms o4-mini-deep-research (p << 0.001), highlighting that the performance gains are attributable to the framework design and not the underlying models. *We have added this ablation in Table 14 in Appendix H.1 in our manuscript.* Furthermore, ScholarEval is the only system to eliminate hallucinated citations (0% invalidity), whereas barebone models (e.g., Llama-3.3-70B) suffer from substantial rates of invalid citations (up to 19.07%).
A lot more details are provided in our response to reviewer qnee (Part 3) containing the new ablation results.

**Dataset Construction and Retrospective Extraction, raised by reviewers EwMQ, uuD6:**

Due to the lack of pre-execution data sources, we extracted ideas from papers in reputable venues (ICLR reviews on Openreview, eLife journal reviews) with consensus reviews. To ensure validity, we collaborated with six domain experts who rigorously checked that the extracted ideas faithfully represented the pre-execution phase and that reviews addressed the idea rather than execution. Experts made edits to 13.40% of ideas and 79.10% of rubrics, highlighting the significant effort put into ensuring alignment. We also employ a strict literature search cutoff date based on publication time to prevent data leakage. Details about our dataset creation process are in section 3 of the main text in our manuscript and we provide full details about the expert validation in Appendix E.3: Expert Validation.
For further details on our response to this concern please refer to our response to Reviewer EwMQ (Part 1).

---

### Author Response · Authors · 2025-12-04
**Summary of Rebuttals for AC (Part 2)**

**Choice of Baselines, raised by reviewers EwMQ, uuD6**

In our response to reviewer EwMQ (Part 3) and to reviewer uuD6 (Part 3), we explain in detail why certain works mentioned in the related work section were not used as baselines. Instead, we prioritized baselines capable of generating detailed feedback in the same form as ScholarEval. Related works (e.g., Idea Novelty Checker, GraphEval) often only output scores or binary verdicts. To further illustrate this point, we have provided a new evaluation of Idea Novelty Checker (Shahid et al.) - which is the system that is the closest in scope and hence the most sensible as a baseline. This evaluation shows that Idea Novelty Checker performs only slightly better than our weakest baseline. *We have added this evaluation in Table 15 of Appendix H.1 in our manuscript.* When it comes to our selection of deep research systems, we have selected o4-mini-deep-research as it is the only suitable deep research system available through API for large-scale evaluation.
For further details please refer to our response to Reviewer EwMQ (Part 3) and to Reviewer uuD6 (Part 3).

**Validity of LLM-based Metrics and Human Agreement, raised by reviewers EwMQ, uuD6**

We give a detailed response of how we validate each LLM metric in our response to Reviewer EwMQ (Part 2).

First, we employ an evaluation protocol that goes beyond reliance on LLMs. First, we conducted a large-scale user study with 18 PhD-level experts across four disciplines, who significantly preferred ScholarEval over deep research baselines in terms of faithfulness, depth, and utility, thereby aligning with the results of our automated evaluation.

Second, we validated each of our automated metrics: (1) Coverage metric relies on the Prometheus framework, which has been extensively used in other works evaluating long form responses and is shown to be effective and aligned with human judgment especially when reference rubrics are provided, which is the case in our setting. (2) Reference Invalidity is not an LLM metric. (3) The quality metrics by LLM-as-a-judge (Evidence, Actionability, Depth): blinded human annotators showed a strong preference for ScholarEval (70.6% for evidence support, 79.4% for Depth, and 82.4% for Actionability), aligning with the LLM judge. We have also addressed the concern of low Cohen’s Kappa scores by identifying the "Kappa paradox," a statistical phenomenon caused by strong class imbalance where both humans and models favored our method - which is the case in our setting where both the humans and llm judge strongly prefer ScholarEval. We now report Gwet’s Agreement Coefficient (AC1), a metric robust to this imbalance, which demonstrates moderate to substantial agreement (e.g., Actionability agreement rises from 0.16 to 0.58). *We have updated Appendix F.3 in our manuscript to reflect the corrected inter-annotator agreement.*
For further details please refer to our response to Reviewer EwMQ (Part 2).

---
We thank the AC for the time and effort dedicated to reviewing submissions given the current circumstances of the conference, and we hope that this summary, alongside our detailed responses to the reviewers, showcases that we have thoughtfully addressed all reviewers' concerns and faciliates the assessment of our submission.

Thank you once again for engaging with our work!

Authors of submission 15672

---

### Note · Authors · 2026-01-04

I have read and agree with the venue's withdrawal policy on behalf of myself and my co-authors.